# Suppression of HSF1 activity by wildtype p53 creates a driving force for p53 loss-of-heterozygosity

Tamara Isermann [1], Özge Çiçek Şener [1,5], Adrian Stender[1,5], Luisa Klemke [1], Nadine Winkler[1], Albrecht Neesse[2], Jinyu Li[3], Florian Wegwitz [4], Ute M. Moll[1,3] & Ramona Schulz-Heddergott [1✉]

The vast majority of human tumors with p53 mutations undergo loss of the remaining wildtype p53 allele (loss-of-heterozygosity, p53LOH). p53LOH has watershed significance in promoting tumor progression. However, driving forces for p53LOH are poorly understood. Here we identify the repressive WTp53–HSF1 axis as one driver of p53LOH. We find that the WTp53 allele in AOM/DSS chemically-induced colorectal tumors (CRC) of p53$^{R248Q/+}$ mice retains partial activity and represses heat-shock factor 1 (HSF1), the master regulator of the proteotoxic stress response (HSR) that is ubiquitously activated in cancer. HSR is critical for stabilizing oncogenic proteins including mutp53. WTp53-retaining CRC tumors, tumor-derived organoids and human CRC cells all suppress the tumor-promoting HSF1 program. Mechanistically, retained WTp53 activates *CDKN1A*/p21, causing cell cycle inhibition and suppression of E2F target MLK3. MLK3 links cell cycle with the MAPK stress pathway to activate the HSR response. In p53$^{R248Q/+}$ tumors WTp53 activation by constitutive stress represses MLK3, thereby weakening the MAPK-HSF1 response necessary for tumor survival. This creates selection pressure for p53LOH which eliminates the repressive WTp53-MAPK-HSF1 axis and unleashes tumor-promoting HSF1 functions, inducing mutp53 stabilization enabling invasion.

[1] Institute of Molecular Oncology, University Medical Center Göttingen, Göttingen, Germany. [2] Department of Gastroenterology, Gastrointestinal Oncology and Endocrinology, University Medical Center Göttingen, Göttingen, Germany. [3] Department of Pathology, Stony Brook University, Stony Brook, NY, USA. [4] Department of Gynecology and Obstetrics, University Medical Center Göttingen, Göttingen, Germany. [5] These authors contributed equally: Özge Çiçek Şener, Adrian Stender. ✉email: ramona.schulz-heddergott@med.uni-goettingen.de

The vast majority (>91%) of human tumors with heterozygous *TP53* (tumor protein p53) mutations spanning 32 cancer types, including colorectal cancer (CRC), undergo loss of the remaining wildtype (WT) *TP53* allele (loss-of-heterozygosity, p53LOH) by mutation, chromosomal deletion, or copy-neutral loss-of-heterozygosity[1–4]. Typically, heterozygous tumors present an instable transition state. Importantly, however, the driving forces behind p53LOH largely remain elusive. Thus, p53LOH has watershed significance in promoting tumor progression[5–11].

CRC, the third leading cause of cancer deaths worldwide, is due to several driver mutations. Next to APC, *TP53* mutations are the second most common alteration in sporadic CRC, affecting >60% of cases[5,12–16]. *TP53* mutations enable the critical transition from late adenoma to invasive carcinoma[17,18]. The vast majority of *TP53* alterations are missense (MS) mutations (termed here mutp53), with hotspot codons R175, G245, R248, R273, and R282 (refs. [19–21]). In addition to loss-of-WTp53 function (LOF), some, especially hotspot MS mutp53 alleles, gain broad tumorigenic gain-of-function (GOF), and actively promote cancer progression in vivo[5,7,22–28]. GOF mutants acquire allele-specific functions, not necessarily shared by other mutants[12,29–33]. The GOF *TP53*[R248Q] allele (termed here Q) is one of the most common across cancer types, including CRC[16]. Recently, we showed that mutp53[R248Q] promotes CRC progression and strong invasion in mice, and that mouse and human p53[R248Q/W] mutants bind to and hyperactivate Stat3 to promote GOF, correlating with poor patient survival[5].

A prerequisite for GOF is the tumor-specific stabilization of mutp53 proteins by the HSP90/HSP70/HSP40 chaperone systems[34–38], providing protection from degradation by E3-ubiquitin ligases MDM2 and CHIP[39,40]. HSF1, the master transcription factor of the cytoprotective inducible heat-shock stress response (HSR), governs the expression of stress-induced chaperones, including HSP90, HSP70, and HSP40 and is the major proteotoxic defense in tumors, preventing aberrant oncoproteins from aggregation[41–43]. Moreover, HSF1 induces noncanonical chaperone-independent tumor-promoting genes, together imparting a key co-oncogenic role on HSF1 in tumorigenesis[44–46]. Notably, since cancer cells experience cumulative stress during tumorigenesis, HSF1 is increasingly activated[47]. In support, in the AOM/DSS (azoxymethane/dextran sodium sulfate) mouse model of CRC pharmaceutical inhibition or knockout of HSF1 suppresses colorectal carcinogenesis[48].

A critical prerequisite for mutp53 stabilization is the loss of the remaining WTp53 allele (p53LOH) in heterozygous mutp53/+ tumors. Heterozygous tumors rarely if ever stabilize mutp53 in vivo[5,7–9]. Thus, GOF mutp53 stabilization and p53LOH are strongly linked.

Recent mouse studies also clearly identify p53LOH as strong tumor-promoting force[6,8,10,11]. Our previous work comparing sarcomas and breast cancer identified that heterozygous mutp53 tumors require a second hit for mutp53 stabilization, i.e., loss of the remaining WTp53 allele[8]. Moreover, mouse intestinal tumors carrying heterozygous mutp53[R270H]GOF (mutp53/+) were selected for p53LOH during metastasis[6]. Also, p53LOH was indispensable for cell survival and single-cell clonal expansion of these cancer cells. Furthermore, mutp53/LOH cells had increased tumor-initiating ability compared with mutp53/+ and p53null cells in vivo[6].

It is imperative to understand the mechanism driving stabilization of GOF mutants. The dependency of mutp53 stabilization on p53LOH appears somehow regulated by the remaining WTp53, but its mechanism is unknown. Using a genetically controlled p53LOH system in a CRC model, we show their causal relationship. We identify that the remaining WTp53 allele in p53[R248Q/+] tumors,

despite a partial dominant-negative effect (DNE) by the Q allele, still represses the HSF1 chaperone axis, thereby preventing mutp53[R248Q] stabilization, GOF, and invasion. This creates a strong selection force for p53LOH. Thus, a single pivotal genetic event, p53LOH, simultaneously provides three major evolutionary forces to drive cancer, (I) loss of residual WTp53 suppressor activity, including the repressive WTp53–HSF1 axis, (II) tumor-promoting HSF1 derepression, and (III) mutp53 stabilization enabling GOF activities. This might provide an explanation why p53LOH strongly correlates with mutp53 stabilization and higher tumor aggressiveness.

## Results

**p53LOH is a prerequisite for mutp53 stabilization and invasion in colorectal cancer.** Recently, we demonstrated that stabilized mutp53[R248Q] protein promotes GOF activities, including proliferation and invasion, in the murine CRC AOM/DSS model[5]. Stabilization of mutp53 proteins specifically in tumor, but not normal cells is a key feature and prerequisite of GOF[9,25]. Since p53LOH is a critical prerequisite for mutp53 stabilization in sarcomas and breast cancer[8], we examined whether p53LOH also induces mutp53 stabilization in the colorectal AOM/DSS model[5]. We combined the humanized constitutive GOF *TP53*[R248Q] allele (short "p53Q") with either p53 WT ("+") or knock-out alleles and determined the effect of p53LOH on mutp53 levels. Indeed, massive mutp53 stabilization was detected in 100% of p53[Q/−], but 0% of p53[Q/+] tumors (Supplementary Fig. 1a, b). Notably, p53LOH increased tumor numbers more in the p53[Q/−] than in the p53[−/−] setting (Supplementary Fig. 1c). Importantly, 100% of heterozygous p53[Q/+] and p53[−/+] tumors remain noninvasive (Supplementary Fig. 1d). Conversely, loss of the remaining WTp53 allele not only enables invasion, but does so more strongly in p53[Q/−] tumors (70% invasion) compared to p53[−/−] tumors (20% invasion), validating that p53Q exerts GOF for invasion (Supplementary Fig. 1d, e). Thus, p53LOH is the critical determinant for CRC invasion.

Since AOM/DSS tumors do not undergo spontaneous p53LOH (as manifested by lack of mutp53 stabilization), they represent a tractable system to experimentally manipulate LOH and show that it enables mutp53 stabilization and CRC invasion. To this end, we used an inducible system that combines the p53Q allele with a floxed WTp53 (p53[fl]) allele. p53[Q/fl] mice were crossed to *villinCreER*[T2] mice to generate Tamoxifen (TAM)-inducible p53LOH restricted to intestinal epithelial cells (p53[Q/Δ]), plus "no LOH" controls (p53[Q/+]; Fig. 1a). Importantly, TAM-mediated p53LOH was uniformly induced at a defined colonoscopy-verified tumor burden (Fig. 1b), and successful recombination was validated by gPCR (Supplementary Fig. 1f). Controls were (I) p53[Q/fl] oil-treated mice, and (II) p53[Q/+] TAM-treated mice to exclude nonspecific TAM effects. At 6–8 weeks post TAM, LOH tumors showed a trend toward increased tumor numbers and significant increases in tumor size compared to both "no LOH" control groups (Fig. 1c, d). When analyzed earlier at 3–5 weeks post TAM (Fig. 1e), tumor burden had not yet increased, indicating that LOH's effect, and subsequent mutp53 GOF effects, on promoting proliferation requires time and is incremental.

In contrast to mice with retained WTp53 alleles, all p53LOH mice exhibited stabilized mutp53 (Fig. 1f), which correlated with overexpressed CyclinD1, a proliferation GOF gene in the AOM/DSS model (Supplementary Fig. 1g)[5]. The WTp53 allele is a major barrier to tumor invasion (Supplementary Fig. 1d, e) and as shown in refs. [5,9,49]. In agreement, "no LOH" mice (oil-treated p53[Q/fl] and TAM-treated p53[Q/+]) failed to develop invasive tumors (Fig. 1h), while induced p53LOH dramatically increased invasive tumor numbers per cohort from 0/27 to 18/49 tumors

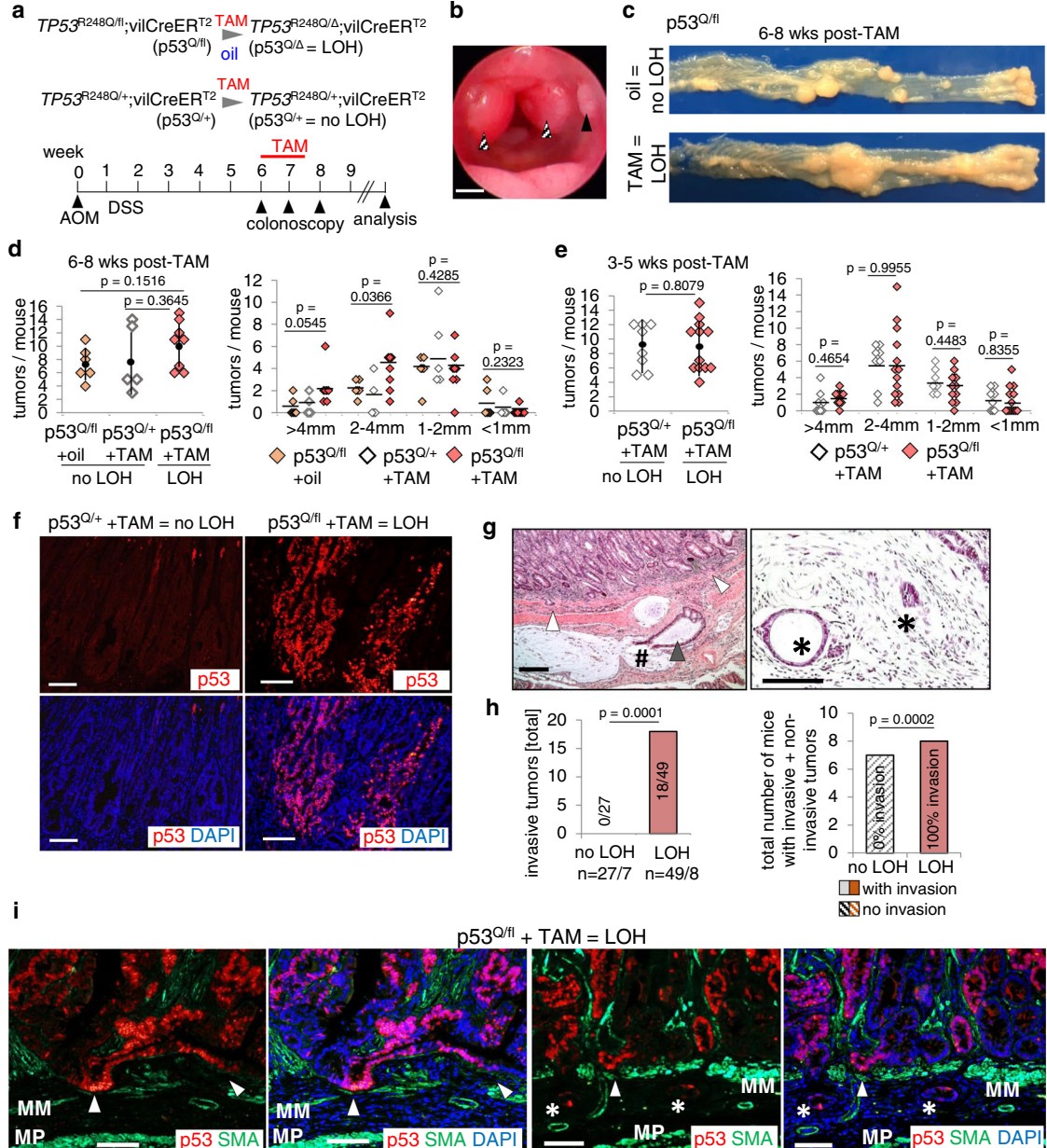

**Fig. 1 p53LOH is a prerequisite for mutp53 stabilization and enables invasion in CRC. a** p53LOH induction scheme in intestinal epithelial cells. The GOF TP53[R248Q] allele ("p53[Q]") is paired with the conditional wild-type Trp53 allele (p53[fl]). A constitutive WTp53 allele ("p53[+]") serves as control. Colorectal tumors were initiated with AOM/DSS. At a defined tumor burden visualized by colonoscopy TAM (Tamoxifen) or oil treatment was administered. Oil-treated p53[Q/fl];vilCreER[T2] and TAM-treated p53[Q/+];vilCreER[T2] control mice carry heterozygous tumors (no LOH), whereas TAM-treated TP53[R248Q/fl];vilCreER[T2] (p53[Q/Δ]) mice carry p53LOH tumors. Mice were analyzed 2–8 weeks after LOH induction. **b** Representative colonoscopy of an untreated TP53[R248Q/+];vilCreER[T2] mouse 8 weeks post AOM. Tumors with scores S2 (solid arrow) and S3 (dashed arrows)[79]. Scale bar, 1 mm. **c** Representative view of dissected colons of p53[Q/fl] mice 6 weeks after p53LOH induction. Left, ileocecal valve; right, anus. **d** Number of tumors per mouse (left) and tumor sizes (right) 6–8 weeks post TAM or oil treatment. Both "no LOH" groups had the same tumor burden. p53[Q/fl] + oil, n = 6; p53[Q/+] + TAM, n = 5; p53[Q/fl] + TAM, n = 8. **e** Number of tumors per mouse (left) and tumor sizes (right) analyzed at 3–5 weeks post TAM. p53[Q/+] + TAM, n = 8; p53[Q/fl] + TAM, n = 13. **f** Representative immunofluorescence of p53 for TAM-treated p53[Q/+] and TAM-treated p53[Q/fl] mice at endpoint 6 weeks post TAM. Scale bars, 100 μm. **g** Representative H&E of two LOH p53[Q/fl] tumors 8 weeks after TAM. Left, extensive invasion within muscularis propria and muscularis mucosae (white arrows). One malignant gland (black arrow) is ruptured, spilling its content into mucous lakes with single tumor cells floating (#). Right, invading glands in submucosa (*). Scale bars, 100 μm. **h** (Left) total numbers of invasive tumors and (right) total numbers of mice with noninvasive and invasive tumors in the "no LOH" group (combined from oil-treated p53[Q/fl] and TAM-treated p53[Q/+] mice) versus the "LOH" group (p53[Q/Δ], TAM-treated p53[Q/fl] mice) analyzed at 6–8 weeks post oil/TAM. (Left) "no LOH", n = 27 total tumors from seven mice and "LOH", n = 49 total tumors from eight mice. (Right) "no LOH" n = 7 mice and "LOH" n = 8 mice. Both Fisher's exact tests. (Left) noninvasive tumors were included in the statistical calculation. Bars, mean. **i** Stabilized mutp53 at the invasive front. Two tumors from different LOH mice 6 weeks post TAM. Representative immunofluorescence, p53 (red), α-SMA (green), and DAPI (blue). MM muscularis mucosae, MP muscularis propria. White arrows, tumor cells invading the MM. Asterix, small invasive cell clumps invading submucosa. Scale bars, 100 μm. **d, e** (Left) dots with vertical black lines represent means ± SD. (Right) horizontal black lines indicate means. Student's t test, two-sided.

(Fig. 1g, h). Notably, mutp53 stabilization is very prominent at the invasive front (Fig. 1i). In sum, while LOH has an incremental effect on tumor proliferation, p53LOH is a dramatic gate-opener unleashing GOF by mediating mutp53 stabilization and enabling invasion.

**The WTp53 allele in heterozygous colorectal tumors retains partial activity and suppresses the HSF1 transcriptional program.** Tumor-specific mutp53 stabilization after p53LOH is dramatic, although its mechanism is incompletely understood. While loss of Mdm2 induction might play some role (Supplementary Fig. 2a, compare *Mdm2* mRNA in p53$^{Q/+}$ versus p53$^{Q/-}$ tumors) in agreement with other studies[25,50], an additional mechanism likely exists to ensure such massive stabilization after p53LOH.

A major pathway for tumor-specific mutp53 stabilization is the intrinsic tumor stress-induced HSF1-governed chaperone system[23,40,42,51,52]. In cancer cells the constitutively (phospho-) activated master transcription factor HSF1 orchestrates the major proteotoxic defense. Thus, we asked whether in heterozygous tumors the remaining WTp53 suppresses global HSF1 activity or distinct HSF1 chaperone targets. This hypothesis assumes that despite the presence of a GOF allele (Q in this case) that might exert a partial DNE, importantly, the remaining WTp53 allele retains partial transcriptional activity. Thus, we treated tumor-bearing p53$^{Q/+}$ mice with Nutlin, a highly specific non-genotoxic p53 activator inhibiting its E3 ligase MDM2, to mimic the general activation state of WTp53 in tumors constitutively stressed by aberrant growth, metabolic stress, hypoxia, and genomic instability (Fig. 2a).

Indeed, RNA-seq analysis revealed that Nutlin induced a broad hallmark WTp53 target response in p53$^{Q/+}$ tumors, indicated by GSEA (Fig. 2b) and Enrichr gene set enrichment analyses (GSEA; https://maayanlab.cloud/Enrichr; Supplementary Fig. 2b). Comparing Nutlin-treated p53$^{-/+}$ with p53$^{Q/+}$ tumors reveals that in addition to a partial DNE considerable residual WTp53 activity remains in p53$^{Q/+}$ tumors (Fig. 2b). This was confirmed on individual genes (*Cdkn1a*, *Gadd45*, and *Sfn*) by quantitative real-time PCR (qRT-PCR; Supplementary Fig. 2c). We conclude that the GOF mutp53$^{R248Q}$ allele fails to exert a complete DNE over the remaining WTp53 allele, as was predicted mainly by in vitro studies[53–56]. Importantly, the residual WTp53 activity was still sufficient to broadly suppress canonical HSF1 target genes as indicated by RNA-seq analysis (Supplementary Fig. 2d), which was confirmed by individual gene analysis (Supplementary Fig. 2e) in p53$^{Q/+}$ tumors. As expected from the double allelic dose, p53$^{+/+}$ tumors showed a stronger HSF1 target gene suppression after Nutlin (Supplementary Fig. 2e). Moreover, a comparison between expression measurements by RNA-seq versus qRT-PCR of individual genes showed consistency in all cases (Supplementary Fig. 2e, f). HSF1 targets were significantly downregulated in p53$^{+/+}$ tumors after Nutlin treatment by both methods.

We next tested whether simple loss of the WTp53 allele without Nutlin is able to activate HSF1. While p53$^{-/-}$ versus p53$^{+/+}$ AOM/DSS mice have accelerated tumor growth (larger tumor numbers and sizes, Supplementary Fig. 2g–j) due to reduced cell cycle inhibitory/pro-apoptotic p53 target gene expression (Supplementary Fig. 2k, l), only some HSF1 target genes increased (Supplementary Fig. 2m). Conversely, stress-activated WTp53, mimicked by Nutlin, suppresses HSF1 activity to prevent chaperone-mediated mutp53 stabilization (Supplementary Fig. 2d, e).

To further strengthen WTp53-mediated HSF1 suppression in heterozygous tumors, we generated CRC tumor-derived organoids (Fig. 2c). Importantly, p53$^{Q/fl}$ organoid cultures maintain their heterozygous p53 status over at least 7 passages (Supplementary Fig. 2n, o). Thus, we treated p53$^{Q/fl}$;vilCreER$^{T2}$ organoids first with 4OH-TAM (4OHT) to induce p53LOH, followed by Nutlin (Fig. 2d, e). Heterozygous EtOH controls showed strong induction of p53 target genes after Nutlin, whereas the p53 response was significantly dampened in p53LOH organoids (Fig. 2d). Cre recombinase-mediated allele deletion is never 100% efficient. Indeed, WT allele-specific qRT-PCR post 4OHT still showed ~15% retained WTp53 allele (Supplementary Fig. 2p), explaining the mild but detectable residual Nutlin response on p53 targets in 4OHT-treated organoids (Fig. 2d). Importantly, however, HSF1 target genes still became derepressed in p53LOH organoids (p53$^{Q/Δ}$ group) versus the non-LOH p53$^{Q/fl}$ group after Nutlin (Fig. 2e), again confirming that p53LOH enables HSF1 activity. Likewise, other p53 activators like Doxorubicin and 5-FU (Supplementary Fig. 2q) also induced HSF1 suppression in heterozygous p53$^{Q/fl}$ organoids (Fig. 2f). In agreement with an upregulated chaperone system after p53LOH, nuclear mutp53$^{R248Q}$ became strongly stabilized after 4OHT (Fig. 2g). Interestingly, in contrast to p53$^{Q/fl}$ heterozygous organoids (Fig. 2d), p53$^{Q/+}$ tumors did not induce *Mdm2*, suggesting that in vivo Mdm2 requires higher levels of Nutlin and/or more time than other p53 targets (Supplementary Fig. 2c). At any rate, this indicates that p53-regulated Mdm2 levels cannot account for the missing mutp53 stabilization in heterozygous tumors.

In sum, in a stressed tumor milieu the WTp53 allele in heterozygous mutp53/+ tumors retains partial activity and suppresses the HSF1 transcriptional program, apparently creating a driving force for p53LOH. p53LOH eliminates the repressive WTp53–HSF1 axis and enables activation of the broad co-oncogenic HSF1 functions. This causes mutp53 protein stabilization that in turn enables tumor growth, but foremost tumor invasion.

**Activated WTp53 represses HSF1 activity in human colorectal cancer cells.** Since mutp53 stabilization specifically arises in the malignant epithelial compartment, we analyzed the mechanism of p53-mediated HSF1 suppression in human CRC cell lines harboring WTp53. We resorted to homozygous WTp53 lines because stable heterozygous human CRC lines are not available. Importantly, measuring the global HSF1-mediated HSR response by heat-shock response element (HSE) luciferase assays confirmed HSF1 suppression upon WTp53 activation by Nutlin (Fig. 3a). Moreover, Nutlin-induced HSF1 suppression was rescued by shp53-mediated depletion, confirming that the Nutlin-induced effect is p53-specific (Fig. 3b).

HSF1 not only orchestrates the cellular chaperone system. In cancer cells, HSF1 also broadly upregulates a large palette of tumor-promoting genes involved in cell cycle, DNA repair, metabolism, adhesion, and protein translation[41,46]. Thus, we analyzed randomly selected HSF1 targets. Notably, upon p53 activation by Nutlin, we observed broad repression of classic HSF1 targets, including *HSP90AA1*, *HSPA1A*, *HSPH1*, *HSPB1*, *DNAJA1*, and *DNAJB1*, validating the mouse model (Fig. 3c). Moreover, Nutlin also regulated the tumor-promoting HSF1 targets *CDC6*, *ITGB3BP*, *RBBP5*, *BST2*, and *FBLN1* (Fig. 3c). Importantly, p53 depletion by siRNAs rescued their repression, confirming that p53 specifically regulates HSF1 activity (Fig. 3d and Supplementary Fig. 3a). The critical phosphorylation site for HSF1 activation is residue Ser326, which serves as functional hallmark of the tumor-promoting HSR response[44,57]. Concomitantly to HSF1 target gene repression, p53 activation was correlated with a profound reduction of pSer326-HSF1 levels in HCT116, RKO (Fig. 3e), and LS513 and LS174T cells (Supplementary Fig. 3b). Again, p53 depletion by siRNAs (Supplementary Fig. 3c) or p53 deletion in isogenic HCT116 cells

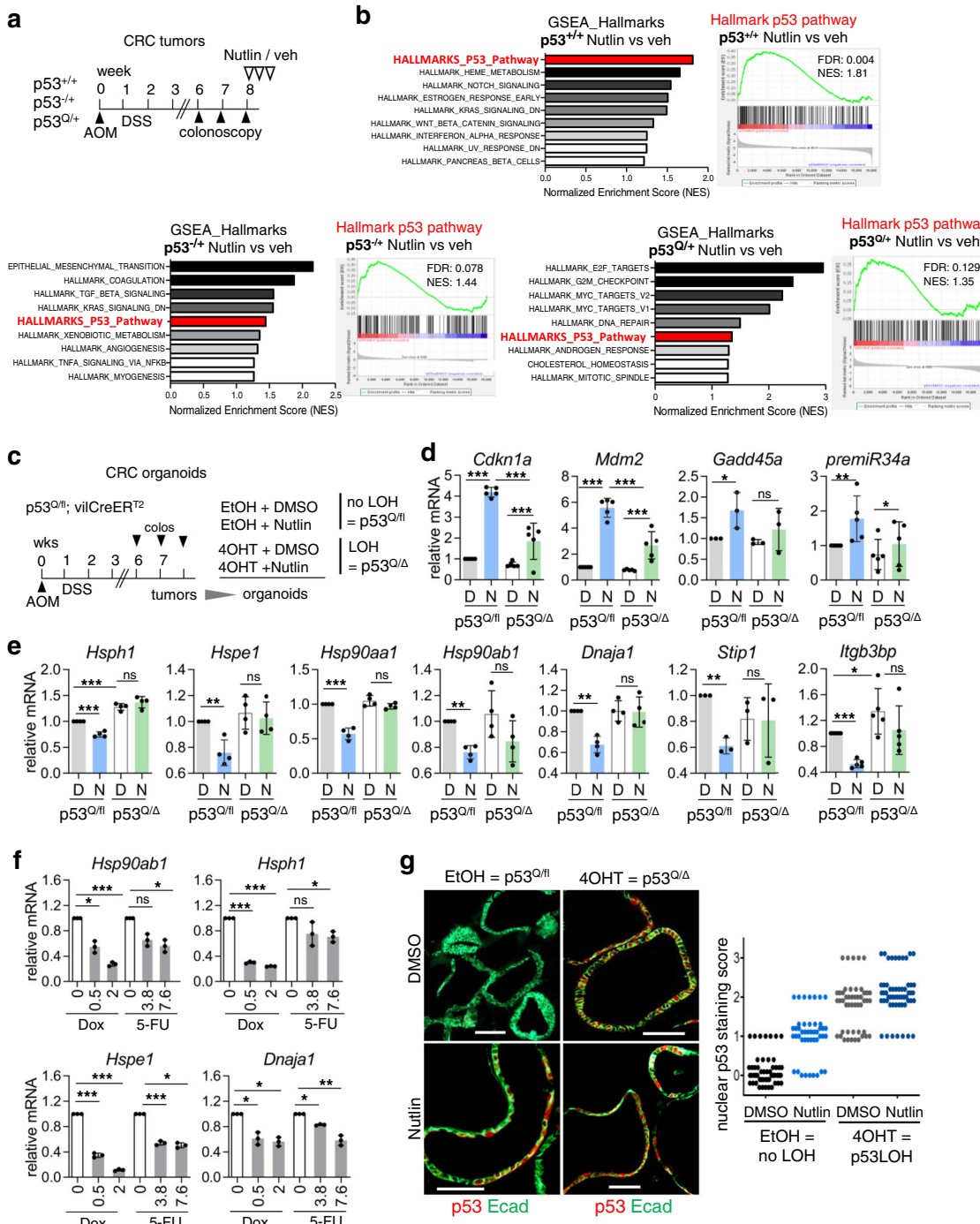

**Fig. 2 The WTp53 allele in heterozygous colorectal tumors retains partial activity and represses HSF1 target gene expression in vivo. a** Scheme of in vivo Nutlin treatment. After AOM/DSS induction tumor burden was monitored by colonoscopy. Mice with at least two to three S2 tumors plus at least one S3 tumor were treated with 150 mg/kg Nutlin/vehicle orally for 3 days. Tumors were analyzed 8 h after last treatment. **b** RNA-seq analysis. (Left) hallmark gene sets from GSEA analyses (sorted by NES) that are associated with WTp53 activity in ±Nutlin-treated p53+/+, p53−/+, and p53Q/+ tumors. (Right) GSEA enrichment plots for the corresponding hallmark p53 pathway. **c** p53LOH induction scheme of tumor-derived organoids. Heterozygous p53Q/fl;vilCreERT2 mice were treated with AOM/DSS. At 6–8 weeks post AOM colonoscopy-visualized tumors were resected, processed for organoid cultures and p53LOH induced by adding 4OHT for 24 h (p53Q/Δ) or EtOH (no LOH, p53Q/fl). Two days later organoids were treated with 10 µM Nutlin or DMSO for 24 h and analyzed. **d, e** mRNA levels of WTp53 target genes (**d**) and HSF1 target genes (**e**) from organoids in **c**. qRT-PCR co-normalized to *Hprt1* and *Rplp0* mRNA. **f** mRNA levels of HSF1 target genes from heterozygous (no LOH) p53Q/fl;vilCreERT2 organoids treated with Doxorubicin (µM) or 5-Fluorouracil (µM) for 24 h. qRT-PCR normalized to *Hprt1* mRNA. Mean ± SD from two independent experiments, one included a technical replicate from the same organoid culture (total *n* = 3). **g** (Left) representative immunofluorescence of the indicated organoid groups. p53 (red); E-cadherin (green) is used as epithelial counterstain for organoid visualization. Scale bars, 100 µm. (Right) Quantification of nuclear p53 staining. Each dot indicates one organoid of the indicated groups. **d, e** Mean ± SD from ≥3 independent experiments of ≥3 cultures. **d–f** Student's *t* test, two-sided, *p ≤ 0.05, **p ≤ 0.01, ***p ≤ 0.001; ns not significant.

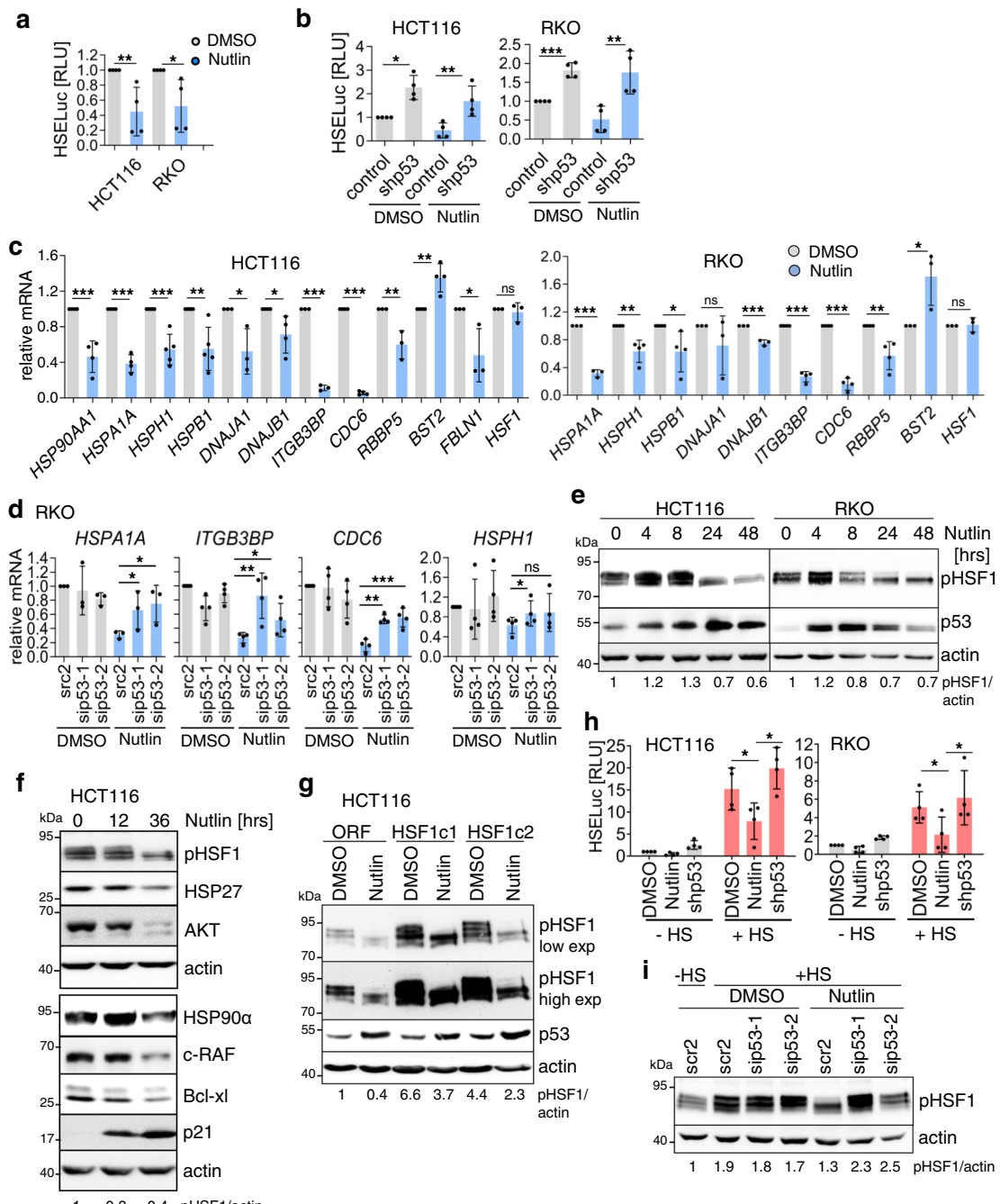

**Fig. 3 Activated WTp53 represses HSF1 activity in human colorectal cancer cells. a** Luciferase reporter assay for heat-shock response elements (HSE) in HCT116 and RKO cells treated with 10 μM Nutlin/DMSO for 24 h. **b** HSE luciferase assay as in **a** after depletion of WTp53 by shRNA. Control, scramble shRNA. Forty-eight hours post transfection, cells were treated with Nutlin/DMSO (10 μM) for 24 h. **c** Chaperone-dependent and -independent HSF1 target gene expression in indicated cells treated with 10 μM Nutlin/DMSO for 24 h. qRT-PCR normalized to *RPLP0* or *HPRT1* mRNA. **d** HSF1 target gene expression in RKO cells upon WTp53 depletion. Forty-eight hours post transfection with sip53RNAs or scrambled control siRNA, cells were treated with Nutlin/DMSO (10 μM) for 24 h. qRT-PCR as in **c**. **e** Activated WTp53 is correlated with suppression of pSer326-HSF1, the key marker of HSF1 activity. Indicated cells were treated ±10 μM Nutlin. **f** Repression of HSF1 targets and destabilization of Hsp90 clients after p53 activation. Cells were treated with 10 μM Nutlin/DMSO. **g** Stably HSF1-overexpressing HCT116 subclones (HSF1c1 and HSF1c2) or empty vector clone (ORF) were treated with Nutlin/DMSO for 24 h. pSer326-HSF1 shown with short and long exposures. **h** Nutlin represses HSF1 activity in heat-shocked cells, rescued by p53 knockdown. HSE luciferase assay as in **b**. Cells were treated with 10 μM Nutlin/DMSO for 24 h. During the final 2 h, cells were heat-shocked for 1 h at 42 °C followed by 1 h recovery. **i** The heat-shock response is attenuated by Nutlin, rescued by p53 knockdown. Cells were transfected with sip53RNAs or scrambled RNA (scr2) and treated as in **h**. Immunoblot of pSer326-HSF1. **e**–**g**, **i** Immunoblots. Actin, loading control. pHSF1/actin, pHSF1 densitometry normalized to loading control. Source data are provided as Source data file. **a**–**d**, **h** Student's *t* test, two-sided. *$p \leq 0.05$, **$p \leq 0.01$, ***$p \leq 0.001$; ns not significant. **a**, **b**, **h** Mean ± SD of ≥3 independent experiments, each measured in triplicates. **c**, **d** Mean ± SD of ≥2 independent experiments, at least one with a technical replicate. Relative values given in [ratio ($2^{-ddCT}$)].

(Supplementary Fig. 3d) abolished pSer326-HSF1 dephosphorylation. Conversely, mutp53-harboring CRC cells failed to repress HSF1 phosphorylation after Nutlin (Supplementary Fig. 3e). Of note, total HSF1 protein remained unchanged, excluding that HSF1 dephosphorylation/inactivation is simply due to reduced total HSF1 levels (Supplementary Fig. 3c). Consequently, HSF1 inactivation reduced heat-shock (HS) component expression, such as Hsp90α and Hsp27 (Fig. 3f). Moreover, HSP90 clients including AKT, c-Raf, and Bcl-xl were also destabilized, confirming the inactivation of the HSF1-HSP90 anti-proteotoxic defense response upon p53 activation (Fig. 3f).

To further strengthen the evidence for this repressive WTp53–HSF1 axis, we generated stable HSF1-overexpressing HCT116 clones, functionally confirmed by increased levels of pSer326-HSF1 (Fig. 3g) and higher expression of HSF1 target genes (Supplementary Fig. 3f). Again, Nutlin treatment correlated with strong dephosphorylation of pSer326-HSF1 (Fig. 3g) and suppressed the increased HSF1 target gene response in these clones (Supplementary Fig. 3f). To finally demonstrate that HSF1 is directly controlled by WTp53, we used HS, the strongest known HSF1 activator, to massively increase endogenous HSF1 activity in human CRC cells. Again, Nutlin strongly repressed HSF1 activity after HS, rescued by p53 knockdown, demonstrating that WTp53 potently counter-regulates even the strongest HSF1 activation (Fig. 3h, i). In sum, the HSF1-mediated stress response is strongly attenuated by activation of WTp53.

**p53 blocks HSF1 activity via p21-mediated cell cycle inhibition in human CRC cells.** To gain further insight into Nutlin-induced HSF1 suppression, we analyzed CDKN1A/p21, a key p53 target gene and potent CDK inhibitor that mediates cell cycle arrest (Supplementary Fig. 4a). Indeed, p21 depletion by siRNAs abolished pSer326-HSF1 dephosphorylation (Fig. 4a) and partly rescued Nutlin-induced HSF1 target gene repression (Fig. 4b and Supplementary Fig. 4a, b), indicating a p53–p21-mediated HSF1 pathway of suppression.

Next we asked whether the p53-p21-mediated HSF1 suppression is linked to and regulated by the cell cycle. p21 binds and inhibits cyclin-dependent kinases (CDKs), thereby preventing phosphorylation of the retinoblastoma protein (RB). Hypo-phosphorylated RB binds to and inhibits E2F transcription factors, preventing S-phase entry[58]. Thus, we tested whether cell cycle inhibitors like CDK4/6 inhibitor Palbociclib phenocopy the p53–p21-mediated HSF1 inactivation. Indeed, both Nutlin- and Palbociclib-treated cells exhibited markedly decreased levels of pSer326-HSF1 in WTp53 cells (Fig. 4c). Moreover, HSF1 targets were suppressed by Palbociclib, mimicking the Nutlin-induced HSF1 response (Fig. 4d). Likewise, combined silencing of CDK4 and CDK6 reduced pHSF1 (and pRB) levels, thereby phenocopying CDK4/6 inhibition by Palbociclib, albeit with lower efficiency (Supplementary Fig. 4c). Not surprisingly, the small-molecule inhibitor Palbociclib, widely used in clinically oncology, has a stronger effect in inhibiting the CDK4/6 complex than siRNAs (Supplementary Fig. 4d).

In further support, Nutlin-derivatives RG7112 and RG7388 (Idasanutlin) also reduced pSer326-HSF1 (Fig. 4e, f). Likewise, in HSF1-overexpressing HCT116 clones cell cycle inhibition by Palbociclib (like Nutlin) repressed pSer326-HSF1 levels (Fig. 4g) and HSF1 target gene expression (Supplementary Fig. 4e).

To pinpoint the specific CDKs involved in activating HSF1, we used RO3306 (inhibits CDK1 and CDK2 at lower concentrations, but CDK4 at higher concentrations) and Roscovitine (inhibits CDK1, CDK2, CDK5, and CDK7, but poorly CDK4/CDK6). Of note, only RO3306 at higher concentrations (10 μM) blocked pSer326-HSF1 like Nutlin and Palbociclib did (Fig. 4h), indicating a specific role for CDK4/6 in HSF1 activation. Furthermore,

in vivo in heterozygous tumor-bearing p53$^{Q/+}$ mice (see Fig. 2a), Nutlin induced a moderate RB and HSF1 dephosphorylation (Fig. 4i). Overall, these data demonstrate that cell cycle inhibition via p53-induced p21-CDK4/6 signaling suppresses HSF1 activity.

**MLK3 links cell cycle to the MAPK stress pathway to activate the HSF1 response. WTp53 activation represses MLK3.** The HSF1 stress response is markedly attenuated by CDK4/6 inhibition (Fig. 4). Thus, we tested whether E2F target genes like CDK1, CDK2, CDC25C, PLK4, and MLK3 control the HSF1-mediated HSR. Indeed, these E2F targets are all strongly repressed by Nutlin-activated p53, an effect largely rescued by concomitant p53 depletion (Fig. 5a). Specifically, MLK3 depletion mimicked the Nutlin response and reduced both pSer326-HSF1 (Fig. 5b) and HSF1 target gene expression (Fig. 5c, d). Notably, MLK3 directly signals to the MEK/ERK stress pathway[59–61] and MEK/ERK activates HSF1 by phosphorylation[42,62,63]. In contrast, depletion of CDK1 or CDK2 failed to reduce pSer326-HSF1 (Supplementary Fig. 5a, b). Moreover, PLK4 protein was not diminished after silencing, albeit PLK4 mRNA was strongly reduced, pointing to a stable PLK4 protein, but excluding PLK4 as HSF1-activating kinase (Supplementary Fig. 5c, d). In contrast, MLK3 depletion reduced both pSer326-HSF1 and MEK phosphorylation (Fig. 5b), revealing a MLK3–MEK–HSF1 signaling axis.

Importantly, MLK3 mRNA and MLK3 protein levels were reduced after p53 activation (Nutlin and RG7112, Fig. 5a, b, e, f and Supplementary Fig. 5e) and cell cycle inhibition (CDK4i and RO3306, Fig. 5e, g), parallel to MEK inactivation (pMEK1, Fig. 5e–g). Direct MEK inhibition by U0126 also blocked HSF1 phosphorylation (Supplementary Fig. 5f). Furthermore, in p53$^{Q/fl}$;vilCreER$^{T2}$ tumor-derived organoids, Nutlin-downregulated MLK3 expression was rescued by 4OHT-induced p53LOH (p53$^{Q/Δ}$;vilCreER$^{T2}$; Fig. 5h). Likewise, other p53 activators (Doxorubicin and 5-FU) also repressed MLK3 expression in heterozygous p53$^{Q/fl}$ organoids (Fig. 5i). Finally, MEK inhibition reduced HSF1 target gene expression in p53$^{Q/fl}$ organoids (Fig. 5j).

Taken together, we confirmed the MAPK stress pathway as major HSF1 activator in human WTp53-harboring CRC cells. We identified MLK3 as upstream link between cell cycle and the MAPK pathway. WTp53 activation represses MLK3, which in turn inactivates the MAPK stress pathway and therefore the HSF1 response.

**In human colorectal cancers p53LOH combined with missense mutp53 tends to shorten patient survival and upregulate HSF1 activity.** Tumors strongly depend on constitutively upregulated chaperones to manage pervasive proteotoxic stress. However, functional WTp53 prevents adaptive upregulation of the HSF1 chaperone system (Figs. 2–5). Thus, to survive and progress, tumors are under strong selection pressure to undergo p53LOH[4,16] and lose WTp53-mediated HSF1 repression. This scenario was confirmed in human CRC. p53LOH occurred in ~75% of patients harboring either all p53 variants (MS missense, FS frameshift, NS nonsense) or MS-only variants (MS; Fig. 6a, COADREAD, the Cancer Genome Atlas (TCGA) data). Importantly, p53LOH combined with MS mutp53 (MS + LOH) showed a trend to shorter survival (median 57.2 month versus 83.2 months in WTp53 patients; Fig. 6b, note that TCGA lacks sufficient numbers of heterozygous patients (MS diploid), precluding statistical survival analysis). Remarkably, HSF1 target genes are concomitantly deregulated in p53LOH CRCs harboring all p53 mutations (MS/NS/FS + LOH, Supplementary Fig. 6a, c) or MS-only variants (MS + LOH, Fig. 6c and Supplementary

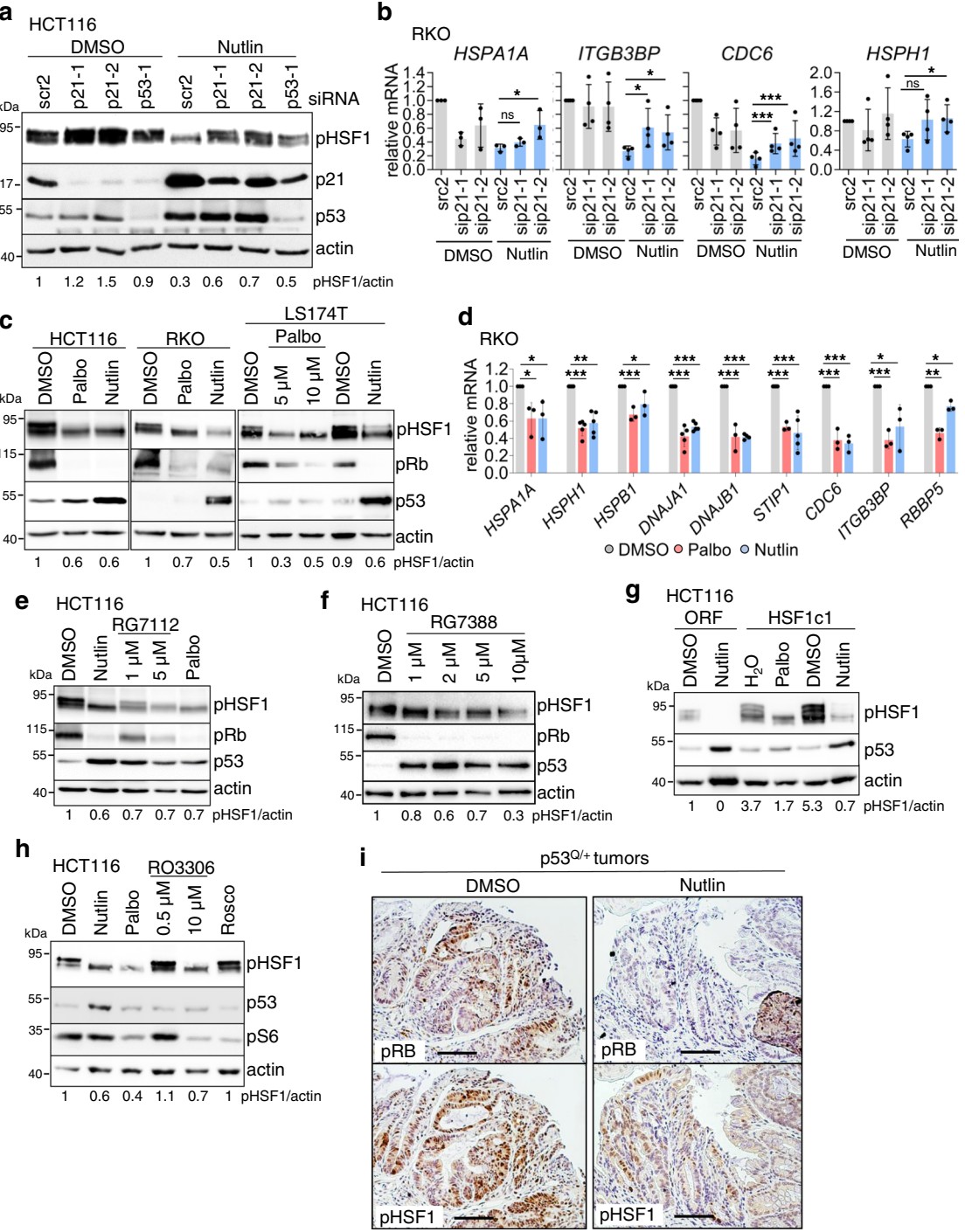

**Fig. 4 p53 suppresses HSF1 activity via cyclin-dependent kinase inhibitor *CDKN1A*/p21 in human CRC cells. a** p21 silencing attenuates Nutlin-induced HSF1 suppression. Forty-eight hours after HCT116 cells were transfected with siRNAs against p21 and p53 or scrambled control siRNA, cells were treated with 10 μM Nutlin /DMSO for 24 h. **b** Rescue of p53-induced HSF1 target gene suppression by p21 depletion in RKO cells treated as in **a**. **c** WTp53-harboring CRC cell lines were treated with DMSO, 10 μM Palbociclib (Palbo), or 10 μM Nutlin for 24 h. Cell cycle inhibition was confirmed by Rb de(hypo) phosphorylation. pRb phospho-Rb. **d** HSF1 target gene repression after direct cell cycle inhibition. RKO cells were treated with 10 μM Palbociclib (CDK4/6 inhibitor), 10 μM Nutlin, or DMSO for 24 h. **e, f** Cell cycle inhibition by p53 inactivates HSF1 activity. HCT116 cells were treated with DMSO, 10 μM Nutlin, 10 μM Palbociclib, RG7112 (**e**) or RG7388/Idasanutlin (**f**) as indicated for 24 h. **g** Cell cycle inhibition prevents pSer326-HSF1 activation in stably HSF1-overexpressing HCT116 cells. HSF1c1 or empty vector clone (ORF) were treated with DMSO, H$_2$O, 10 μM Nutlin, or 10 μM Palbociclib for 24 h. **h** CDK4/6 inhibition causes HSF1 suppression. HCT116 cells were treated with DMSO, 10 μM Nutlin, 10 μM Palbociclib, 0.5 and 10 μM RO3306, and 20 μM Roscovitine (Rosco) for 24 h. **i** Representative immunohistochemistry of serial sections for pRB and pHSF1 from p53$^{Q/+}$ tumors treated with DMSO or Nutlin as in Fig. 2a ($n = 6$ tumors each). Scale bars, 100 μm. **b, d** qRT-PCR, normalized to *RPLP0* mRNA. Relative values given in [ratio (2$^{-ddCT}$)]. Mean ± SD of ≥2 independent experiments, at least one with a technical replicate. Student's *t* test, two-sided. *$p \le 0.05$, **$p \le 0.01$, ***$p \le 0.001$; ns not significant. **a, c, e–h** Immunoblot analyses. Actin, loading control. pHSF1/actin, pHSF1 densitometry normalized to loading control. Source data are provided as Source data file.

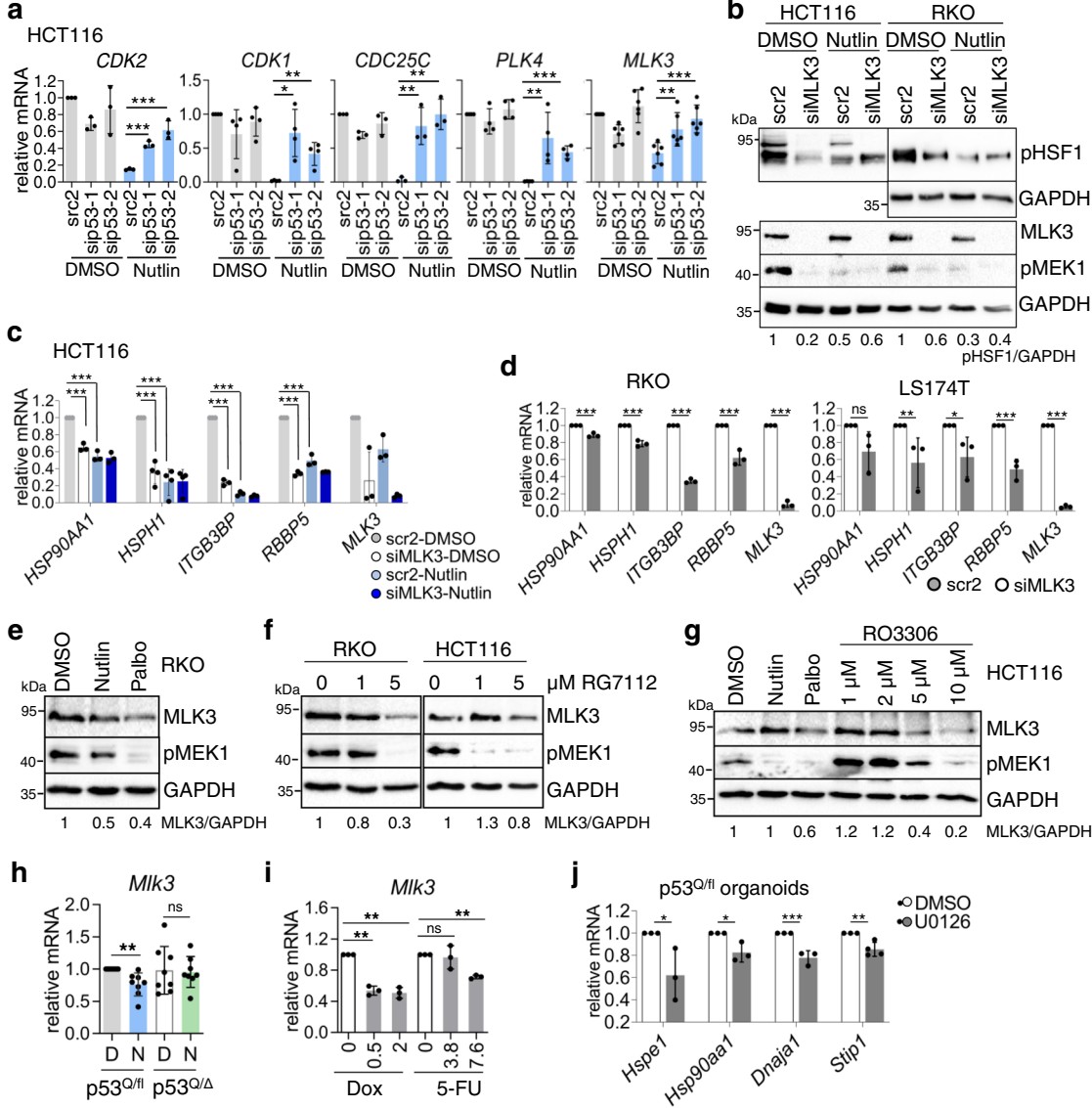

**Fig. 5 MLK3 links cell cycle to the MAPK stress pathway to activate the HSF1 response.** WTp53 activation represses MLK3. **a** The expression of cell cycle progression genes is inhibited by p53 activation. HCT116 cells were transfected with siRNAs for p53 or scrambled control siRNA (scr2) for 48 h, followed by DMSO or 10 μM Nutlin treatment for 24 h. qRT-PCRs for the indicated mRNAs normalized to *RPLP0* mRNA. MLK3 mixed lineage kinase 3, PLK4 polo-like kinase 4. **b** MLK3 silencing suppresses pSer326-HSF1, mimicking the effect of p53 activation. The indicated cells were transfected with an siRNAs pool against MLK3 or scrambled control siRNA (scr2). Forty-eight hours post transfection, cells were treated with DMSO or 10 μM Nutlin for 24 h. Immunoblot analysis. GAPDH, loading control. **c** MLK3 silencing abrogates HSF1 target gene expression, mimicking the effect of p53 activation. HCT116 cells were transfected and treated as in **b**. qRT-PCRs for the indicated mRNAs normalized to *RPLP0* mRNA. **d** HSF1 target gene expression is attenuated after MLK3 depletion. Indicated cells were transfected with an siRNAs pool against MLK3 or scrambled control siRNA (scr2). Seventy-two hours post transfection, qRT-PCRs for the indicated mRNAs were performed. Normalized to *RPLP0* mRNA. **e**–**g** Cell cycle inhibition reduces MLK3 expression and causes MEK1 inactivation. The indicated cells were treated for 24 h with DMSO, 10 μM Nutlin, or 10 μM Palbociclib (CDK4i) (**e**, **g**), RG7112 (**f**), and RO3306 (**g**) at the indicated concentrations. Immunoblot analysis. GAPDH, loading control. MLK3/GAPDH, MLK3 densitometry normalized to loading control. **h** *Mlk3* mRNA levels of isolated p53^Q/fl;vilCreER^T2 tumor organoids treated as in Fig. 2c. qRT-PCR normalized to *Hprt1* mRNA. Mean ± SD of eight independent experiments from ≥3 organoid cultures/3 different mice. **i** *Mlk3* mRNA levels of heterozygous p53^Q/fl;vilCreER^T2 organoids treated as indicated for 24 h. Dox Doxorubicin in μM, 5-FU 5-Fluorouracil in μM. qRT-PCR normalized to *Hprt* mRNA. Mean ± SD from two independent experiments, one included a technical replicate from the same organoid culture (total n = 3). **j** p53^Q/fl;vilCreER^T2 organoids treated with MEKi U0126 for 6 h. qRT-PCR of the indicated HSF1 target genes normalized to *Hprt* mRNA. Mean ± SD of ≥2 different organoid cultures with two technical replicates each. Student's (**a**, **c**, **d**) mean ± SD of two independent experiments, at least one with a technical replicate. Relative values as ratio (2^−ddCT). **a**, **c**, **d**, **h**, **I**, **j** Students *t* test, two-sided. *$p \leq 0.05$, **$p \leq 0.01$, ***$p \leq 0.001$; ns not significant. **b**, **e**–**g** Source data are provided as Source data file.

Fig. 6b). Notably, deregulation of HSF1 targets was identified for a stringent direct target gene set regulated by HSF1 binding (ChIP-seq/RNA-seq) after HSF1 knockdown in breast cancer cells[46] (Fig. 6c and Supplementary Fig. 6a), as well as for a second independent target gene set generated from HS treated cancer

cells[64] (Supplementary Fig. 6b, c). Moreover, high *HSF1* mRNA expression strongly correlates with p53LOH CRCs (Fig. 6d, e), regardless of whether all p53 mutants were considered (Fig. 6d top) or specifically MS + LOH cases (Fig. 6d bottom). Mutant p53 patients with p53LOH and high *HSF1* expression showed

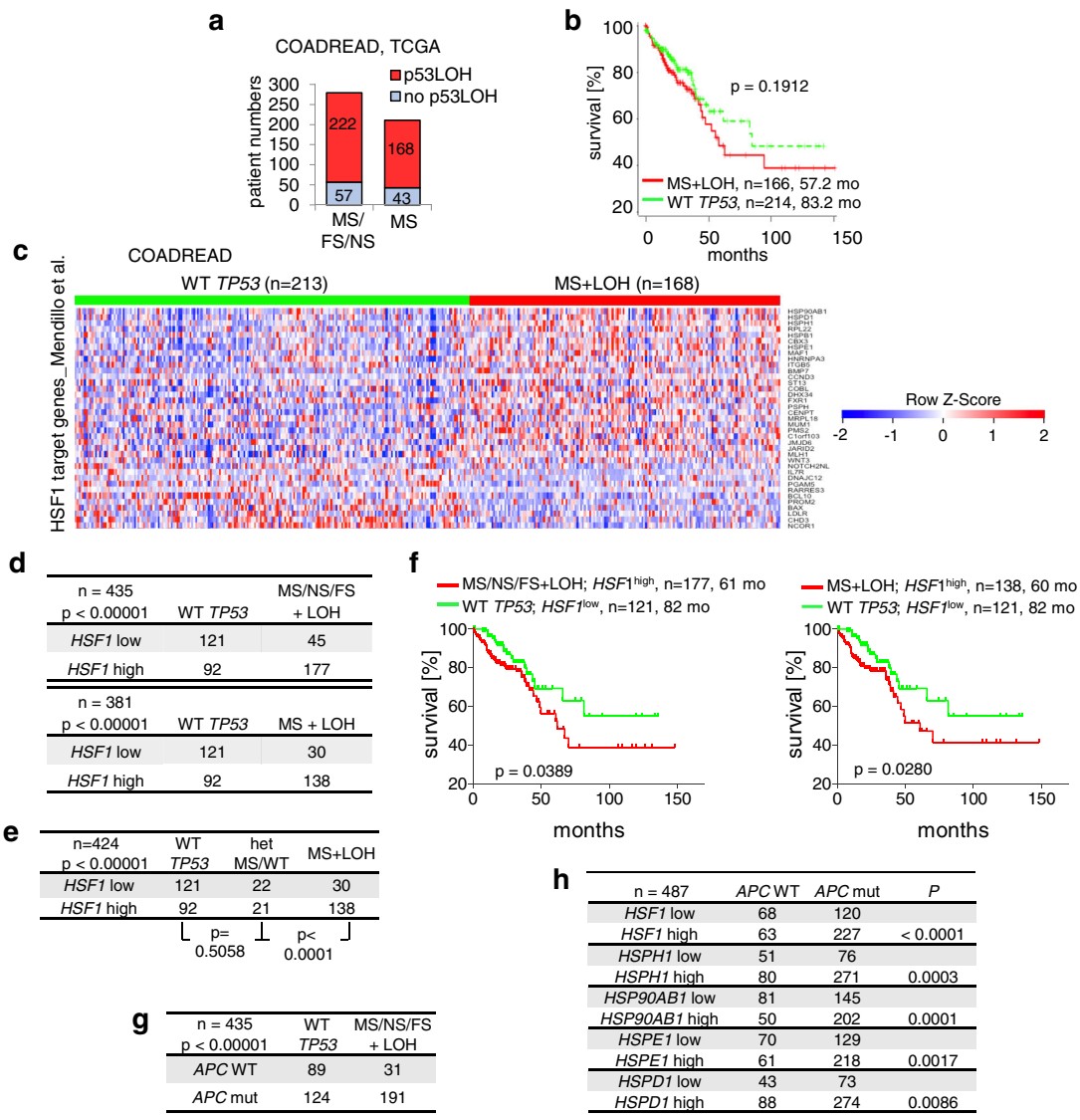

**Fig. 6 In human colorectal cancers p53LOH combined with missense mutp53 shortens patient survival and upregulates HSF1 activity. a–h** Data from colorectal adenocarcinoma patients. COADREAD, TCGA dataset. **a** Tumor samples stratified for p53LOH versus no-p53LOH and grouped for either all *TP53* mutations (MS missense, FS frameshift, NS nonsense) versus missense mutations-only (MS). p53LOH samples (shallow deletions) were determined by TP53 copy number alterations. Numbers in columns indicate patient numbers. **b** Survival curve of CRC patients harboring homozygous WT *TP53* versus patients harboring missense (MS) p53 mutations plus p53LOH (shallow deletions). Number of patients and mean survival in months as indicated. Kaplan–Meier statistic, two-sided, log-rank test, $p = 0.19$. Note that TCGA data contains insufficient numbers of heterozygous patients (mutp53/+) with clinical data, precluding statistical survival analysis. **c** Heatmap of HSF1 target genes from colorectal carcinomas. Patients with homozygous $TP53^{+/+}$ (WT *TP53*) versus patients with *TP53* missense (MS) mutations plus p53LOH (MS + LOH). All genes with *t* test *p* values <0.001 were plotted as heatmap rank-ordered by their relative expression change from top to bottom. Colors represent *Z*-scores for each gene from blue (downregulated) to red (upregulated) expression. A homemade HSF1 target gene panel from Mendillo et al. was used[46]. **d** Correlation between *HSF1* mRNA expression and p53 mutational status in colorectal carcinomas. Patients with homozygous $TP53^{+/+}$ (WT *TP53*) versus patients with either MS/FS/NS + LOH (top, $n = 435$ patients) or patients with MS + LOH (bottom, $n = 381$ patients) with regard to their *HSF1* expression. Fisher's exact test, two-sided. **e** Loss of the remaining WTp53 allele correlates with high *HSF1* mRNA expression in colorectal carcinomas. Patients with WT *TP53* versus MS/WT (HET) versus MS + LOH. Chi-square statistic, $n = 424$ patients. Fisher's exact test, two-sided, between WT versus HET and between HET versus MS + LOH. **f** Survival of patients with MS/NS/FS + LOH (left) or MS + LOH (right), and $HSF1^{high}$ versus patients with WT *TP53* and $HSF1^{low}$. Number of patients, mean survival in months and Kaplan–Meier statistic log-rank test, two-sided, as indicated. **g, h** Correlation between *APC* mutations and p53 mutational status (**g**) or mRNA expression of *HSF1* and HSF1 target genes (**h**). Analysis as in **d**. Patient numbers as indicated. Fisher's exact test, two-sided.

reduced survival compared to WTp53 patients with low *HSF1* (Fig. 6f). Interestingly, *APC* mutations, known as oncogenic initiators in CRC, also correlate with p53LOH (Fig. 6g) and HSF1 target gene expression (Fig. 6h). The higher N and M metastatic stages of CRC patients with p53LOH confirm the scenario that WTp53 is a gatekeeper for invasion (Supplementary Fig. 6d, e).

Moreover, p53LOH breast cancers (BRCA TCGA) also exhibited upregulated HSF1 targets (Supplementary Fig. 6f–i). Furthermore, breast cancer cells repressed pSer326-HSF1 when harboring WTp53, but not mutp53 (Supplementary Fig. 6j). Together, these data support that in human colorectal and breast cancers p53LOH overrides HSF1 repression by WTp53 and

enables pleiotropic tumor-promoting HSF1 functions contributing to poorer prognosis.

**In murine CRC organoids p53LOH de-represses HSF1 activity and triggers mutp53-driven invasion.** Three days after p53LOH induction mesenchymal markers Vimentin and SnaI were not yet increased (Fig. 7a). To support our hypothesis that the establishment of GOF after p53LOH takes time, we analyzed the p53$^{Q/fl}$;vilCreER$^{T2}$ tumor-derived organoids over a longer time span of 20 days (Fig. 7b and Supplementary Fig. 7a). To optimize recombination efficiency, we used three weekly pulses of 4OHT. p53LOH was confirmed by ~80% loss of the WTp53 allele (Supplementary Fig. 7b), and decreased p53 target gene expression measured by RNA-seq analysis (Supplementary Fig. 7c) and by individual p53 target qRT-PCRs (Supplementary Fig. 7d). Together, this confirmed that the retained WTp53 allele in p53$^{Q/fl}$ had residual transcriptional activity. Indeed, after long-term p53LOH EMT (epithelial to mesenchymal transition) markers were increased (Fig. 7c) and invasive branching morphology[7] appeared in p53$^{Q/\Delta}$ organoids (Fig. 7d). Moreover, HSF1 targets —which were all repressed after short 24 h Nutlin treatment (Fig. 2e), exhibited upregulation after long-term p53LOH (Fig. 7e, f). Moreover, upon p53LOH organoids increased invasiveness in transwell assays (Fig. 7g). Notably, GOF activities such as Stat3 target genes[5] were also upregulated (Supplementary Fig. 7e).

Moreover, to mimic the chronic stress present in tumor milieus that provides selection pressure for spontaneous p53LOH, we treated heterozygous p53$^{Q/fl}$ organoids for 21 days with Nutlin (Fig. 7h). After an initial cell death wave due to Nutlin-activated WTp53 (Supplementary Fig. 7f), many organoids—those that had undergone p53LOH (Fig. 7h, bottom) and reduced p53 target gene expression (Supplementary Fig. 7g)—were able to regrow and now showed invasive morphologic structures with branchings and protrusions (Supplementary Fig. 7f). Moreover, they exhibit upregulation of HSF1 targets (Fig. 7i), EMT markers (Fig. 7j), and Stat3 targets (Supplementary Fig. 7h). To further support that a stressed tumor milieu enforces p53LOH, we analyzed the KPC (Kras; p53$^{R172H/+}$) pancreatic cancer model exhibiting aggressive tumor growth and a high spontaneous p53LOH frequency of ~70%, indicated by conversion from undetectable to positive p53 staining[9,24]. Indeed, p53$^{high}$ cases (LOH) tended to have shorter survival (median = 132 days) than p53$^{low}$ cases (no LOH, median = 187 days; Fig. 7k). Moreover, p53$^{high}$ (p53LOH) colocalizes with Hsp70$^{high}$ and pHSF1$^{high}$ staining in tumor epithelial cells. This correlation further supports HSF1 axis-mediated mutp53 stabilization (Fig. 7l and Supplementary Fig. 7i).

In sum, these data further support that p53LOH-mediated HSF1 activation causes mutp53 stabilization and enables invasiveness.

## Discussion

Here, we use an autochthonous immune-competent mouse model that recapitulates human CRC[65] and identify one mechanism that drives the critical p53LOH event in p53 mutant heterozygous tumors. We show that the remaining WTp53 allele in p53$^{R248Q/+}$ tumors, despite a partial DNE by the Q allele, retains broad partial p53 activity. One of these activities represses the cytoprotective HSF1 chaperone axis. We identify a repressive WTp53–p21–MLK3–MAPK–HSF1 signaling cascade as the underlying mechanism that creates a driving force for losing the WTp53 allele. Hence, WTp53-mediated HSF1 suppression exerts selection pressure for p53LOH. Conversely, p53LOH, once it occurs, is a dramatic all-or-none gate-opener for tumor progression, removing remaining p53 tumor suppression and

derepressing the HSF1-HSP90 chaperone system. Notably, a fully activated HSF1 axis is a powerful co-oncogenic driver of tumorigenesis[44,48,66].

In case of the R248Q GOF allele, the retained WTp53, via its ability to repress the HSF1-regulated chaperone system, concomitantly prevents stabilization of mutp53 protein in heterozygous tumors, thereby blocking the oncogenic potency of the Q allele. Conversely, p53LOH unleashes its broad GOF functions by protein stabilization, in turn enabling tumor invasion[5,15,25,67].

Our findings, corroborated by murine CRC organoids, human cell lines and human tumors in which stabilized mutp53 critically enables tumor invasion[17,18], reveal the pivotal significance of the repressive WTp53–HSF1 axis. Thus, a single genetic event, p53LOH, kills three birds with one stone: (I) losing WTp53 suppressor activity including HSF1 repression, (II) upregulating tumor-promoting HSF1 activity, and (III) enabling mutp53 protein stabilization, thereby unleashing the GOF potential of specific alleles.

Selection pressure for p53LOH is a major force in the progression of most epithelial-derived carcinomas, including CRC. Ninety percent of human tumors are carcinomas and indeed, the vast majority (>91%) of human tumors with p53 mutations spanning 32 cancer types undergo p53LOH. This was definitively shown in a comprehensive study with five data platforms on 10,225 human TCGA cancers[4]. This fact strongly suggests if not proves that in the vast majority of heterozygous solid cancers the DNE of mutant p53 alleles is only partial and weak, leaving behind significant residual wtp53 activity, which provides selection pressure for LOH. It follows that if there were strong complete DNE, there would be no selection pressure for losing the WT allele and LOH would not be observed. Indeed, we observe incomplete DNE exerted by the Q allele as indicated by broad residual WTp53 activity in heterozygous Q/+ CRC tumors and organoids (Fig. 2). In support, CRISPR/Cas9 engineered p53$^{R248W/+}$ HCT116 cells, while showing partial DNE, still have sufficient residual wtp53 activity to stop cell cycle after Nutlin treatment[53]. Moreover, incomplete DNE was also observed in an Apc-driven mouse model of small intestinal cancer in Trp53$^{R270H/+}$ mice[7]. In general, many MS mutants are unable to exert complete DNE in carcinomas and sarcomas, including the hotspot p53$^{R248Q}$ allele[5,8,38,55,68].

Studies of normal tissues of mutp53 knock-in mice established that MDM2 degrades mutant and WTp53 equally well, keeping both mutant and WT levels below immunohistochemical detection[9,25,27]. Conversely, in response to stress both WT and mutp53 proteins stabilize[55,69,70]. The basis of DNE is mixed MUT/WT tetramers formation (heterotetramers)[68,71]. We and others speculate that for complete DNE to occur in solid tumors, the MUT/WT protein ratio has to greatly shift in favor of MUT[55,68]. Yet, while DNE requires acute stress to increase the low-abundance mutp53 protein and stoichiometrically overwhelm co-expressed WTp53, stress equally stabilizes WTp53 levels, thus not shifting the ratio. Only a highly active HSF1 chaperone system with its mutp53-selective accumulation can induce the required stoichiometric MUT predominance over WT. However, the repressive WTp53–HSF1 axis prevents such unilateral mutp53 stabilization, and might explain the strong selection pressure for p53LOH.

In myeloid malignancy (AML) complete DNE of the R248Q allele was observed[72]. Notably, however, baseline R248Q protein levels in untreated TP53 R248Q/+ isogenic AML cells were already very high, comparable to our homozygous R248Q/− levels in CRC, indicating constitutive stabilization of mutp53 in heterozygous AML. Importantly, our heterozygous CRC tumors typically lack mutp53 stabilization. This indicates that in heterozygous carcinomas the MUT/WT protein ratio does not

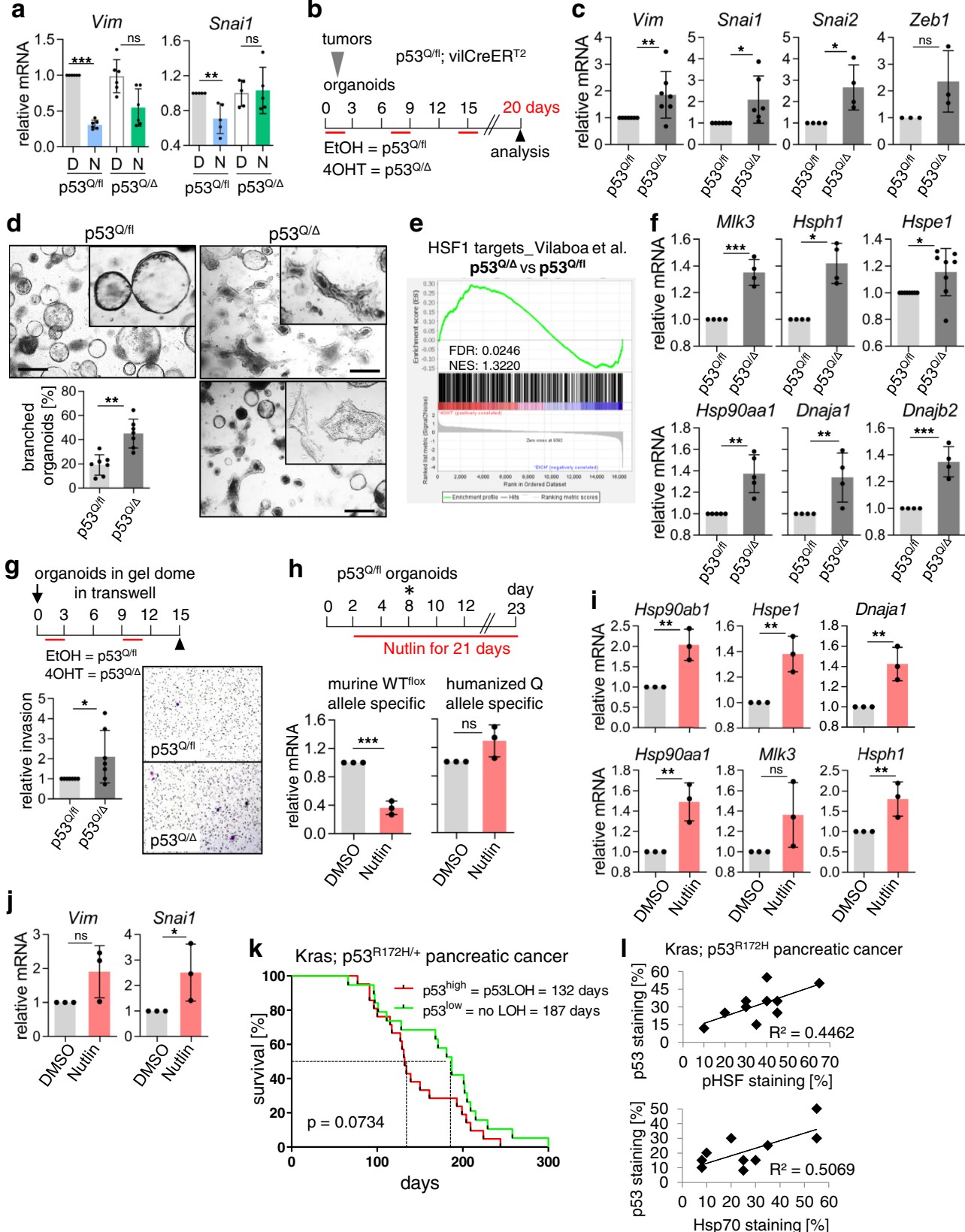

constitutively undergo the shift in favor of stable MUT. Hence, even after Nutlin activation we only see a partial DNE, leaving sufficient WTp53 activity intact to suppress the HSF1/Hsp90 axis and exert selection pressure for LOH. Conversely, heterozygous AML already appears to have a constitutive ratio shift favoring

stable MUT protein, which helps explaining the strong and exclusive DNE seen in AML cells[72].

AOM/DSS CRCs require WTp53 activation by Nutlin (mimicking high proliferative stress or chemotherapy) to fully regulate HSF1 (Fig. 2e and Supplementary Fig. 2d, e), patient

**Fig. 7 In murine CRC organoids p53LOH de-represses HSF1 activity and triggers mutp53-driven invasion. a** mRNA levels of Vim and Snai1 of indicated organoids treated as in Fig. 2c. Two days after p53LOH induction organoids treated with 10 µM Nutlin/DMSO for 24 h were harvested for qRT-PCR and normalized to $Hprt1$ mRNA. Mean ± SD from ≥3 independent experiments of ≥3 cultures. **b** Treatment scheme used in **c–e** for "long-term p53LOH" (p53$^{Q/\Delta}$) of p53$^{Q/fl}$;vilCreER$^{T2}$ organoids over 20 days, induced by adding weekly 4OHT for 48 h (red lines). EtOH, control (no LOH, p53$^{Q/fl}$). **c** mRNA levels of EMT markers in the indicated organoids generated in **b**. qRT-PCR normalized to $Hprt1$ mRNA. Mean ± SD of ≥3 independent experiments from two cultures. **d** Representative images of organoids. p53LOH induces branching, associated with invasiveness[7]. Quantifications of branched organoids per image field (4× magnification) as percentage of total organoid numbers. Nine to 15 images per condition were counted. Mean ± SD of seven independent experiments from two different cultures. **e** RNAs-eq analysis of EtOH-treated (p53$^{Q/fl}$) and 4OHT-treated (p53$^{Q/\Delta}$) organoids. Enrichment plots for HSF1 target gene set from Vilaboa et al. generated on heat-shock-induced Hela cells[64]. **f** mRNA levels of HSF1 target genes of indicated organoids generated as in **b**. qRT-PCR normalized to $Hprt1$ or $Rplp0$ mRNA. Mean ± SD of ≥3 independent experiments from two cultures. **g** Organoid invasion assay. (Top) treatment scheme of p53$^{Q/fl}$;vilCreERT2 organoids with weekly 4OHT or EtOH (red lines). (Right) representative images from the underside of transwells stained with crystal violet. (Left) quantifications of migrated cells. At least four fields per transwell were counted. Mean ± SD of five independent experiments from two different cultures. (**h**) Long-term chronic Nutlin treatment of heterozygous p53$^{Q/fl}$;vilCreER$^{T2}$ organoids used in **h** and **i** to induce spontaneous p53LOH. Two days after plating, organoids were treated with 5 µM Nutlin/DMSO for 21 days. At day 8, dead organoids were removed by splitting. See also Supplementary Fig. 7f. (Bottom) mRNA levels of the murine Trp53 allele and the humanized TP53 allele of heterozygous p53$^{Q/fl}$;vilCreER$^{T2}$ organoids using species-specific primers. qRT-PCR normalized to $Hprt1$ mRNA. Mean ± SD from two independent experiments, one included a technical replicate from the same organoid culture (total $n = 3$). **i, j** mRNA levels of HSF1 target genes (**i**) or EMT markers (**j**) of long-term Nutlin-treated organoids from **h**. qRT-PCR normalized to $Hprt1$ mRNA. Mean ± SD from two independent experiments, one with a technical replicate from the same organoid culture (total $n = 3$). **k** Survival curve of KPC mice stratified for p53$^{high}$ (p53LOH, $n = 25$ mice) versus p53$^{low}$ (no LOH, $n = 15$ mice). Kaplan–Meier statistic, log-rank test, $p = 0.0734$. **l** Correlations in quantitative immunostaining between p53 and (left) Hsp70 (HSF1 target) or (right) pHSF1 in pancreatic KPC tumors. See also Supplementary Fig. 7i. **k, l** Scoring used: p53$^{high}$ > 20% of tumor cells stained; p53$^{low}$ < 20% of tumor cells stained. **a**, **c**, **d**, **f–j** Student's $t$ test, two-sided. *$p ≤ 0.05$, **$p ≤ 0.01$, ***$p ≤ 0.001$; ns not significant.

p53LOH tumors intrinsically exhibit upregulated HSF1 targets compared to WTp53 tumors (Fig. 6). We speculate that heterozygous human tumors are sufficiently constitutively stressed to activate WTp53 and repress HSF1, and that upon p53LOH human tumors massively upregulate HSF1 activity. In contrast, we posit that baseline AOM/DSS-induced tumors have insufficient stress levels to drive p53LOH spontaneously, explaining the missing spontaneous p53LOH in the AOM/DSS model, compared to KRAS-driven mouse models of pancreas and lung cancer[3,9]. Notably, in human CRCs strong constitutive oncogenic stress arises from APC and K-RAS mutations[73,74]. Thus, baseline murine AOM/DSS CRCs might not be stressed enough to spontaneously activate WTp53 and suppress HSF1 (Supplementary Fig. 2l). This might also explain the preferred order in which cancer-causing mutations occur during tumorigenesis[74,75].

Our data provide one explanation why tumor heterozygosity tends to be unstable and why p53LOH strictly correlates with mutp53 stabilization and higher tumor aggressiveness.

## Methods
Detailed information of used materials is listed in Supplementary Table 1 and Supplementary Table 2.

**Mouse experiments and genotyping.** Experiments using animal materials were approved by institutional (Göttingen University Medical Center Ethikkommission) and state (Niedersächsisches Landesamt für Verbraucherschutz und Lebensmittelsicherheit, LAVES, Lower Saxony, Germany) committees, ensuring that all experiments conform to the relevant regulatory standards.

The humanized constitutive TP53$^{R248Q}$ (called p53$^Q$) knock-in allele has been described in detail[5,22,26]. Briefly, the human TP53 sequence containing the R248Q mutation in exon 7 replaces part of the mouse Trp53 (exons 4–9). To generate heterozygous mice with one conditional murine Trp53 WT allele (p53$^{fl}$), we crossed the p53$^Q$ allele with mice harboring the floxed WTp53 allele[76] flanked by loxP sites in introns 2 and 10 to generate p53$^{Q/fl}$. To remove the floxed WTp53 allele from colonic epithelial tissue, we crossed p53$^{Q/fl}$ mice with villinCreER$^{T2}$ (called "ERT2") transgenic mice. Moreover, the classic Trp53 knock-out mice (p53$^{-/-}$ mouse)[77] were crossed to the p53$^Q$ allele to generate non-tissue specific, constitutive TP53 alterations (e.g., Supplementary Figs. 1 and 2). For all genotypings, we isolated DNA with DirectPCR lysis Reagent (tail) (7Bioscience GmbH). PCR was performed with OneTaq® Quick-Load® 2× Master Mix (New England Biolabs) according to the manufacturer's guidelines, using the primers specified in Supplementary Table 2. All mouse strains were maintained on a C57BL/6 background for at least 6 generations. For experiments, randomly assigned 10-week-old males and females weighing at least 20 g were used. Mice were kept under pathogen-free barrier temperature-controlled (20–22 °C)

conditions, with a 12 h day and 12 h dark/light cycle, with free access to water and standard rodent diet.

Formaline-fixed paraffin-embedded (FFPE) samples from KPC mice were provided by Albrecht Neesse[78]. KPC mice were generated by interbreeding Kras$^{G12D}$, Tp53$^{R172H}$, and Pdx-Cre mouse strains[9].

**Cell culture, treatment, and transfection.** Human CRC cell lines RKO, LS513, LS174T (all harboring WTp53), and SW480 (harboring mutp53 R273H/P309S) were cultured in RPMI 1640 medium, isogenic HCT116 WTp53 and HCT116 p53null were cultured in McCoys medium, all supplemented with glutamine, 10% fetal bovine serum, and penicillin/streptomycin and grown in a humidified atmosphere at 37 °C with 5% CO$_2$. All cell lines were regularly tested for mycoplasma contamination using the MycoAlert Mycoplasm detection kit (Lonza).

siRNAs were purchased from Ambion/Thermo Fisher Scientific (siRNAs are specified in Supplementary Table 1) and transfected with Lipofectamine 2000 (Invitrogen). Nutlin-3a (BOC Biosciences), Palbociclib (Sigma), Idasanutlin (RG3788, SelleckChem), RG7112 (SelleckChem), RO3306 (Sigma), and Roscovitine (Cell Signaling) were dissolved according to manufacturer's guidelines and used as indicated.

For stable HSF1 expression in HCT116 cells, HEK293 cells were co-transfected with lentiviral packaging vectors (pMD2.G from Addgene and pCMV-R8.91 from PlasmidFactory Bielefeld) and the Precision LentiORF HSF1 lentiviral plasmid (Id: PLOHS_100008319) or a Precision control plasmid (Dharmacon). After standard lentivirus production, HCT116 cells were transduced in the presence of 8 µg/mL polybrene and cells were selected with Hygromycin for several days. Single-cell clones were expanded and validated for HSF1 overexpression by immunostaining with phospho-Ser326 HSF1 (Abcam). Cell clones (HSF1c1, HSF1c2, and ORF control) were cultured in McCoys medium and supplemented, as described above.

**CRC induction, colonoscopy, and treatment.** Murine colorectal cancer (CRC) was induced by a single intraperitoneal injection of the colon-selective carcinogen AOM (10 mg/kg in 0.9% sodium chloride, Sigma) at the age of 10 weeks. After 1 week rest, an acute colitis was induced with 1.5% (in p53-deficient mice) or 1.8% (in p53-proficient mice) DSS (MP Biomedicals) for 6 days in the drinking water.

Visualization of tumor growth by mini endoscopy/colonoscopy (Karl Storz GmbH) started 6 weeks after AOM induction. Tumor sizes were scored according to the Becker and Neurath score[79]. Briefly, tumor sizes are calculated relative to the width (luminal circumference) of the colon and scored as sizes 1–5 (S1–S5) with the following specifications: S1 = just detectable, S2 = 1/8 of the lumen, S3 = 1/4 of the lumen, S4 = 1/2 of the lumen, and S5 > 1/2 of the lumen. Notably, between 6–8 weeks post AOM ~80% of mice had at least one S3 tumor and at least three S2 tumors.

For analysis of TP53$^{R248Q}$ ("Q") mice with either a constitutive p53 WT (+) or KO (−) allele, we chose an endpoint type of analysis, ending at 12 weeks after AOM in p53-proficient mice (at least one WTp53 allele), or at 10 weeks after AOM in all p53-deficient mice (deleted or mutated). This design prevented loss of mice due to colonic obstruction, anal prolapse, or lymphoma development in p53-deficient mice.

For analysis of the inducible p53LOH mouse model, we used the $TP53^{R248Q}$ ("Q") allele combined with the conditional floxed WTp53 allele (p53$^{fl}$) to create heterozygous p53$^{Q/fl}$;vilCreER$^{T2}$ tumors. We specifically induced p53LOH after a defined endoscopy-verified tumor burden was reached (at least one S3 tumor in addition to at least three S2 tumors). After tumor verification, TAM (Sigma) was given by seven serial intraperitoneal injections (1 mg daily per injection in a 1:10 ethanol/oil mixture) to activate the inducible recombinase (villinCreER$^{T2}$) and cause p53LOH. Tumor growth was continued to be visualized by colonoscopy over 2–8 weeks after LOH induction by TAM.

At endpoints, all mice were euthanized and the entire colon and rectum were harvested. Colons were longitudinally opened, cleaned, and displayed. Tumor numbers were counted and tumor sizes measured with a caliper. Tumor biopsies were taken from all mice. To ensure complete sampling of the organ, each colon/rectum was "swiss rolled", fixed in 4% paraformaldehyde/PBS and bisected. Both halves were placed face down side-by-side into a single cassette for histologic processing, paraffin embedding, and subsequent tissue analysis.

Nutlin-3a (BOC Biosciences) treatment was given by oral gavage with 150 mg/kg (dissolved in 2% KUCEL/0.2% Tween-80) per dose over three consecutive days. Mice were sacrificed and colorectal tumors harvested 4 h after the last treatment.

**Histological analysis.** Standardized immunohistochemical staining were performed on murine FFPE tissues. The following primary antibodies were used and listed in detail in Supplementary Table 1: p53 FL393 (Santa Cruz); cyclinD1, pan-Cytokeratin, and α-smooth muscle actin/SMA (all Abcam); phospho-HSF1 (Bioss); and pRB and Hsp70 (both Cell Signaling). The ImmPRESS™ Peroxidase polymer reagent based on 3, 3-diaminobenzidine (DAB, Vectorlabs), or Alexa Fluor®488-coupled and Alexa Fluor®647-coupled secondary antibodies (immunofluorescence) were used as detection systems. Hematoxylin (DAB) or DAPI (immunofluorescence) were used as counterstains. Stained sections were analyzed by microscopy (AxioScope, Zeiss) with ZENblue software V3.0 (Zeiss). Grading for phospho-HSF1 and p53 was as follows: p53$^{high}$, >25% with intense nuclear staining; Hsp70$^{high}$, >50% with high intensity.

**Immunoblots.** Whole-cell protein lysates were prepared with RIPA buffer (1% TritonX-100, 1% desoxycholate, 0.1% SDS, 150 mM NaCl, 10 mM EDTA, 20 mM Tris-HCl pH 7.5, and complete protease inhibitor mix, Roche). Tumor tissues were minced and lysed with RIPA buffer followed by sonication. After centrifugation, protein concentrations were determined by BCA protein assay (Pierce). Equal amounts of protein lysates were separated by SDS–polyacrylamide gel electrophoresis, transferred onto nitrocellulose membranes (Millipore), blocked with 5% milk, and probed with the following antibodies: murine p53 (CM5, Vector Laboratories), human p53 (DO-1, Santa Cruz sc-126), total HSF1, pMEK1 and CDK1 (all Santa Cruz), HSP90α (Millipore), HSP27, AKT, cRAF, Bcl-Xl, CDKN1A/p21, phospho-RB, and phospho-S6 (all Cell Signaling), MLK3, phospho-Ser326 HSF1, and CDK2 (all Abcam), PLK4 (Protein Technologies), GAPDH and beta-Actin (both Abcam). Detailed information of antibodies are listed in Supplementary Table 1. Full scan blots of all generated immunoblots are provided as Source data file.

**Quantitative real-time PCR.** Total RNA from cells, tumor tissues, or organoids was isolated using the Trizol reagent, following manufacturers' guideline (Invitrogen/Thermo Fisher Scientific). Tumor tissues were first homogenized using a homogenizer (T10 basic ULTRA-TURRAX). Equal amounts of RNA were reverse-transcribed (M-MuLV Reverse Transcriptase, NEB), and qRT-PCR analysis was performed using a qPCR Master Mix (75 mM Tris-HCl pH 8.8, 20 mM (NH$_4$)$_2$SO$_4$, 0.01% Tween-20, 3 mM MgCl$_2$, SYBR Green 1:80,000, 0.2 mM dNTPs, 20 U/mL Taq-polymerase, 0.25% TritonX-100, and 300 mM Trehalose). qRT-PCR for mRNAs were normalized to $RPLP0$ or $HPRT1$ mRNA, dependent on cDNA input and CTs values of the housekeeping genes. For indicated qRT-PCRs, mRNAs were co-normalized to $RPLP0$ and $HPRT1$, meaning $RPLP0$ and $HPRT1$ were pipetted on the same plate as the gene of interest and calculated individually for each housekeeping gene. Relative values are given in [ratio ($2^{-ddCT}$)]. "Independent experiment" means a biological replicate. A "technical replicate" means an independent cDNA synthesis from the same mRNA. qRT-PCR runs were pipetted mostly in triplicates, but at least in duplicates. Primers are specified in Supplementary Table 2.

**Dual-luciferase reporter assay.** HSF1 firefly luciferase plasmids harboring seven HSE elements (pGL4.41 [luc2P/HSE/Hygro] vector) and the pRL (Renilla) luciferase reporter plasmid (pRL-TK) were purchased from Promega. Cells were seeded and 24 h later were co-transfected with 100 ng HSF1$Luc$ plasmids and 200 ng pRL-TK plasmid, using Lipofectamine 2000 (Invitrogen). Forty-eight hours post transfection, cells were treated as indicated and firefly luciferase and Renilla luciferase activities measured using a dual-luciferase assay. Briefly, cells were lysed with PLB (passive lysis buffer, 5× E194A) and incubated for 15 min. Supernatants were first incubated and measured with firefly luciferase buffer (25 mM glycylglycine, 15 mM K$_2$HPO$_4$, 4 mM EGTA pH 8.0, 15 mM MgSO$_4$, 4 mM ATP pH 7.0, 1.25 mM DTT, 0.1 mM CoA, and 80 µM luciferin) and then with Renilla luciferase buffer (1.1 M NaCl, 2.2 mM Na$_2$EDTA, 0.22 M K2HPO$_4$ pH 5.1, 0.5 mg/mL BSA, 1.5 mM

NaN$_3$, and 1.5 µM coelenterazine). Relative light units (RLUs) were measured in a Luminometer Berthold Centro LB 960 plate reader. Values were normalized to Renilla activity and relativized to the control treatment. Firefly expression normalized to Renilla expression and RLUs calculated.

**Murine organoids, media, culturing, and treatment.** For preparation of organoid media, HEK293T cells stably expressing mRspondin or mNoggin (kindly provided by Dr. Tiago De Oliveira), or mWnt3a cells were cultured in DMEM (Gibco) supplemented with GlutaMAX™ (Gibco), 10% FBS (Merck), penicillin–streptomycin (10,000 U/mL, Gibco), and sodium pyruvate (Gibco) in a humidified atmosphere at 37 °C with 5% CO$_2$. For HEK293T, mRpondin-I 300 µg/mL Zeocin (InvivoGen) and for HEK293T, mNoggin 500 µg/mL G418 (Geneticin, InvivoGen) were added to the medium during cultivation. After HEK293 cell expansions, culturing media were replaced by conditioned medium (CM) containing Advanced DMEM/F12 (Gibco) supplemented with GlutaMAX™ (Gibco), penicilli–streptomycin (10,000 U/mL, Gibco), and 10 mM HEPES (Gibco). A total of 50 mL of CM were added per 175 cm$^2$ flask and HEK293 cells allowed to grow for 1 week. Each CM media were sterilly filtered and aliquoted. Since mRspondin-I and mNoggin proteins are each fused to an Fc-tag, the quality of each batch was tested by Dot-blot analysis. Organoid media was composed of 50% CM Wnt3a, 20% CM mNoggin, 10% CM mRspondin-I, N2 and B27 (both Gibco), 5 µM CHIR 99021 (Axon Medchem), 3.4 µg/mL ROCK inhibitor (Y-27632), 500 nM A83-01, 10 mM nicotinamide (Sigma-Aldrich), 80 µM N-acetyl-L-cysteine (all Sigma-Aldrich), and 200 ng/mL rmEGF (ImmunoTools).

For organoid preparation, tumor-harboring mice were sacrificed, and the colons harvested. Tumors were dissected, washed, and minced, and incubated with 2 mg/mL collagenase type I solution (Gibco, dissolved in advanced DMEM/F12) at 37 °C for 30 min, while pipetting up and down every 10 min to dissociate the tumors. Small tumor fragments were transferred into a new Falcon tube using a cell strainer (100 µm mesh size). Fragments were centrifuged and washed with advanced DMEM/F12. After centrifugation, tumor fragments were resuspended in cold Matrigel (Corning) and plated as gel drops on culture plates. After Matrigel polymerization at 37 °C, organoids were cultured in organoid media and cultivated in a humidified atmosphere at 37 °C with 5% CO$_2$. Medium was exchanged every 2–3 days. Splitting of organoids was performed when organoids started to accumulate dead cells in the lumen (approximately once a week). To this end, organoids were recovered from Matrigel and disrupted manually by pipetting using 1 mL blue tips. For enzymatic dissociation, organoids were incubated with 0.25% trypsin at 37 °C for 10 min, washed with advanced DMEM/F12, centrifuged, and cultured as described above. Experiments with colonic organoids were done between passages 3 and 8. p53LOH was induced with 1 µM 4OHT (Sigma) for 24–48 h (as indicated in the figure legends) in CHIR 99021-free and Rock-free organoid media. Nutlin was also used in CHIR 99021-free and Rock-free organoid media.

Quantifications of branched organoids per image field (4× magnification) as percentage of total organoid numbers. An organoid with at least one protrusion was counted as "branched" organoid. Nine to 15 images fields per condition were counted.

For statistics "an organoid culture" was generated from two to three tumors from one mouse. "Independent experiments" are generated from different passages of an organoid culture.

**Immunofluorescence staining of organoids.** Organoids were fixed within Matrigel domes with 2%/0.1% paraformaldehyde/glutaraldehyde/PBS for 30 min. After intensively washing steps with PBS, gel domes with fixed organoids were removed from the plate and transferred into a tube. Sucrose infiltration was started with 20% sucrose/PBS, followed by 40 % sucrose/PBS, each incubated overnight or longer at 4 °C until the domes settled down. After sucrose infiltration, organoids were embedded in TissueTEK (Tissue-Tek® O.C.T™ Compound) and 10 µm cryosections were cut. Sections were air-dried for 30 min at RT, pre-wetted with PBS and quenched with 10 mM NaBH4/PBS twice for 5 min at room temperature each time. After washing steps, samples were permeabilised with 0.1% TritonX-100/PBS for 10 min at RT and blocked with 10% FBS/1% BSA/PBS for 1 h. For staining, samples were co-incubated with the p53 antibody FL393 (Santa Cruz) and E-Cadherin (BD Biosciences) overnight at 4 °C. Primary antibodies were detected by AlexaFluor488- and AlexaFluor647-conjugated secondary antibodies (Molecular Probes). Organoids were DAPI counterstained and mounted in Fluoromount media (DAKO). Images were taken using a standard fluorescence microscope (Carl Zeiss AG) with the ZENblue imaging software V3.0 from Zeiss. Figures were further prepared using Adobe Photoshop software (version 12.0 × 32). Nuclear p53 scoring was done as follows: score 0 = no positive nucleus per organoid; score 1 = 1–20% positive nuclei per organoid; score 2 = 20–50% positive nuclei per organoid; and score 3 > 50 % positive nuclei.

**Organoid invasion assay.** Organoids were plated as small 20 µL gel drops (gel domes) in pre-equilibrated transwell inserts. A total of 1 mL of normal organoid medium were added to the well plate (lower chambers) and 500 µL of EGF-free and diluted organoid medium (1:1 with RPMI pur) were added in the organoids-containing inserts (upper chambers of inserts). One day post plating, first 4OHT

treatment started for 24 h, followed by two weekly repetitions. After long-term p53LOH, inserts were prepared for fixation and staining. Therefore, organoids in gel domes from the upper layer of the inserts were gently scraped. Migrated cells on the lower side of inserts were fixed in 100% methanol for 10 min, stained with crystal violet (0.1% (w/v) crystal violet and 20% (v/v) EtOH) for 20 min and washed with water. After air drying of inserts, membrane of the inserts were visualized and imaged under the microscope.

**Analysis of human patient TCGA data.** We used TCGA CRC (COADREAD) and breast cancer (BRCA) databases in this analysis. Human genomic data, including RNA expression, DNA copy number alteration, gene mutation, and clinical information, was downloaded from cBioPortal (http://www.cbioportal.org/study/summary?id=coadread_tcga_pan_can_atlas_2018). Study names: colorectal adenocarcinoma (TCGA, PanCancer Atlas, 594 total samples) and breast invasive carcinoma (TCGA, PanCancer Atlas, 1084 total samples). *TP53* WT (WTp53) group are those samples without *TP53* mutations. *TP53* MS mutant group was samples with *TP53* MS (missense) mutations, and *TP53* LOF group was determined by samples with all *TP53* mutations (MS, FS = frameshift, and NS = nonsense). To identify tumors harboring p53LOH, we selected samples that had both a mutated *TP53* gene and a shallow deletion in DNA copy number. Two independent lists of HSF1 target genes were chosen: a set of direct target genes from Mendillo et al.[46], identified by integrated ChiP-seq/RNA-seq after HSF1 knockdown in breast cancer cells, and a set of genes from Vilaboa et al.[64] in HS treated HeLa cells (Supplementary Data 1, upregulated genes after heat-shocked HeLa cells). We compared the expression values (by RNA-seq) of HSF1 target genes from mutant p53/p53LOH tumors with samples that harbored WT *TP53* ($TP53^{+/+}$). Further we applied survival analysis to check patients with *TP53* mutation (MS, NS, and FS) and a p53LOH compared to a WTp53 patient group. R language (The R Project for Statistical Computing, https://www.r-project.org) was used in the analysis. R package "gplots" was used to generate heatmaps. R package "survival" were used for survival analysis, including calculating log-rank $p$ values and generating Kaplan–Meier curves.

**mRNA-sequencing and analysis of murine tumors and organoids.** Tumor RNA samples were generated as pools. For $p53^{+/+}$ tumors, two biological replicates (R) each consisting of two pooled tumors (R1 = T1–2 and R2 = T3–4 out of two mice) per treatment group (DMSO versus Nutlin) were analyzed. For $p53^{-/+}$ and $p53^{Q/+}$ tumors, three biological replicates (R) each consisting of three pooled tumors (R1 = T1–3, R2 = T4–6, and R3 = T7–9 out of three mice) per treatment group were analyzed. Organoid RNA samples (R1–R3) were generated from different organoid cultures treated with EtOH or 4OHT. RNA samples were sequenced by Novogene (Cambridge Science), including mRNA library preparation (poly A enrichment), NovaSeq 6000 PE150 sequencing, and raw Data quality control. The GALAXY environment (https://galaxy.gwdg.de/) was used to process raw-sequencing data. Quality check was performed with FastQC (version 0.72). Reads were trimmed (11 bp from the 5′ end, FASTQ Trimmer tool version 1.1.1) and subsequently aligned to the murine genome (GRCm38, RNA STAR, version 2.5.2b-2). Finally, aligned reads were assigned to genomic features with FeatureCounts (version 1.6.3). DESeq2 (version 2.11.40.6) was carried out with default parameters for differential expression analyses. Upregulated genes (base mean > 10, log2 (fold change) > 0.7, and $p$ value < 0.05) were selected for Enrichr analyses (https://maayanlab.cloud/Enrichr/). For WTp53 target genes normalized counts generated from DESeq2 were used to perform GSEA (Broad Institute, version 4.1.0) with the following parameters: number of permutations = 1000, type = gene_set, no_collapse, max size = 1000, and min size = 15. For HSF1 analysis, the HSF1 target gene set was extracted from Vilaboa et al.[64] (Supplementary Data 1, upregulated genes after heat-shocked HeLa cells) and uploaded in GSEA as GMX file.

**Quantification, statistical analysis, and reproducibility.** Statistics of each experiment, such as number of animals, number of tumors, biological replicates, technical replicates, precision measures (mean, ±SEM and ±SD), and the statistical tests used for significance are provided in the figures and figure legends. Unpaired (two-sided) Student's $t$ test was used to calculate the $p$ values for comparisons of tumor numbers and sizes and mRNA expression levels.

Densitometric measurements for quantification of immunoblot bands were done with the gel analysis software Image Lab™ (BioRad, version 5.2.1) and normalized to loading controls. Dot plots were generated with EXCEL or GraphPad PRISM 9 [version 9.1.0 (221)].

The following designations for levels of significance were used within this manuscript: *$p \leq 0.05$; **$p \leq 0.01$; ***$p \leq 0.001$; ns, not significant.

The following list of figures with representative images indicates in brackets how many times each experiment was repeated independently with similar results. Figure 1f, g ($p53^{Q/+}$ + TAM-treated $n = 27$ tumors out of seven mice and $p53^{Q/fl}$ + TAM $n = 49$ tumors out of eight mice); Fig. 1i ($p53^{Q/fl}$ + TAM $n = 22$ tumors out of three mice); Fig. 3e (four independent experiments); Fig. 3f, g (two independent experiments); Fig. 3i (two independent experiments); Fig. 4a, c, e (two independent experiments); Fig. 4f (one independent experiment); Fig. 4g, h (two independent experiments); Fig. 4i ($p53^{Q/+}$ + DMSO $n = 15$ tumors out of two mice and $p53^{Q/+}$ + Nutlin $n = 18$); and Fig. 5b, e, f, g (two independent experiments);

Supplementary Figure 1b ($p53^{-/+}$ and $p53^{Q/+}$ mice $n = 7$ each, and $p53^{-/-}$ and $p53^{Q/-}$ mice $n = 16$ each); Supplementary Fig. 1e ($p53^{Q/-}$ mice $n = 16$); Supplementary Fig. 1g ($p53^{Q/fl}$ + TAM mice $n = 22$ tumors out of three mice); Supplementary Fig. 1f ($n = 6$ $p53^{Q/fl}$ + TAM tumors were analyzed); Supplementary Fig. 2o ($n = 8$ different passages from three organoid cultures were analyzed); Supplementary Fig. 3b–d (two independent experiments); Supplementary Fig. 3e (three independent experiments); Supplementary Fig. 4c, d (two independent experiments); Supplementary Fig. 5a, b (two independent experiments); Supplementary Fig. 5d (four independent experiments); Supplementary Fig. 5e (for HCT116 and RKO cells three independent experiments and for LS174T two independent experiments); Supplementary Fig. 5f (one experiment); Supplementary Fig. 6j (two independent experiments); Supplementary Fig. 7f (three independent experiments); and Supplementary Fig. 7i ($n = 10$ tumors were analyzed).

**Reporting summary.** Further information on research design is available in the Nature Research Reporting Summary linked to this article.

## Data availability

mRNA-seq data of Nutlin-treated murine tumors were deposited in the public repository Array Express database under the accession number E-MTAB-10041. mRNA-seq data of murine organoids after p53LOH generated during this study are available under the accession number E-MTAB-10416 in the public repository Array Express database. The remaining data are available within the article, Supplementary Information or Source data file and available from the authors upon request. Source data are provided with this paper.

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

## Acknowledgements

We thank Nina Pfisterer and Lukas Gebauer (Molecular Medicine M.Sc.-Program Göttingen) for technical assistance. R.S.-H. is supported by the DFG (SCHUH3160/3-1), the Heidenreich-von-Siebold Program (University Medical Center Göttingen), and the KH-Bauer Program (G-CCC Göttingen). A.N. and R.S.-H. are funded by the Clinical Research Unit KFO5002 (DFG). A.N. is supported by the German Cancer Aid (70113213, Max Eder group), and U.M.M. by NIH NCI (2R01CA176647), Deutsche Forschungsgemeinschaft (MO1998/2-1), and the Stony Brook Foundation TRO program.

## Author contributions

Conceptualization: R.S.-H.; methodology: T.I., Ö.Ç.Ş., A.S., L.K., A.N., J.L., F.W., U.M.M., and R.S.-H.; experimentation: T.I., Ö.Ç.Ş., A.S., L.K., N.W., J.L., F.W, and R.S.-H.; interpretation of data: T.I., Ö.Ç.Ş., A.S., L.K., N.W., A.N., J.L., F.W, U.M.M., and R.S.-H.; writing original draft: U.M.M. and R.S.-H.; writing review and editing: R.S.-H., U.M.M. and all authors; funding acquisition: R.S.-H. and U.M.M; and supervision: R.S.-H.

## Funding

## Competing interests

The authors declare no competing interests.
