## [Peer Review File · Nature Communications]

REVIEWER COMMENTS

Reviewer #1 (Remarks to the Author):

Using two very good experimental mouse models, the authors first showed that mutp53R248Q is stabilized in colorectal cancer only when WTP53 is absent. Moreover, only such tumors are invasive. This is essentially consistent with existing knowledge that in human cancers, the presence of mutp53 is associated with advanced stages of the disease, metastasis, recurrence, and patient's poor prognosis, even when compared with TP53 deletion.

Then, the authors tested the hypothesis that despite the presence of a GOF allele (R248Q), the remaining WTP53 allele at least partially retains its transcriptional activity. They assumed that WTP53 is constitutively activated (e.g. by aberrant growth and metabolic stress, hypoxia, and genomic instability) in tumors and that nutlin treatment would mimic these conditions.

The crucial to state if WTP53 is activated by nutlin in the presence of GOF allele (R248Q) are results shown in Fig. 2C. The degree of activation of known p53 targets (3 from 4 tested genes slightly activated) does not convince that WTP53 (in p53Q/+ tumors) becomes active (very small changes estimated by RT-qPCR normalized to only one reference gene are not reliable). Nevertheless, Fig. S2F shows that one copy of WTP53 can activate its target genes, or rather, that known p53 targets are expressed in p53-/+ and p53Q/+ tumors.

The observation that some HSF1 target genes are downregulated after nutlin treatment (i.e. p53 activation) is really interesting. Hsph1, Hspe1, Hsp90aa1, and Itgb3bp are shown in Fig. 2D as repressed. On the other hand, at least Hspe1, Hsp90aa1 and Itgb3bp basal expression does not depend on WTP53 or mutp53R248Q, although Hspa1a and Hspbp1 expression is elevated in p53-/- and p53Q/- tumors (Fig. S2G).

At this step, based on shown results, it is difficult to agree with the conclusion (line 169) that "stress-activated WTP53 prevents chaperone-mediated mutp53 stabilization (Figure 2D)" and especially that "WTP53 in heterozygous mutp53/+ tumors creates the driving force for p53LOH" (line 171). This is simply not shown here. Such a possibility should be discussed in the Discussion section.

The authors proved that WTP53 activation by nutlin is connected with a lower level of HSF1 S326 phosphorylation and lower mRNA level of some HSF1-dependent genes. They claimed that p53 suppresses HSF1 activity via CDKN1A/p21, CDK4/6. Thus, in fact, the cell cycle inhibition led to the repression of HSF1 activity (measured as S326 phosphorylation and expression of some target genes). It seems that not typical HSF1 targets (e.g. CDC6, ITGB3BP) are affected stronger than typical ones (HSPs). This suggests that another signaling is highly possible, even bypassing HSF1. Since HSF1 can be differentially phosphorylated and activated/deactivated depending on the "stressor" (see e.g. Asano et al.; Sci Rep. 2016;6:19174. doi: 10.1038/srep19174. IER5 generates a novel hypo-phosphorylated active form of HSF1 and contributes to tumorigenesis) it could be that S326 is not the best choice to monitor HSF1 activity in this case (although very useful for heat-induced activation). To prove that HSF1 is here a key signaling component, its knockout should be tested.

It is widely accepted that mutp53 is stabilized by chaperones. Obtained data showing a correlation between WTP53 activation and HSPs down-regulation are consistent with the observation that as long WTP53 is present - mutp53 is not stabilized. According to this observation, the up-regulation of HSF1 targets should be expected in WTP53-deficient tumors. In Fig. S2G, only Hspa1a (known HSF1 target) and Hspbp1 (not necessarily transcriptionally regulated by HSF1) are elevated.

To summarize:

From the Abstract:

"We find that the WTP53 allele in AOM/DSS-induced colorectal tumors (CRC) of p53R248Q/+ mice retains its haploid transcriptional activity. Notably, WTP53 represses heat-shock factor 1 (HSF1) activity, the master transcription factor of the proteotoxic stress defense response (HSR) that is ubiquitously and constitutively activated in cancer tissues. HSR is critical for stabilizing oncogenic proteins including mutp53. WTP53-retaining murine CRC tumors and tumor-derived organoids and human CRC cells all suppress the tumor-promoting HSF1 transcriptional program."

Actually, it was only shown that WTP53 could potentially (e.g. after nutlin treatment) suppress the expression of some HSF1-dependent genes. And WTP53-retaining murine CRC tumors and tumor-

derived organoids and human CRC cells may potentially suppress the tumor-promoting HSF1 transcriptional program. Therefore, although highly probable, I treat the above statement as speculation based on correlations shown in the manuscript and existing literature data.

“Mechanistically, the retained WTP53 allele activates CDKN1A/p21, leading to cell cycle inhibition and suppression of the E2F target gene MLK3. MLK3 links cell cycle to the MAPK stress pathway to activate the HSR response. We show that in p53R248Q/+ tumors WTP53 activation by constitutive stress (emanating from proliferative/metabolic stresses and genomic instability) represses MLK3, consequently inactivating the MAPK-HSF1 response necessary to ensure tumor survival. This creates strong selection pressure for p53LOH which eliminates the repressive WTP53-HSF1 axis and unleashes the tumor-promoting HSF1 functions, inducing mutp53 stabilization and enabling invasion.”

Repression of MLK3 was shown in cell lines, not in tumors. Why it creates selection pressure for LOH?

There are many serious and minor errors in the manuscript, some of which (and additional comments) are listed below.

1. Fig. 1G – check the description: the left and right graphs show contradictory results. The wrong legend.

2. There is no information about how effective is WTP53 removal after Tamoxifen treatment in the experiment shown in Fig. 1. On the other hand, it is not so important since these mice are used later only for organoid culture and here the efficiency of recombination is shown.

3. Line 159: Fig. 2C; “We conclude that, surprisingly, the GOF mutp53R248Q allele fails to exert a dominant-negative effect over the remaining WTP53 allele as predicted by many, mainly in vitro, studies.”

Based on results shown in Fig. 2C it is difficult to agree with such statement.
and

Line 333: “we find in our CRC model that the remaining WT allele in heterozygous tumors is fully activatable, excluding a dominant-negative effect (DNE) by the counterpart mutp53 allele”.

Presented data does not support this conclusion. p53Q/+ tumors should be compared to p53-/+ tumors. Such a comparison is shown only in Fig. S2F. Assuming that RT-qPCR results are reliable and Cdkn1a is regulated mainly by WTP53 (as indicated by lack or very low level of expression in p53-/- and p53Q/- tumors; although it is still detected in p53Q/- tumors and organoids after TAM/4OHT-induced CRE/loxP recombination) its expression is lower in p53Q/+ tumors than in p53-/+ and p53+/+ tumors.

4. Fig. 2C, 7B, and line 156 – mistakes in description: Cdnk1a instead Cdkn1a

5. Supp Fig. 2G – HspA1A mRNA level is shown. There is no primers (mouse) for this gene in Table S2.

6. Line 189: “Nutlin also suppressed the tumor-promoting HSF1 targets CDC6, ITGB3BP, RBBP5, BST2 and FBLN1 (Figure 3C)”

BST2 is not suppressed.

7. Fig. 3E: “Activated WTP53 suppresses pSer326-HSF1”

Correlation is only shown, not HSF1 suppression by WTP53.

8. Line 206-207: “Nutlin strongly dephosphorylated pSer326-HSF1 (Figure 3G)”

Correlation is only shown, not HSF1 dephosphorylation by nutlin.

9. “HSF1 binding to HSE-Luc reporter was measured”

Activity of Luc was measured, not HSF1 binding.

10. Line 632: Supp Fig. 5: “(A, B) Depletion of CDK1 (A) and CDK2 (B) fail to abrogate HSF1 activity.”

Only HSF1 phosphorylation at S326 was shown, not HSF1 activity.

11. Supp Fig. 3F – HSPAA1 ? Probably HSP90AA1.

12. Fig. 4F – two lower blots do not fit to the upper ones.

13. Fig. 5F and G – please check GAPDH which seems to be the same on both blots.

14. Line 248 - pSer-325 – should be 326.

15. Line 656: Fig. 6B: “The mean survival of WTP53 patients (n=214) is 83.2 month versus 57.2 months for patients with MS plus p53LOH 658 (mutp53/-)”.

The description is not consistent with the Figure which indicates that WT p53 patients have shorter survival time. The wrong legend.

16. Fig. 6C and S6A-C: Hierarchical clustering should be performed in such analyses. Red/blue

Row Z-Score – not properly labelled (-1, -2, 0, 1, 2).

17. Why it was stated that HSF1 negatively regulates a subset of target genes (line 667). Only correlation between p53 status and the level of expression of the potential HSF1 target genes in colorectal adenocarcinoma patients is shown.

18. Line 275: “BRCA cells repressed pSer326-HSF1 when harboring WTp53 but not mutp53 (Figure S6D)”.

Based on blots shown in S6D it can be concluded that nutlin does not activate p53 in MDAMB-231 (its level is stable). It looks that there is some technical problem with pHSF1 blot from MCF7. Thus, the conclusion (line 275) is not well supported.

19. 1124-1125: “Densitometric measurements for quantification of immunoblot bands were done with the gel analysis software Image Lab™ (BioRad) and normalized to loading controls.” Such analyses are not present in the manuscript.

20. I would like to know what kind of statistics was applied to calculate statistical significance in qPCR when “mean ±SEM of 2 independent experiments, each repeated twice in triplicates” is shown.

Reviewer #2 (Remarks to the Author):

In this paper Sener et al investigate the mechanism of P53 LOH in colorectal cancer. Using a number of animal and cell culture models they propose that loss of wt P53 in R248Q tumours is required for HSF1 activation which in turn promotes the gain of function effects of R248Q mutations. This is an interesting study which I think casts some light on the mechanisms and importance of P53 LOH. The data mostly appears robust and the conclusions around the function of P53 in suppressing HSF1 activation and P53 LOH appear sound. However, a number of further analyses are required to convince that R248Q has gain of function properties particularly in light of recent findings that P53 point mutations do not confer GOF properties (S. Boettcher et al., Science 365, 599 (2019)). I think that a number of additional experiments are needed to address this point or, if not possible, the language on GOF should be toned down throughout the manuscript:

Major points

The overall tone of the paper is that R248Q confers a GOF phenotype when it is stabilised in the context of loss of WT P53. This is based on prior literature from the same lab and also on the experiments carried out in Fig 1 and S1. However, an alternative explanation could be that R248Q mutation inactivates P53 and LOH of WT P53 leads to essentially a null allele. It would be helpful to expand on the previous paper and strengthen these arguments for this paper. For example, the experiments in Fig 1 should include P53^{fl/fl} mice as a control. Comparing R248Q/+ to R248Q/fl does not allow determination of a gain of function effect over complete loss of P53 function. In this cohort, tumours from these groups should be harvested at an early point post induction (similar to Fig 1E) and compared by RNAseq/histology etc for changes that would be consistent with R248Q conferring gain of function effects and not acting solely as a loss of function allele.

The authors should also compare the tumour burden and invasiveness in their model S1C and D between P53^{-/-} and P53⁻/R248Q mice, are these changes significant? Previous reports (ie Schwitalla et al., Cancer Cell) have shown that loss of P53 is sufficient to confer an invasive (and metastatic) colorectal tumour phenotype so losing P53 activity is clearly sufficient for this without gain of function mechanisms. The authors also find evidence of invasive tumours in their P53^{-/-} mice although it does look more prevalent in the R248Q mice. Again, more analyses of tumours from this model would help convince of the GOF effects of R248Q (for example RNAseq and histological analysis of non invasive tumours from both models (to ensure like for like comparisons)). The human data in figure 6 also isn't particularly convincing and doesn't strengthen the case for a GOF effect of R248Q. What does survival look like if comparing truncating, stop gain LOH tumours? It would be expected that if missense mutations confer GOF then these should have worse prognosis than truncating mutations.

Minor points

- 1) There is some data quantification lacking. Throughout the manuscript, Western blots are shown with no quantification of the data. This needs to be included throughout.
- 2) Figure 4 isn't particularly convincing. siRNAs should be used to validate the role of cdk2/4

particularly as the authors outline the lack of specificity of the inhibitors used.

3) To strengthen the link to MAPK signalling, inhibitors of this pathway (ie MEK inhibitors) should be used and effects on HSF1 stress response measured.

4) There is a lack of evidence from the in vivo model to support the in vitro mechanistic data. IHC analysis of the nutlin treated tumour model for components of the downstream altered pathways should be carried out.

5) Fig 7F. The authors propose that following LOH 'upregulation of EMT genes and pro-invasive HSF1 target genes like HspH167 and Itgb3bp enable invasiveness in a stressed tumor milieu following p53LOH'. This statement isn't well supported by the data and should be toned down.

Overall, I think this is an interesting paper and provides some mechanistic insight into the potential role of P53 LOH in colorectal cancer. However, at the moment, I don't think the data supports statements such as 'Our data provide an explanation for the longstanding puzzle why tumor heterozygosity tends to be unstable and why p53LOH strictly correlates with mutp53 stabilization and higher tumor aggressiveness.' Further data is needed to support statements like this and/or a more balanced discussion is needed.

Reviewer #3 (Remarks to the Author):

In this manuscript, the authors propose that p53 loss of heterozygosity (LoH) serves as the major driving force behind the progression and invasiveness of colorectal tumors (CRC). While p53 mutation rate is elevated in this type of cancers, mutant p53 require a stabilization process in order to exert their gain of pathogenic functions (GoF). The authors show that the loss of the remaining wildtype (wt) p53 allele in heterozygous tumors is a pre-requisite for the stabilization of mutant p53 proteins. The manuscript by Sener, Stender and coworkers identifies the proteotoxic stress defense response (HSR) and particularly HSF-1, its main effector, as the key factor behind p53 LoH. HSF-1 is indeed kept in check as long as a residual wtp53 remains through a repressive axis: WTP53-p21-MLK3-MAPK-HSF1 thereby preventing mutp53-R248Q protein stabilization and GoF in mutp53/+ tumors.

The provided data shed a new light on the crucial yet understudied process that is p53 LoH which will prove valuable to the field. The manuscript is well-written, clearly illustrated and the authors rely on a relevant mouse model. However, several aspects need to be clarified and properly reviewed by the authors.

Point by point:

1. Figure 1 and S1: the authors use 2 pertinent intestinal/colorectal models to clearly demonstrate an LOH-induced mutant p53 stabilization, which consequently increases colorectal tumors. However, the total number of tumors (Figure 1D, left panel) and the total mouse number with invasive tumors (figure 1G, right panel) in the TAM-inducible model are unchanged in the "LOH mice" versus "no LOH mice". Only size (tumor growth) and invasiveness are changed. This should be clearly indicated in the text describing the figure.

2. Figure 2 and S2: The presented data indicate that wtp53 retains the ability to activate several transcriptional targets (e.g. Cdkn1a, Gadd45a and Sfn) but not Mdm2. This interesting point needs to be clarified. Are the observed effects (i.e. no dominant negative effect of p53-R248Q, suppression of HSF1 target genes by residual WTP53 activity) restricted to the use of nutlin or do other p53 inducers (irradiation, uv, doxorubicin...) leads to the similar effects?

The authors conclude to an absence of DNE from p53-R248Q although Cdkn1A and Bbc3 (S2-F) show a marked reduction in p53Q/+ vs. both p53+/+ and p53+/- without nutlin could suggest otherwise. Nutlin treatment leads to a balanced increase of both wt and mutant forms of p53 (p12-L332) thereby limiting the observation of a DNE which requires a high mt/wt ratio. The statement should be amended in order to reflect a more balanced conclusion as it is the case in discussion.

3. Figure 3: The authors have knocked p53 out using shp53 in homozygous WTP53 HCT116 and

RKO lines to demonstrate that Nutlin-induced HSF1 suppression is rescued by p53 depletion. Conversely p53^{-/-} HCT116 cells can be used to express WTp53 and induce an expected repression of HSF-1 to complete the analysis. Is p53-R248Q unable to activate HSF-1 in HCT116?

4. The ratio between wt and mutant p53 appears crucial to mutp53 DN activity which could be clarified by using a response dose of mutp53 in p53^{-/-} HCT116 cells. This should lead to a gradual effect on HSF-1 while the opposite could be observed by adding increasing doses of wtp53. Such effects on p53 targets have been documented in model organisms (see Brachmann et al. 1996, Monti et al. 2002, 2011, Billant et al. 2016).

Minor points:

1. It is indicated in the Methods section the use of HCT115 p53 null cells. Are they HCT 116 p53^{-/-} cells?

2. Sup Figure 3A: error bars are not homogeneous

3. Some typographical errors, ex.: In the chapter: "Activated WTp53 represses HSF1 activity in human colorectal cancer cells": line 197: Conversely, mutp53-harboring CRC...

5. The title suggests that the authors have studied the role of HSF-1 in mutant p53-mediated invasion thoroughly, but additional validation is required, preferentially using invasion assays to provide further knowledge of HSF1-mediated EMT process, particularly regarding the already largely accepted anti-invasive activity of p53.

Conversely, the role of HSF-1 on pro-invasive and EMT induction activities of mutant p53 should be clarified, in regards to its GOF activity versus dominant negative activity on WTp53.

6. Figure 6: as invasion is the acquired phenotype of p53LOH tumors, the analysis of tumor samples should also include "disease free survival" analysis, not only global survival.

7. Figure 6: It is claimed (in the discussion section) that in human CRCs strong constitutive oncogenic stress from K-RAS/EGFR/ TGFβR/ PDGFR mutations are preeminent, which promotes sufficient proliferative stress levels to drive p53LOH spontaneously. To strengthen this point, the analysis should also provide the mutation status of K-RAS/EGFR/ TGFβR/ PDGFR. Is there a correlation between MS p53 + LOH and K-RAS/EGFR/ TGFβR/ PDGFR mutations?

8. Figure 6: On the same line, does upregulation of HSF1 targets correlate with mutation status of K-RAS/EGFR/ TGFβR/ PDGFR?

9. Figure 7B: How do the authors explain the increase in Mdm2 and Cdkn1a expression after treatment with Nutlin of 4OHT (p53LOH)?

10. Figure 7E: the figure is not sufficiently described, in particular for the E-Cadherin staining. It remains unclear what we need to see. Is it a loss of the E-Cadherin staining, in agreement with an EMT process in p53LOH organoids?

11. Figure 7F: is there any significant difference in miR34 expression between EtOH+Nutlin and 4OHT+Nutlin, compared to their respective DMSO controls?

More importantly, there is no difference between EtOH-DMSO in Heterozygous organoids and 4OHT-DMSO in p53LOH organoids (in miR42a, Vimentin and Snai1 expression), suggesting that the EMT process is not complete. The authors should clarify this point.

12. Figure 7: to conclude on an EMT process:

- The levels of E-Cadherin should be analyzed and compared to N-Cadherin/Vimentin/Snail levels. Is there a Cadherin shift associated with EMT process?

- The effect of HSF-1 on EMT process (i.e. miR34, Ecadherin/ Vimentin/Snail levels) should be analyzed, possibly in cell lines (HCT116, RKO...)

13. The authors speculate that WTp53-mediated HSF1 suppression exerts a strong selection pressure for p53LOH. However, they also show that “murine CRCs might not be stressed enough to spontaneously activate WTp53 and suppress HSF1”, indicating that baseline AOM/DSS-induced tumors have insufficient stress levels to drive p53LOH spontaneously. The authors speculate that “in human CRCs strong constitutive oncogenic stress from K-RAS/EGFR/ TGFβR/ PDGFR mutations are preeminent”, providing sufficient proliferative stress levels to drive p53LOH. However, the link between K-RAS/EGFR/ TGFβR/ PDGFR mutations and HSF1 activation/repression is not addressed in this manuscript.

Point-by-Point Rebuttal Letter

We are very grateful and would like to thank *all three* Reviewers for their very careful, constructive and generally very positive evaluation of our paper. Their insightful comments were really helpful and led to improved clarity and impact of this work. In response, we performed many new experiments and extended our analyses which substantiated our conclusion. Also, the text has been clarified and toned down where appropriate.

Attached is the Point-by-Point Rebuttal letter. Reviewer text in black, our answers in blue. The rebuttal is Reviewer- specific to make it easier on our Reviewers and not send them looking all over the place. Where helpful in answering the question, we also added some previously published data. Some newly generated data are shown as Reviewer-only Figures since they were not deemed essential for the paper. In sum, we are confident that all remaining concerns have now been adequately addressed.

Added text is in RED. Graphical abstract provided.

The following figures were newly generated and are marked in RED.

Figure 2B	Suppl Figure S1G
Figure 2F	Suppl Figure S2B
Figure 5H	Suppl Figure S2D
Figure 4I	Suppl Figure S2N
Figure 5H	Suppl Figure S2P
Figure 5I	Suppl Figure S4C
Figure 5J	Suppl Figure S4D
Figure 6D	Suppl Figure S5F
Figure 6E	Suppl Figure S6B
Figure 6F	Suppl Figure S6C
Figure 6G	Suppl Figure S6D
Figure 6H	Suppl Figure S6E
Figure 7B	Suppl Figure S6H
Figure 7C	Suppl Figure S6I
Figure 7D	Suppl Figure S7A
Figure 7E	Suppl Figure S7B
Figure 7F	Suppl Figure S7C
Figure 7G	Suppl Figure S7D
Figure 7H	Suppl Figure S7E
Figure 7I	Suppl Figure S7F
Figure 7J	Suppl Figure S7G
Figure 7K	Suppl Figure S7H
Suppl Figure S1F	

The following figures were improved and are marked in GREEN

Figures 2E (previously Figure 7C)	Suppl Figures S2L
Figures 3C, D	Suppl Figure S3A
Figure 4B, D	Suppl Figure S4B
Suppl Figures S1C, D	Suppl Figure S6J

Highlights of Revision

A. mutp53GOF

Many recent comprehensive reviews from the last few years discuss our current knowledge of allele-specific mutp53 GOF effects *in vivo*. Examples (incomplete list) are De Sal G (PMID: 26734571), Vousden K (PMID: 24651012), Lozano G (PMID: 29099488), Prives C (PMID: 22713868), Chan SC & AJ Levine (PMID: 29099487), Sabapathy & DP Lane (PMID: 28948977), Pfister N & Prives C (PMID: 27836911), Stein & Aloni-Grinstein (PMID: 31817996), Wawrzynow & Zylicz (PMID: 29355591), and Yamamoto & Iwakuma (PMID: 30577483).

With regard to the p53 R248 hotspot, our lab and several others previously established the *in vivo* GOF activity of the humanized p53^{R248Q/W} allele in lymphoid and epithelial malignancies in mouse models as well as in familial and sporadic human cancers. Both p53^{R248Q/-} and p53^{R248Q/R248Q} mice have shorter survival than p53^{-/-} mice (*Hanel (Moll) et al. 2013, PMID: 23538418*). Notably, we showed that Li–Fraumeni patients have higher tumor numbers, accelerated tumor onset and shorter tumor-free survival by 10.5 (!) years when harboring codon R248Q mutations as compared to Li–Fraumeni patients with codon G245S mutations or p53 deletions/loss (*Hanel (Moll) et al. 2013*). Other human GOF mutants were identified by the French Li Fraumeni Working Group (*Bougeard et al. 2008; Zerdoumi et al. 2013*).

Notably, subsequent TCGA analysis confirm their prognostic significance also in *sporadic* human cancers: Specifically p53 R248Q/W and R282W alleles exert strong GOF in 6 major human tumor entities (colorectal, ovarian, breast, glioblastoma, bladder and lung). These patients have a 2-fold higher mortality than patients with tumors harboring p53 nonsense alleles (*Xu et al. (Fang) 2014, PMID: 24603336*). We confirmed this sporadic human cancer data in an extended TCGA analysis (*Schulz-Heddergott (Moll) et al. 2018, PMID: 30107178*).

Importantly, genetic ablation of the p53^{R248Q} allele at the time point of clinically advanced p53^{R248Q/-} lymphoma markedly extends mouse survival, indicating tumor dependency on the GOF allele for tumor maintenance and metastasis (*Alexandrova (Moll) et al. 2015, Nature, PMID: 26009011*).

Moreover, the murine p53^{R245W} (equivalent to human p53^{R248W}) allele is sufficient for mammary tumorigenesis in the absence of any other genetic lesions. A few mammary epithelial cells with somatic p53^{R245W} heterozygosity (p53^{R245W/+}) are able to aggressively initiate and drive breast cancer development and metastases after intraductal injection of adenoviral Cre recombinase in a switchable (WT to mutant) breast cancer model. 55% of these breast cancers underwent spontaneous p53LOH and 22% underwent partial p53LOH. Interestingly, the corresponding somatic p53^{R172H} allele is weaker and requires either p53^{R172H/+} plus γ -irradiation or an initial p53^{R172H/-} status to be tumorigenic in this model (*Zhang (Lozano) et al. 2018, Nature Comm*).

We also showed in the AOM/DSS colorectal carcinoma (CRC) model that p53^{R248Q} exerts clear GOF over p53^{-/-}, dramatically promoting invasion and increasing tumor numbers. Mechanistically, stabilized p53 R248Q protein binds to and deregulates pStat3 by displacing its major negative regulator phosphatase SHP2, correlating with poor CRC patient survival.

Genetic ablation of mutp53^{R248Q} reduces growth and invasion of established CRCs. Moreover, Hsp90 inhibition reduces the mutp53 R248Q level and inhibits CRC tumors, again indicating tumor dependency on continuously stabilized mutp53 R248Q protein (*Schulz-Heddergott (Moll) et al. 2018, Cancer Cell, PMID: 30107178*).

B. Newly added data to further support GOF activity of the p53^{R248Q} allele, requested by Reviewers 2 and 3 (new Figures 7 and S7)

1. Murine CRC organoids with induced p53LOH, analyzed after an extended time (20 days after first 4OHT treatment) (new Figures 7B-F and S7A-D). p53LOH de-represses HSF1 activity and triggers GOF mutp53 driven invasion.

To further support our hypothesis that the establishment of GOF after p53LOH takes time, we analyzed p53^{Q/fl}; vilCreERT2 tumor-derived organoids over a longer time span.

- I. After 20 days of 4OHT (inducing p53LOH), we now see
 - clear upregulation of EMT markers Vimentin, Snai1, Snai2 and Zeb1 (new Figure 7C)
 - change to invasive organoid morphology with branchings and protrusions as defined by *Nakayama et al PMID: 32393735, 2017* (new Figure 7D)
 - constant upregulation of HSF1 targets (new Figure 7E)
 - stronger invasion in organoid transwell assays (new Figure 7F)
 - downregulation of p53 targets (new Figure S7C)
 - upregulation of Stat3 target genes, indicating p53^{R248Q}-mediated GOF activity (established in *Schulz-Heddergott et al Cancer Cell 2018*) (new Figure S7D)

2. Chronic stress mimicking the tumor milieu in heterozygous p53^{R248Q/fl} organoids induces spontaneous p53LOH, de-represses HSF1 activity, and triggers mutp53-driven Stat3 programs and invasion (new Figures 7G-I and S7E-G)

To mimic the chronic stress present in tumor milieus which provides selection pressure for spontaneous p53LOH, we treated heterozygous Q/fl organoids for 21 days with Nutlin (new Figure 7G).

After an initial cell death wave due to Nutlin-activated Wtp53 (= black structures with complete loss of integrity of the outer epithelial barrier (new Figure S7E)

- many organoids - most likely those that had undergone p53LOH - are able to re-grow with invasive morphologic structures such as branchings and protrusions (new Figure S7E)
- upregulation of EMT markers (new Figure 7I)
- upregulation of HSF1 targets (new Figures 7H)
- upregulation of Stat3 targets (new Figure S7G)

3. Analysis of KPC mice (Kras; p53^{R172H/+}) tends to show correlation between p53LOH (indicated by p53^{high} staining) and poorer survival (new Figures 7J, K and S7G)

To further support that a stressed tumor milieu enforces p53LOH, we analyzed the KPC (Kras; p53^{R172H/+}) pancreatic cancer model which exhibits aggressive tumor growth. Importantly, this model undergoes high spontaneous p53LOH at a frequency of ~70%, indicated by conversion from undetectable to positive p53 staining (*Olive et al. 2004; Hingorani et al. 2005*).

To this end we used our own KPC cohorts (generated by coauthor Albrecht Neesse, PMID: 28077438 and PMID: 24874484), stained tumor sections for p53 and scored the percentage of positive tumor cells (see Methods). Based on the very strong correlation between stabilized mutp53 (= immunohistochemical p53 positivity) and sequence-verified p53LOH in the *same* tumors (*Hingorani et al. 2005, PMID: 15894267*), p53^{high} cases (approx. 60%) were scored as having undergone p53LOH. p53^{low} cases were scored as heterozygous (no LOH), harboring non-stabilized mutp53.

Of note, due to the high stroma content of KPC PDAC tumors, the alternative technique of qRT-PCR on whole tumor lysates is uninformative. Indeed, we failed to see qRT-PCR based upregulation of p53 and HSF1 targets due to dilution of the tumor epithelial signal by the abundant stroma. Thus, we think that p53 IHC staining is the better choice in KPC tumors.

I. Kaplan-Meier survival of p53LOH versus no LOH

p53^{high} cases (LOH, n=25, median = 133 days) tended to have shorter survival than p53^{low} cases (no LOH, n=15, median = 187 days), log-rank test, p=0.0734 (new Figure 7J).

II. Correlation between p53LOH and high Hsp70 and high phospho-HSF1 staining

We now show that p53^{high} (p53LOH) colocalizes with Hsp70^{high} and pHSF1^{high} staining in tumor epithelial cells. This correlation further supports HSF1 axis-mediated mutp53 stabilization (new Figures 7K and S7H).

REVIEWER COMMENTS

Reviewer #1

Rev1 Using two very good experimental mouse models, the authors first showed that mutp53R248Q is stabilized in colorectal cancer only when Wtp53 is absent. Moreover, only such tumors are invasive. This is essentially consistent with existing knowledge that in human cancers, the presence of mutp53 is associated with advanced stages of the disease, metastasis, recurrence, and patient's poor prognosis, even when compared with TP53 deletion. Then, the authors tested the hypothesis that despite the presence of a GOF allele (R248Q), the remaining Wtp53 allele at least partially retains its transcriptional activity. They assumed that Wtp53 is constitutively activated (e.g. by aberrant growth and metabolic stress, hypoxia, and genomic instability) in tumors and that Nutlin treatment would mimic these conditions.

Crucial to state if Wtp53 is activated by Nutlin in the presence of GOF allele (R248Q) are results shown in Fig. 2C. The degree of activation of known p53 targets (3 from 4 tested genes slightly activated) does not convince that Wtp53 (in p53^{Q/+} tumors) becomes active (very small changes estimated by RT-qPCR normalized to only one reference gene are not reliable).

Answer: The Reviewer justifiably would like to see more p53-activated target genes. To this end, we now greatly extended Figure 2 and performed RNAseq analysis on AOM-DSS tumors from mice comparing the genotypes p53^{+/+} versus p53^{-/+} versus p53^{Q/+} that had been DMSO or Nutlin-treated *in vivo*. We now state that although the Q allele exerts a partial dominant-negative effect (DNE, compare p53^{-/+} versus p53^{Q/+}), DNE is *incomplete*. Importantly, the Wtp53 allele in Nutlin-treated heterozygous mutp53/+ tumors still retains partial activity as indicated by enrichment of the broad Hallmark Wtp53 target gene set encompassing 200 genes (new Figures 2B and Suppl S2B). Moreover, we did qRT-PCR analysis on individual p53 target genes normalized to 2 different reference genes which serve as reliable additional confirmation of the RNAseq data (now Suppl Figure S2C).

In further support of Figure 2, Iyer *et al.* 2016 (now cited as Ref. 53) also shows that CRISPR/Cas9 engineered heterozygous p53^{R248W/+} HCT116 cells (while showing partial DNE) still have sufficient residual wtp53 activity to stop cell cycle after Nutlin treatment, just as we see in our *in vivo* tumor and *ex vivo* organoid models.

Ref 53. Iyer, S.V. *et al.* Allele-specific silencing of mutant p53 attenuates dominant-negative and gain-of-function activities. *Oncotarget* 7, 5401-5415 (2016).

Likewise, concerning HSF1 repression in Nutlin-treated heterozygous tumors, we now show that the residual Wtp53 activity is sufficient to broadly regulate HSF1 target genes in p53^{Q/+} tumors as indicated by RNAseq analysis (new Figure S2D), and confirmed by individual gene analysis via qRT-PCRs (Suppl Figure S2D).

Concerning reference genes used in our qRT-PCRs for Wtp53 activity, we initially tested 4 different ones (*RPLP0/36B4*, *HPRT1*, *GAPDH* and *ACTB*) in our organoid samples (see e.g. in Figures 2E).

Reviewer-only Figure 1. qRT-PCR measurements of reference housekeeping genes in organoid samples from Figure 2D, E. CT values of 4 separate runs are shown. D = DMSO; N = Nutlin.

Importantly, all reference genes showed very similar CT-values for all 4 genotypes/conditions, and thus are not regulated (Reviewer-only Figure 1. Therefore we decided to use RPLP0 and HPRT1 as reference genes in e.g. Figures 2D, E, F and as further mentioned in figures legends.

Rev1_Nevertheless, Fig. S2F shows that one copy of WTp53 can activate its target genes, or rather, that known p53 targets are expressed in p53^{-/+} and p53^{Q/+} tumors.

Answer: We agree. Our results are also in agreement with the literature analyzing heterozygous mutp53/+ tumors cited in the manuscript (*Ghaleb A et al, 2019; Terzian T et al, 2008; Iyer et al, 2016*).

Rev1_The observation that some HSF1 target genes are downregulated after Nutlin treatment (i.e. p53 activation) is really interesting. Hsph1, Hspe1, Hsp90aa1, and Itgb3bp are shown in Fig. 2D as repressed. On the other hand, at least Hspe1, Hsp90aa1 and Itgb3bp basal expression does not depend on WTp53 or mutp53R248Q, although Hspa1a and Hspbp1 expression is elevated in p53^{-/-} and p53^{Q/-} tumors (Fig. S2G).

Answer: The reviewer is correct, the basal expression of Hspa1a (aka Hsp70) and Hspbp1 depends on WTp53. Now we also add DnajA1 (aka Hsp40) to this group of de-repressed HSF1 targets in p53^{Q/-} tumors (improved Figure 2L).

Data in new/ revised Figures 2E, 2F, S2L, 3C, 7E also support that DnaJA1, DnaJB1 or DnaJB2 are downregulated after Nutlin-mediated p53 activation. Data in revised Figures 3D, S3A, 4B, S4B (compare the src2 lanes) and new Figure 7E show downregulation of HspH1.

In general, the non-responder HSF1 targets (Hspe1, Hsp90aa1 and Itgb3bp) might differ from responder targets only by gradation. Non-responders can be viewed in analogy to Mdm2: Mdm2 has 2 promoters, a p53-independent basal promoter responsive to several other transcription factors and generating basal levels of Mdm2 (which is sufficient to keep mutp53 degraded in unstressed tissues), as well as a second p53-responsive promoter in Intron 1 which generates induced levels of Mdm2 for negative feedback degradation of stress-induced p53.

In analogy, we hypothesize that 2 classes of p53-repressed HSF1 targets exist. The first class is very sensitive and already a basal "tonus" of p53 in unstressed / low-stressed tumors suppresses these genes, e.g. HspA1A and HspB1 and DnajA1 (Suppl Figure S2L). The second class has a higher threshold for p53 suppression. Basal condition (with only basal p53 'tonus') does not repress them (e.g. HspE, Hsp90AA1, Itgb3bp) and they do not contribute to LOH pressure under unstressed / low-stress conditions (Suppl Figure S2L). However, once stress is increased (mimicked by Nutlin) this class also becomes repressed (shown in Suppl Figure S2D) and contributes to LOH pressure. Thus, it appears that WTp53

in baseline AOM/DSS tumors suppresses the lower threshold class and requires further stress for the higher threshold class.

Conversely, the reason that mutp53 after LOH is stabilized without basal Hsp90aa1 de-repression is that the Hsp90 chaperone machinery is a multiprotein complex that includes the limiting components Hsp70 and Hsp40 (*Whitesell and Lindquist. 2005, PMID: 16175177; Schopf and Buchner. 2017, PMID: 28429788; Walerych et al., 2009, PMID: 19749793; Wiech et al., 2012, PMID: 23251530; Genest et al., 2019, PMID: 30401745; Dahiya et al., 2019, PMID: 31027879*). Notably, Hsp90 α (*aka Hsp90AA1*) is *the* most abundant cellular protein in mammalian cells even at basal levels, making up 2-3% of the total cellular protein. In contrast, the much less abundant Hsp70 (*aka HspA1a*) and Hsp40 (*aka DnajA1*) are induced after LOH, together causing Hsp40/Hsp70/Hsp90-mediated mutp53 stabilization.

Rev1_At this step, based on shown results, it is difficult to agree with the conclusion (line 169) that “stress-activated Wtp53 prevents chaperone-mediated mutp53 stabilization (Figure 2D)” and especially that “Wtp53 in heterozygous mutp53/+ tumors creates the driving force for p53LOH” (line 171). This is simply not shown here. Such a possibility should be discussed in the Discussion section.

Answer: All the additional experiments in this revision puts our findings on much firmer ground.

Nevertheless, the reviewer justifiably would like us to tone down our conclusions. We completely agree and adjusted the text accordingly throughout Title, Abstract, Results and Discussion. For example, we no longer claim that the Wtp53-HSF1 axis we identified is “*the*” crucial driving force for LOH but “*a*” driving force (meaning one of likely several others still to be discovered). We also now state that the Q allele exerts a partial DNE.

To further support that Wtp53 in heterozygous mutp53/+ tumors creates a driving force for p53LOH, we did additional experiments (see rebuttal pages 3 - 4 for details).

First, to mimic the chronic stress present in tumor milieu that provides selection pressure for spontaneous p53LOH, p53^{Q/+} AOM/DSS tumor-derived organoids were treated with +/- Nutlin for an extended time over 21 days (**new Figures 7G-I and Suppl S7E-G**). Now we see *spontaneous endogenous* p53LOH concomitantly with a pronounced increase of HSF1 target gene expression in re-growing organoids. Due to limited material, we were not able to stain the organoids histologically for mutp53 stabilization.

Second, we added the *in vivo* Kras; mutp53^{R172H/+} driven KPC pancreatic carcinoma (PDAC) model (**new Figures 7J, K and Suppl S7H**). PDAC tumors with stabilized mutp53 are known to have undergone spontaneous endogenous p53LOH (*Hingorani et al. 2005, PMID: 15894267*). We now show that mice with p53LOH tumors have a strong tendency for shorter survival versus mice with no LOH tumors (**new Figure 7J**). Moreover, stabilized mutp53 in these PDAC tumors correlates with upregulation of Hsp70 protein and increased phosphorylation of HSF1 protein (**new Figure 7K and Suppl S7H**).

Rev1_The authors proved that Wtp53 activation by Nutlin is connected with a lower level of HSF1 S326 phosphorylation and lower mRNA level of some HSF1-dependent genes. They claimed that p53 suppresses HSF1 activity via CDKN1A/p21, CDK4/6. Thus, in fact, the cell

cycle inhibition led to the repression of HSF1 activity (measured as S326 phosphorylation and expression of some target genes). It seems that not-typical HSF1 targets (e.g. CDC6, ITGB3BP) are affected stronger than typical ones (HSPs). This suggests that another signaling is highly possible, even bypassing HSF1.

Answer: ITGB3BP and CDC6 were shown by the Lindquist lab, an international leader in HSF1 research, to be non-chaperone HSF1 targets in cancer cells (*Mendillo et al., 2012*, PMID: 22863008). As likely explanation for why they are more strongly affected than typical HSP targets in our scenario, CDC6 and ITGB3BP are *also* regulated by cell cycle pathways, e.g. via E2F/Rb and E2F/DREAM complexes (e.g. *Uxa et al. 2019*, PMID: 31400114). Since Nutlin strongly regulates the cell cycle via p53 targets and therefore also affects the E2F pathways, ITGB3BP and CDC6 expression is most likely *doubly* regulated by both HSF1 *and* E2Fs.

At any rate, to further support our signaling pathway we now added Hsp40 members such as DNAJA1, DNAJB1 or DNAJB2, all canonical HSF1 targets (new in Figures 2E, 2F, S2L, 3C, 7E). Moreover, we added HspH1 (new in Figures 3D, S3A, 4B, S4B, 7E).

Rev1_Since HSF1 can be differentially phosphorylated and activated/deactivated depending on the “stressor” (see e.g. Asano et al.; *Sci Rep.* 2016;6:19174. doi: 10.1038/srep19174. IER5 generates a novel hypo-phosphorylated active form of HSF1 and contributes to tumorigenesis) it could be that S326 is not the best choice to monitor HSF1 activity in this case (although very useful for heat-induced activation).

Answer: It is widely accepted that pSer326-HSF1 is a key activating phosphorylation site and serves as marker of HSF1 activity (e.g. *Gomez-Pastor et al., 2018*, PMID: 28852220; *Dai et al., 2012*, PMID: 22945628; *Mendillo et al., 2012*, PMID: 22863008; *Chou et. Al., 2012*, PMID: 2276810). Still, we agree with the Reviewer that pSer326-HSF1 detection alone is not sufficient and should be functionally confirmed. For this reason we also tested a battery of HSF1 targets and used HSELuc promoter activity assays (e.g. Figures 3C,D, S3A and 3A,B,H, respectively) to be sure that an increase in pSer326-HSF1 indicates active HSF1 in our system.

Reviewer-only Figure 2. IER5 mRNA expression in HCT116 and RKO cells after DMSO or Nutlin treatment. qRT-PCR assay as in Figure 3C.

The cited Asano publication in *Sci Rep.* shows that IER5 is a novel p53 target gene and that ectopic IER5 overexpression in p53-deficient cells (H1299 and 293T, which are p53null or p53-incompetent cells, respectively) leads to a broadly hypo-phosphorylated, yet *active* HSF1 protein.

We confirmed in our hands that IER5 is a novel p53 target. Nutlin induces IER5 expression in wtp53-containing HCT116 cells and to a minor degree also in RKO cells (see **Reviewer-only Figure 2**). Moreover, in our ‘long p53LOH’ organoid experiments, IER5 is down-regulated after LOH (new **Suppl Figure S7C**). However, rather than HSF1 activation, we see *HSF1 repression* across all our systems spanning primary mouse tumors (**Suppl Figures S2D, S2E, S2L**), tumor-derived organoids (**Figures 2E, F and 7E, H**) and human CRC cells (**Figures 3-5**). A likely explanation for the observed discrepancy in HSF1 activity in p53-

proficient versus p53-deficient cells might lie in the presence (this study) or absence (IER5 paper) of a broader p53 response. In WTp53 cells, other p53 and/or E2F/Rb targets likely regulate additional HSF1 phosphorylation sites which in sum might cause HSF1 repression instead of the activation observed in p53-deficient cells.

Rev1_To prove that HSF1 is here a key signaling component, its knockout should be tested.

Answer: Studies by Lindquist and coworkers showed that HSF1 is a major cancer driver in mouse models (*Dai et al., 2005*, PMID: 17889646). Importantly, Jiaqiu Li et al showed that pharmacological inhibition or genetic knockout of HSF1 suppresses colorectal carcinogenesis in the AOM/DSS mouse model, now cited in our revised manuscript (*Li et al., 2018*; PMID: 29730197). Furthermore, we and others previously published that siHSF1 knockdown decreases HSP gene expression and consequently destabilizes HSP90 clients such as mutp53 and MIF (*Schulz et al., 2012*; JExMed; *Schulz et al., 2014 CCDIS*; *Li et al. 2011*, PMID: 21478269; *Li et al., 2014*, PMID: 24763051).

Nevertheless, to further confirm that migration of mutp53 tumor cells depends on HSF1, we used mutp53-harboring SW480 and DLD1 cells, well-known for their invasiveness, silenced HSF1 and performed transwell assays (**Reviewer-only Figure 3**). Interestingly, due to the long half-life of chaperone-stabilized mutp53, we had to silence for 6 days to analyze the effects of mutp53 degradation in transwell assays. Note that after 6 days of siHSF1 mutp53 degradation in DLD1 is stronger than in SW480 (consistent with our previous observation that mutp53 of SW480 is more stable than that of DLD1; *Schulz-Heddergott et al. 2018*). In complete agreement, the degree of migration inhibition correlates with the degree of mutp53 knockdown which in the case of DLD1 is nearly complete. In contrast, SW480 cells show significant but less drastic inhibition, commensurate with their only modestly decreased mutp53 levels. After only 3 days of HSF1 silencing, while HSF1 knockdown is highly efficient, mutp53 levels in both cell lines remain high and correlate with a failure to inhibit migration. Together, these data show that the level of mutp53 determines GOF activities such as migration, but dependent on HSF1.

6 days silencing

3 days silencing

Reviewer-only Figure 3. *left* HSF1 knockdown in human CRC cell lines carrying p53 S241F(DLD1) and p53 R273H/P309S (SW480) missense mutations. At 72 hr after transfection with two different HSF1 siRNAs or scrambled (scr) control siRNA, protein lysates were prepared. Immunoblot analysis. *Top* 6 days of silencing (siRNA addition on day 1 and day 3); *bottom* 3 days silencing (siRNA addition on day 1 only).

right Quantification (right) of transwell migration assay of H1299 cells expressing p53 R248Q treated +/- IL-6. (E and F) Mean \pm SEM of two independent experiments in duplicates, Student's t test.

Rev1_ It is widely accepted that mutp53 is stabilized by chaperones. Obtained data showing a correlation between WTP53 activation and HSPs down-regulation are consistent with the observation that as long as WTP53 is present - mutp53 is not stabilized. According to this observation, the up-regulation of HSF1 targets should be expected in WTP53-deficient tumors. In Fig. S2G, only Hspa1a (known HSF1 target) and Hspbp1 (not necessarily transcriptionally regulated by HSF1) are elevated.

Answer: See our answer above (page 6-7).

Rev1_ From the Abstract: "We find that ... WTP53-retaining murine CRC tumors and tumor-derived organoids and human CRC cells all suppress the tumor-promoting HSF1 transcriptional program." Actually, it was only shown that WTP53 could potentially (e.g. after Nutlin treatment) suppress the expression of some HSF1-dependent genes. And WTP53-retaining murine CRC tumors and tumor-derived organoids and human CRC cells may potentially suppress the tumor-promoting HSF1 transcriptional program. Therefore, although highly probable, I treat the above statement as speculation based on correlations shown in the manuscript and existing literature data.

Answer: We believe that our revised data is now much stronger and shows that

- i) in heterozygous tumors tumor-derived organoids the remaining WTP53 allele partially retains transcriptional activity and represses canonical and non-canonical

HSF1 target genes such as HspH1, Hsp90AA1 and HspE1 after stress-induced p53 activation (Figures 2E, F, S2D, S2E, S2J and new Figures 7E,H)

- II) In addition, p53 activation suppresses HSF1 reporter activity in HSE/luc assays and pSer326-HSF1 phosphorylation, together indicating *broad* repression of the HSF1 transcriptional program in wtp53 cells (Figures 3-5)
- III) after p53LOH, mutp53 protein levels become stabilized *in vivo* (Figures 1F,I and Suppl 1B,G) and in organoids (Figure 2G)

To further strengthen our conclusions that the HSF1 transcriptional program is regulated by WTP53, we newly added

1. Additional HSF1 targets from the DnaJ (Hsp40) family and the Hsp90 family, measured in tumors, organoids and cell lines (e.g. improved Figures 2E, S2L, 3C,D, new Figures 2F and 7E).
2. The Hallmark p53 response in Nutlin-treated tumor samples analyzed by RNAseq shows a broad residual WTP53 activity in heterozygous tumors (new Figures 2B and S2B) with downregulation of HSF1 target genes (new Figure S2E and Suppl Figure S2D).
3. Additional p53 activators Doxorubicin and 5-FU also suppress HSF1 target gene expression in heterozygous p53^{Q/fl} organoid cultures (new Figure 2F).

Rev1_ Repression of MLK3 was shown in cell lines, not in tumors. Why it creates selection pressure for LOH?

Answer: We expanded our analysis and now show that MLK3 is also repressed in heterozygous tumor-derived organoids (new Figure 5H) after Nutlin and after Dox and 5-FU (new Figure 5I). Instead of tumors which contain a lot of confounding stroma that could impact interpretation, we analyzed the axis in the pure epithelial system of organoids. After 24 hrs of Nutlin or genotoxic treatment MLK3 expression is suppressed in heterozygous organoids (new Figures 5H,I). Conversely, after long-term Nutlin-induced spontaneous p53LOH, MLK3 is upregulated due to loss of the WTP53 allele (new Figures 7G, H). MLK3 as E2F target is repressed by Nutlin or genotoxics and consequently the MLK3-MAPK-HSF1 axis is down-regulated. This creates selective pressure for LOH in heterozygous cells.

Rev1_ There are many serious and minor errors in the manuscript, some of which (and additional comments) are listed below.

1. Fig. 1G – check the description: the left and right graphs show contradictory results. The wrong legend.

Answer: Now corrected, thank you, we had erroneously mixed up the legends in old Figure 1G (now Figure 1H). Both panels ‘total invasive tumors’ and ‘total mouse numbers with invasive/non-invasive tumors’ show that invasion is enabled after p53LOH. We added mouse and tumor numbers to the left graph.

2. There is no information about how effective is WTP53 removal after Tamoxifen treatment in the experiment shown in Fig. 1. On the other hand, it is not so important since these mice are used later only for organoid culture and here the efficiency of recombination is shown.

Answer: Still, to address this concern *in vivo*, we now provide a recombination-specific genotyping PCR to show recombination efficiency in TAM-induced tumors (new Figure S1F).

3. Line 159: Fig. 2C; “We conclude that, surprisingly, the GOF mutp53R248Q allele fails to exert a dominant-negative effect over the remaining WTP53 allele as predicted by many, mainly *in vitro*, studies.” Based on results shown in Fig. 2C it is difficult to agree with such statement.

and Line 333: “we find in our CRC model that the remaining WT allele in heterozygous tumors is fully activatable, excluding a dominant-negative effect (**DNE**) by the counterpart mutp53 allele”.

Presented data does not support this conclusion. p53Q/+ tumors should be compared to p53-/+ tumors. Such a comparison is shown only in Fig. S2F. Assuming that RT-qPCR results are reliable and Cdkn1a is regulated mainly by WTP53 (as indicated by lack or very low level of expression in p53-/- and p53Q/- tumors; although it is still detected in p53Q/- tumors and organoids after TAM/4OHT-induced CRE/loxP recombination) its expression is lower in p53Q/+ tumors than in p53-/+ and p53+/+ tumors.

Answer: The Reviewer is absolutely correct, we do have a *partial DN* effect by qRT-PCR (for Cdkn1a and Bbc3, now Figure S2K) and by RNAseq analysis (new Figures 2B). Importantly, however, despite a partial dominant-negative effect by the Q allele, we clearly identify that the remaining WTP53 allele in p53^{R248Q/+} tumors is *still partially and broadly active* and represses the HSF1 chaperone axis, thereby preventing mutp53^{R248Q} stabilization, GOF and invasion. We corrected the text accordingly in several places.

4. Fig. 2C, 7B, and line 156 – mistakes in description: Cdnk1a instead Cdkn1a

Answer: Corrected, thank you.

5. Supp Fig. 2G – HspA1A mRNA level is shown. There is no primers (mouse) for this gene in Table S2.

Answer: The primer sequences are now added to Table S2.

6. Line 189: “Nutlin also suppressed the tumor-promoting HSF1 targets CDC6, ITGB3BP, RBBP5, BST2 and FBLN1 (Figure 3C)”. BST2 is not suppressed.

Answer: Correct, BST2 is a special case. Now replaced by the word ‘regulated’ instead of ‘suppressed’. BST2 is a *known repressed* HSF1 target and therefore up-regulated after Nutlin.

7. Fig. 3E: “Activated WTP53 suppresses pSer326-HSF1”. Correlation is only shown, not HSF1 suppression by WTP53.

Answer: We now corrected the wording to “Activated WTP53 is *correlated* with suppression of pSer326-HSF1.....”.

8. Line 206-207: "Nutlin strongly dephosphorylated pSer326-HSF1 (Figure 3G)" Correlation is only shown, not HSF1 dephosphorylation by Nutlin.

Answer: Correct, now reworded accordingly.

9. "HSF1 binding to HSE-Luc reporter was measured". Activity of Luc was measured, not HSF1 binding.

Answer: Correct, this wording was removed.

10. Line 632: Supp Fig. 5: "(A, B) Depletion of CDK1 (A) and CDK2 (B) fail to abrogate HSF1 activity." Only HSF1 phosphorylation at S326 was shown, not HSF1 activity.

Answer: Correct, now corrected with "...pSer326-HSF1 phosphorylation" (legend Figure S5A, B).

11. Suppl Fig. 3F – HSPAA1 ? Probably HSP90AA1.

Answer: Corrected, now 'HSP90AA1.'

12. Fig. 4F – two lower blots do not fit to the upper ones.

Reviewer-only Figure 4. RG7388 treatment of HCT116 cells. Raw immunoblot of Figure 4F after pRB (top) and actin (bottom) staining. Sequential stainings (with stripping in between) was done on the same membrane cut into 2 parts.

Answer: We went back to the original experiment and carefully re-checked that it was done correctly. We verified that all immunoblot staining in Figure 4F were done sequentially *on the same membrane* that had been cut into an upper and a lower part (see black line). The upper part was first stained for pHSF1, followed by pRB. The lower part was first stained for p53, followed by actin. **Reviewer-only Figure 4** shows as example the raw films with molecular weight markers on both sides after pRB (strong upper band in lane 2) and after actin (lower bands across all lanes). Since the upper membrane is after pRB staining, pHSF1 bands are no longer visible. The lower membrane is after actin staining, thus p53 is only barely visible (see very faint bands in lanes 3-6 above actin).

13. Fig. 5F and G – please check GAPDH which seems to be the same on both blots.

Answer: Correct, thank you, we accidentally used the same GAPDH twice (immunoblots and processing of Figures 5F,G were done on the same day). We now corrected Figure 5F with the correct corresponding GAPDH loading control from HCT116 cells.

14. Line 248 - pSer-325 – should be 326.

Answer: Corrected.

15. Line 656: Fig. 6B: "The mean survival of WTp53 patients (n=214) is 83.2 month versus 57.2 months for patients with MS plus p53LOH 658 (mutp53/-)". The description is not consistent with the Figure which indicates that WT p53 patients have shorter survival time. The wrong legend.

Answer: The reviewer is correct. We accidentally switched the line colors red/green. Now replaced with the correct color code.

16. Fig. 6C and S6A-C: Hierarchical clustering should be performed in such analyses.

Answer: The reviewer suggested to use hierarchical clustering. In data mining hierarchical clustering is a particular exploratory statistical method in unsupervised learning with the goal to discover an *unknown* internal structure intrinsic to the data. However, this was not our goal here since the preceding Figures 1-5 already had established the link between WTP53 status and HSF1 axis suppression. Thus, the goal in Figure 6 was to test whether this correlation that we had established in mouse tumors and organoids and in human CRC cells also holds for the human COADREAD tumor data set (TCGA). To this end we grouped patients into p53 status (WTP53 *versus* mutant+LOH p53) and using the published authoritative list of HSF1 target genes (*Mendillo et al., 2012, PMID: 22863008*), we asked the algorithm to rank order the HSF1 genes by fold-change expression (heatmaps in Figures 6C and now Suppl S6A,B,C,F,,G,H and I). In this analysis, the mutant+LOH p53 cases indeed exhibit mostly upregulation of the HSF1 program.

Upon the Reviewer's request we now used the *same* COADREAD data set and the *same* gene list to generate heatmaps with hierarchical clustering, either by patients alone (**top Reviewer-only Figure 5**, dendrogram in columns) or by patients & genes (**bottom Reviewer-only Figure 5**, dendrograms in rows and columns). Importantly, this analysis gives *exactly the same result* as Figure 6C (and by extension as Suppl Figures S6A, D, E) with separation along the red and green mutational p53 status (top bars), but is hard to visualize because of the different ordering. E.g., the few dark blue patient samples in the left part of the heatmaps are very prominent yet distract from the information contained. The fact that they cluster is likely due to variables beyond the p53 status that also bear on HSF1 target gene expression, which is to be expected due to tumor heterogeneity. If we re-order the patients in Reviewer-only Figure 7 by grouping them by p53 status into green versus red groups, we arrive at our Figure 6C and Suppl Figures S6A, D, E.

Reviewer-only Figure 5. The same COADREAD data set and gene list as in Figure 6C was used to generate heatmaps with hierarchical clustering, either by patients alone (top) or by patients & genes (bottom). Importantly, this type of analysis gives exactly the same result as Figure 6C (and by extension Figures S6A, D, E) with separation between red and green mutational p53 status, but it is very hard to visualize because of the different ordering.

Red/blue Row Z-Score – not properly labelled (-1, -2, 0, 1, 2).

Answer: Thank you, we corrected the Z-score labeling.

17. Why it was stated that HSF1 negatively regulates a subset of target genes (line 667). Only correlation between p53 status and the level of expression of the potential HSF1 target genes in colorectal adenocarcinoma patients is shown.

Answer: We deleted this sentence to avoid confusion.

18. Line 275: “BRCA cells repressed pSer326-HSF1 when harboring WTp53 but not mutp53 (Figure S6D)”. Based on blots shown in S6D it can be concluded that Nutlin does not activate p53 in MDAMB-231 (its level is stable). It looks that there is some technical problem with pHSF1 blot from MCF7. Thus, the conclusion (line 275) is not well supported.

Answer: We repeated this immunoblot which confirmed our original result, excluding a technical problem (revised Suppl Figure S6J).

19. 1124-1125: “Densitometric measurements for quantification of immunoblot bands were done with the gel analysis software Image Lab™ (BioRad) and normalized to loading controls.”

Such analyses are not present in the manuscript.

Answer: We now performed densitometric measurements for pSer326-HSF1. (See also Reviewer 2, minor point 1).

20. I would like to know what kind of statistics was applied to calculate statistical significance in qPCR when “mean \pm SEM of 2 independent experiments, each repeated twice in triplicates” is shown.

Answer: We used Student t-tests with *at least* 4 qRT-PCR runs per sample. Briefly, we measured *at least* two biological replicates. From every biological replicate we did 2 independent cDNA syntheses followed by qRT-PCR of each (i.e. 2 technical replicates of each biological replicate), yielding $2 \times 2 = 4$ (*or > 4*) total qRT-PCR runs. Most technical samples were pipetted in triplicates on the 96-well plate. For clarification we changed the wording in the Method section to: “Mean \pm SEM of ≥ 2 independent experiments, each with two technical replicates, pipetted mostly in triplicates but at least in duplicates.”

Reviewer #2

In this paper Sener et al investigate the mechanism of P53 LOH in colorectal cancer. Using a number of animal and cell culture models they propose that loss of wt P53 in R248Q tumours is required for HSF1 activation which in turn promotes the gain of function effects of R248Q mutations. This is an interesting study which I think casts some light on the mechanisms and importance of P53 LOH. The data mostly appears robust and the conclusions around the function of P53 in suppressing HSF1 activation and P53 LOH appear sound. However, a number of further analyses are required to convince that R248Q has gain of function properties particularly in light of recent findings that P53 point mutations do not confer GOF properties (S. Boettcher et al., Science 365, 599 (2019)). I think that a number of additional experiments are needed to address this point or, if not possible, the language on GOF should be toned down throughout the manuscript:

Answer: In the delineation of DNE versus GOF for missense mutant p53 alleles, Boettcher et al concluded that in the *myeloid* tumor entity which they studied there is *complete* DNE, no GOF. However, numerous studies show that DNE and GOF are both context dependent.

The Boettcher paper - which we cite in the Discussion [Ref 72, now 71] - is a very systematic and important study because the small subgroup of AML patients who do harbor p53 mutations (less than 10 % of all AML patients) have a horrifically poor prognosis, far worse than the majority of AML patients who have WTP53 status. As per cBioportal datasets, myeloid leukemias have a very low p53 mutation frequency (only 8.6 % of cases have MS/NS/FS p53 mutations). Of those 8%, only half show p53LOH (TCGA Acute Myeloid leukemia study, NEJM 2013, 187 total patients; 20 patients with p53 mutations, 11 patients of those have MS mutations, and of those 5/11 show shallow deletions = p53LOH). Thus, given that 92% of AMLs are WTP53, the pressure for p53LOH in AML is far lower.

In contrast, p53LOH pressure is huge in most epithelial-derived solid cancers including CRC which harbor frequent p53 mutations (of which 72% are missense) with an extremely high p53 MUTATION/LOH rate of > 91 %. This was definitively shown in a huge comprehensive study of 5 data platforms on 10,225 human TCGA cancers (*Donehower et al., 2019, PMID: 31365877*). The fact that 91% do undergo LOH strongly suggests if not proves that in the vast majority of heterozygous solid cancers the DNE is only partial and weak, leaving behind significant residual wtp53 activity. This provides the selection pressure for LOH. Because if there were putative strong complete DNE, there would be no selection pressure for losing the WT allele and LOH would not be observed.

Furthermore, in a 2020 review (p53 tetramerization: at the center of the dominant-negative effect of mutant p53, PMID: 32873579) by Guillermina Lozano, another leading scientist in the p53 field, she talks about the mutant p53 dominant-negative effect (DNE) as “ *an elusive phenotype* ” and emphasizes that p53 tetramerization is at the center of the DNE of mutant p53. She states that “ p53 mutants with a functional tetramerization domain form mixed tetramers, which *in some cases* have DNE that inactivate wild-type p53. Posttranslational modifications and protein–protein interactions alter p53 tetramerization affect transcription, stability, and localization. These regulatory components *limit the DNE* of mutant p53 on wild-type p53 activity. ”

Here we would like to point out a key figure in Boettcher et al. which is important in the context of our paper. In Boettcher Figure 3A (= Reviewer-only Figure 6), R248Q protein levels in *untreated* (-) heterozygous R248Q/+ TP53 isogenic AML cells are already very high even at baseline, comparable to our homozygous R248Q/- levels in CRC, thus indicating constitutive stabilization of mutp53 protein in heterozygous AML. Importantly, in our heterozygous CRC tumors we *never* see mutp53 protein stabilization (Figures 1F, S1B and 2G). This indicates that in heterozygous epithelial-derived carcinomas the MUT/WT protein ratio does *not* constitutively undergo the shift in favor of stable MUT. Hence, even after Nutlin activation we only see a *partial* DNE, leaving sufficient WTp53 activity intact to suppress the HSF1/Hsp90 axis and exert LOH pressure. Conversely, heterozygous AML already appears to have a constitutive ratio shift favoring stable MUT protein, which helps explaining the strong and exclusive DNE Boettcher et al observed in AML cells.

[REDACTED]

We can only speculate why myeloid malignancies *constitutively* have this shift and stabilize mutp53. They are driven by receptor mutations such as FLT3 and cKIT to trigger MAPK such as MEK1/2 which are known to *strongly* activate HSF1. Such strong receptor based MEK activation might readily overwhelm the WTp53-HSF1 repression axis. In other words, in myeloid malignancies the proliferative signals from mutant FLT3/cKIT to activate MEK-HSF1 are much stronger than the stress activated WTp53 signals that repress MEK and HSF1. Future studies looking to correlate which stress type is stronger might shed light on the context dependency for different tumor types.

The importance of MUT/WT protein stoichiometry was also emphasized in a recent commentary by David P Lane (a founding father and thought leader in the p53 field) of the Boettcher 2019 paper (PMID: 31395768). David Lane writes: “ Extensive studies have implied that the mutant p53 proteins may act directly as oncoproteins not only through DNEs on wild- type p53 but also through GOF, such as the interaction with new protein targets, and that this contributes greatly to the importance of p53 in human cancer. Such neomorphic functions would represent important targets for cancer therapy (he cites our paper *Schulz-Heddergott 2018*). These studies are based on a number of different experimental approaches, but perhaps the most compelling are mouse studies in which the mutant p53 protein is genetically deleted from growing tumor cells, resulting in inhibition of tumor growth (he cites our paper *Schulz-Heddergott 2018*). In the study by Boettcher no evidence for GOF is seen either in cell-based assays or in analysis of clinical samples. Reconciling these

different findings is not straightforward but *strongly implies that GOF must exert itself specifically in epithelial malignancies (and not myeloid malignancies), consistent with its role in promoting invasion and metastasis.*"

Moreover, in their discussion Boettcher et al. also considered GOF to be context dependent and present in solid cancers but not in myeloid cancer because of the differential presence or absence of transcription factors such as p63, NF-Y, NRF2 and ETS2 as biochemical complexing partners of mutp53 GOF. We speculate that maybe in the p53^{R248Q/+} DNE context with mixed heterotetramers, p53^{R248Q} is not able to bind to tumor drivers such as STAT3, ETS2, SREBP1/2, VDR etc, in contrast to homozygous 'all-mutant' homotetramers in the p53^{R248Q/-} scenario.

Major points

Rev2_The overall tone of the paper is that R248Q confers a GOF phenotype when it is stabilised in the context of loss of WT P53. This is based on prior literature from the same lab and also on the experiments carried out in Fig 1 and S1.

Answer: In general regarding mutp53 GOF, many recent comprehensive reviews from the last few years discuss our current knowledge of allele-specific mutp53 GOF effects *in vivo*. Examples (incomplete list) are De Sal G (PMID: 26734571), Vousden K (PMID: 24651012), Lozano G (PMID: 29099488), Prives C (PMID: 22713868), Chan SC & AJ Levine (PMID: 29099487), Sabapathy & DP Lane (PMID: 28948977), Pfister N & Prives C (PMID: 27836911), Stein & Aloni-Grinstein (PMID: 31817996), Wawrzynow & Zyllicz (PMID: 29355591), and Yamamoto & Iwakuma (PMID: 30577483).

With regard to the p53 R248 hotspot, our lab and several others previously established the *in vivo* GOF activity of the humanized p53^{R248Q/W} allele in lymphoid and epithelial malignancies in mouse models as well as in familial and sporadic human cancers. Both p53^{R248Q/-} and p53^{R248Q/R248Q} mice have shorter survival than p53^{-/-} mice (*Hanel (Moll) et al. 2013, PMID: 23538418*). Notably, we showed that Li–Fraumeni patients have higher tumor numbers, accelerated tumor onset and shorter tumor-free survival by 10.5 (!) years when harboring codon R248Q mutations as compared to Li–Fraumeni patients with codon G245S mutations or p53 deletions/loss (*Hanel (Moll) et al. 2013*). Other human GOF mutants were identified by the French Li Fraumeni Working Group (*Bougeard et al. 2008; Zerdoumi et al. 2013*).

Notably, subsequent TCGA analysis confirm their prognostic significance also in *sporadic* human cancers: Specifically p53 R248Q/W and R282W alleles exert strong GOF in 6 major human tumor entities (colorectal, ovarian, breast, glioblastoma, bladder and lung). These patients have a 2-fold higher mortality than patients with tumors harboring p53 nonsense alleles (*Xu et al. (Fang) 2014, PMID: 24603336*). We confirmed this sporadic human cancer data in an extended TCGA analysis (*Schulz-Heddergott (Moll) et al. 2018, PMID: 30107178*).

Importantly, genetic ablation of the p53^{R248Q} allele at the time point of clinically advanced p53^{R248Q/-} lymphoma markedly extends mouse survival, indicating tumor dependency on the GOF allele for tumor maintenance and metastasis (*Alexandrova (Moll) et al. 2015, Nature, PMID: 26009011*).

Moreover, the murine p53^{R245W} (equivalent to human p53^{R248W}) allele is sufficient for mammary tumorigenesis in the absence of any other genetic lesions. A few mammary epithelial cells with somatic p53^{R245W} heterozygosity (p53^{R245W/+}) are able to aggressively initiate and drive breast cancer development and metastases after intraductal injection of adenoviral Cre recombinase in a switchable (WT to mutant) breast cancer model. 55% of these breast cancers underwent spontaneous p53LOH and 22% underwent partial p53LOH. Interestingly, the corresponding somatic p53^{R172H} allele is weaker and requires either p53^{R172H/+} plus γ -irradiation or an initial p53^{R172H/-} status to be tumorigenic in this model (*Zhang (Lozano) et al. 2018, Nature Comm*).

We also showed in the AOM/DSS colorectal carcinoma (CRC) model that p53^{R248Q} exerts clear GOF over p53^{-/-}, dramatically promoting invasion and increasing tumor numbers. Mechanistically, stabilized p53 R248Q protein binds to and deregulates pStat3 by displacing its major negative regulator phosphatase SHP2, correlating with poor CRC patient survival. Genetic ablation of mutp53^{R248Q} reduces growth and invasion of established CRCs. Moreover, Hsp90 inhibition reduces the mutp53 R248Q level and inhibits CRC tumors, again indicating tumor dependency on continuously stabilized mutp53 R248Q protein (*Schulz-Heddergott (Moll) et al. 2018, Cancer Cell, PMID: 30107178*).

Rev2_It would be helpful to expand on the previous paper and strengthen these arguments for this paper.

Answer: The Reviewer has an excellent point, thank you, we indeed had not mentioned our own background work in this manuscript. For CRC, we previously extensively analyzed the p53^{R248Q} GOF function in the AOM/DSS mouse model and showed that p53^{R248Q} GOF mutants bind to and hyperactivate pSTAT3, driving tumor invasion (*Schulz-Heddergott et al., 2018*). This and an earlier paper from our lab (*Alexandrova et al., 2005. Nature, PMID: 26009011*) are now included in the Introduction.

In the non-induced constitutive p53LOH model (= 100% of cells have LOH, Suppl Figure S1), we again showed p53^{R248Q} GOF by increased tumor numbers (Suppl Figure S1C) and increased invasion (Suppl Figures S1D,E) in AOM/DSS CRC. Moreover, in Suppl Figure S2J proliferative genes (= GOF genes) were upregulated in constitutive p53^{Q/-} compared to p53^{-/-} tumors (PCNA, Ccnb1 and Ccnd1). We would like to point out that increased invasion is already a functional read-out for GOF, hardly seen in p53^{-/-} C57BL6 mice (Suppl Figure S1D left see 'LOH' group, LOF p53^{-/-} 20% vs GOF p53^{Q/-} 70%). This is now better explained in this Figure.

Rev2_For example, the experiments in Fig 1 should include P53fl/fl mice as a control.

Answer: The reason why we had not included p53^{fl/fl}; vilCreERT2 organoids in the initial design of this study to address GOF was that we previously already extensively addressed the R248Q/W GOF *in vivo* in our AOM/DSS CRC mouse model (*Schulz-Heddergott (Moll) et al Cancer Cell 2018*).

Unfortunately, it was not possible now to extend our *in vivo* study and include the p53^{fl/fl} mice because it would have taken an estimated additional > 1.5 years, surpassing any reasonable time limit for revision. During the COVID pandemic the formal approval process to renew our

IACUC license (which had expired in March 2020; and Renewals are very bureaucratic and not issued by our University of Goettingen but by the German State of Lower Saxony's Health Department) in order to add additional mice, plus the new breedings for the required mouse numbers and the actual experiments with evaluation would have been extremely lengthy. So we apologize for this omission.

Rev2_However, an alternative explanation could be that R248Q mutation inactivates P53 and LOH of WT P53 leads to essentially a null allele.

Answer: From all we know about the p53^{R248} allele (see above) it is highly unlikely that the p53^{R248Q} + LOH leads to a p53null situation in CRC. Importantly, the strong increase in invasiveness (Figure 1H) is not seen in our p53null mice with a C57BL6 background (improved Suppl Figure S1D). Just 20% of p53^{-/-} mice under AOM/DSS developed invasive tumors in the C57BL6 background, whereas 70% of p53^{Q/-} mice developed at least one invasive tumor. With inducible p53LOH *all* mice developed at least one invasive tumor after 6-8 weeks. Thus, in this system Q is a GOF and not a LOF allele because we see an increase in invasion from 20% (p53^{-/-}) to 70% (p53^{Q/-}) (Suppl Figure S1D).

Importantly, the second main focus of this study is de-repression of the HSF1 axis by p53LOH which occurs independently whether the Q allele acts as LOF or GOF (see e.g. Figures 6 and S6).

Rev2_.... tumours should be harvested ... (similar to Fig 1E) and compared by RNAseq/histology etc for changes that would be consistent with R248Q conferring gain of function effects and not acting solely as a loss of function allele.

Answer: In the inducible p53LOH system with its heterogeneous response after TAM treatment and its tumor stroma (Figures 1F,I), we believe that histology is the technique of choice because when gene expression is measured on whole tumor lysates by qRT-PCR, mild responses of single genes might be diluted out and lost. We therefore used tumors 6-8 wks post-TAM (Figure 1D) and stained serial sections side-by-side (same batch) for CyclinD1 and p53. As seen in new Suppl Figure S1G, areas with stabilized p53 correlate with CyclinD1-positive (upregulated) proliferative areas.

As stated above, GOF of R248Q *in vivo* was extensively shown in *Schulz-Heddergott Cancer Cell 2018*. To further strengthen this point in the current paper we analyzed p53^{Q/fl}; vilCreERT2 organoids after long-term (20 days and 3 weekly pulses of 4OHT, new Figure 7B) induction of p53LOH in transwell and branching assays and by expression analysis. We confirmed that mutp53 GOF activities take time to manifest. Invasive EMT marker expression is increased after long-term p53LOH (new Figure 7C, compare to short term in Figure 7A). So is the invasive GOF branching phenotype (new Figure 7D) and invasive migration across a membrane (new Figure 7F). Finally, expression of Stat3 target genes mediating GOF are up-regulated after long-term induced p53LOH (new Figure S7D).

Rev2_The authors should also compare the tumour burden and invasiveness in their model S1C and D between P53^{-/-} and P53^{-/-}/R248Q mice, are these changes significant? Previous reports (ie Schwitalla et al., Cancer Cell) have shown that loss of P53 is sufficient to confer an invasive (and metastatic) colorectal tumour phenotype so losing P53 activity is

clearly sufficient for this without gain of function mechanisms. *The authors also find evidence of invasive tumours in their P53^{-/-} mice although it does look more prevalent in the R248Q mice.* Again, more analyses of tumours from this model would help convince of the GOF effects of R248Q (for example RNAseq and histological analysis of non invasive tumours from both models (to ensure like for like comparisons).

Answer: As shown in Suppl Figure S1C, tumor numbers per mouse are not significantly different (although there is a trend) due to low effect size. In contrast, invasiveness, a much more robust GOF phenotype, is significant in our model. We now include these calculations (improved Suppl Figure S1D).

The senior author of the Schwitalla et al. paper the Reviewer cites, Prof. Florian Greten, is also co-author on our *Schulz-Heddergott 2018* paper. Thus, we are well aware that p53null mice develop invasive CRC tumors - but importantly – this occurs largely on a BalbC background used by Schwitalla et al., thus it is mouse strain-dependent. BalbC mice are much more CRC -prone than the C57BL6 mice that we use for all our studies (Suzuki et al., 2006, PMID: 16081511). In C57BL6 mice, as explained above, p53^{-/-} mice show a low overall incidence for invasion (20%), while 70% of Q^{-/-} mice are invasive.

Rev2_The human data in figure 6 also isn't particularly convincing and doesn't strengthen the case for a GOF effect of R248Q. What does survival look like if comparing truncating, stop gain LOH tumours? *It would be expected that if missense mutations confer GOF then these should have worse prognosis than truncating mutations.*

Answer: Note that *not* all missense mutant p53 alleles acquire GOF since GOF activities are allele- and context-dependent (Prives C, PMID: 22713868; Pfister & Prives, PMID: 27836911). Importantly, for the hotspot p53^{R248Q/W} allele TCGA analyses by Xu & Fang (PMID: 24603336) showed prognostic significance of p53^{R248Q/W} (and p53^{R282W}) patients in 6 major sporadic tumor entities (colorectal, ovarian, breast, glioblastoma, bladder and lung). These patients have a 2-fold higher mortality than patients with nonsense alleles. Subsequently, we showed a worse prognosis with poorer patient survival for p53^{R248Q/W} in stomach cancer, esophageal cancer and head & neck cancer (Figures 7I, 7J and Suppl Fig 7I in *Schulz-Heddergott et al., 2018*) and in sporadic pancreatic ductal adenocarcinoma (cBioportal TCGA data, unpubl. Schulz-Heddergott et al 2020),

Please note that in order to specifically analyze CRC patients with R248Q/W mutations, even the most current TCGA database does not contain sufficient case numbers for statistical analysis. Indeed however, and as correctly predicted by the Reviewer, an analysis with all available CRC cases (COADREAD) comparing FS/NS+LOH (n=45) versus MS^{hotspots} + LOH (n=84) show a trend (p=0.11) towards shorter survival of patients with MS^{hotspots} (defined as p53 R175|R273|R248|R282|G245|R249) compared to frameshift/nonsense mutations. This graph is now shown as Reviewer-only Figure 7, but can be added to the manuscript if the Reviewer wishes.

Rev2 Minor points

1) There is some data quantification lacking. Throughout the manuscript, Western blots are shown with no quantification of the data. This need to be included throughout.

Answer: We now quantified all our immunoblots for pHSF1 and MLK3 (see extended/new Figures 3E-G, I, S3B-E, 4A,C,E-H, S4C,D, 5B,E-G, S5A,B,D,F, S6J).

2) Figure 4 isn't particularly convincing. siRNAs should be used to validate the role of cdk2/4 particularly as the authors outline the lack of specificity of the inhibitors used.

Answer: A role for CDK1 and CDK2 in pSer326-HSF1 phosphorylation was excluded in Suppl Figures S5A-B. Instead we showed a role for CDK4 and CDK6 (Figure 4). As requested by the Reviewer, we now silenced CDK4 and CDK6 with siRNAs in wtp53 harboring HCT116 cells (new Figures S4C, S4D). Indeed, combinatorial silencing of CDK4 or CDK6 reduced pHSF1 (and pRB) levels (new Figure S4C), thereby phenocopying CDK4/6 inhibition by Palbociclib, albeit with lower efficiency. Not surprisingly, the small-molecule inhibitor Palbociclib, widely used in clinically oncology, has a stronger effect in inhibiting the CDK4/6 complex than siRNAs (new Figure S4D). Single knockdown had no consistent effect.

New Suppl Figure S4C, S4D. **A** pHSF1 reduction by CDK4/6 silencing. The Wtp53 harboring CRC cell line HCT116 was treated with scrambled siRNA or different siRNAs against CDK6 alone, CDK4 alone or a combination of both for 72 hrs. Top and bottom are independent repeats of the experiment. **B** Reduction in pRB as functional control for cell cycle inhibition. Palbociclib (Palbo, 10 μ M) was used for 24 hrs. Cell cycle inhibition was confirmed by Rb de(hypo)phosphorylation. Immunoblot analyses. Actin, loading control.

3) To strengthen the link to MAPK signaling, inhibitors of this pathway (ie MEK inhibitors) should be used and effects on HSF1 stress response measured.

Answer: Good point, thank you. As requested by the reviewer, we now performed this experiment with the well-characterized MEK inhibitor U0126 and confirmed the known MEK-HSF1 axis in human CRC cell lines (new Suppl Figure S5F) as well as in heterozygous organoids (new Figure 5J). Moreover, we also showed that MLK3 as MEK regulator is down-regulated after p53 activation by chemotherapeutics (new Figure 5I).

4) There is a lack of evidence from the in vivo model to support the in vitro mechanistic data. IHC analysis of the Nutlin treated tumour model for components of the downstream altered pathways should be carried out.

Answer: Good point, thank you. Downstream components of the altered pathway are e.g. pRB, MLK3 and pMEK. We stained DMSO/Nutlin-treated p53^{Q/+} tumors for pHSF1 and pRB by immunohistochemistry and indeed see reduced phosphorylation of both proteins (new Figure 4I).

We also attempted to histologically stain tumors for MLK3 but failed due to insufficient antibody quality which produced very high background staining (we tried several different antibodies). As work-around we used heterozygous tumor-derived organoids and performed qRT-PCR for MLK3. We confirmed the reduction of MLK3 mRNA expression after Nutlin (new Figure 5H) as well as after Dox and 5-FU (new Figure 5I). Moreover, MLK3 is up-regulated after long-term p53LOH induction (new Figures 7E, H).

Furthermore, we treated heterozygous organoids with the MEKi U0126 and showed that HSF1 targets are downregulated (new Figure 5J).

5) Fig 7F. The authors propose that following LOH, upregulation of EMT genes and pro-invasive HSF1 target genes like HspH1 and Itgb3bp enable invasiveness in a stressed tumor milieu following p53LOH. This statement isn't well supported by the data and should be toned down.

Answer: The Reviewer is correct. Our initial attempt to see upregulation of EMT genes failed because GOF takes time and that is the reason why we did not see EMT upregulation at only 4 days after Nutlin (see old Figure 7F, now Figure 7A).

We now did new experiments with 'long-term' p53LOH and analyzed tumor-derived organoids for EMT markers and HSF1 targets 20 days after 4OHT-induced p53LOH. Indeed, we now see a strong upregulation of both groups of readouts (new Figures 7C and 7E). Moreover, transwell assays and induced morphological branching of organoids confirmed an increase in the invasive phenotype after p53LOH (new Figures 7D and 7F). Finally, extended Nutlin stress in p53^{Q/m} organoids over 21 days induces spontaneous p53LOH accompanied by upregulation of EMT markers and HSF1 targets (new Figures 7G-I).

Rev2_Overall, I think this is an *interesting paper* and provides some mechanistic insight into the potential role of P53 LOH in colorectal cancer. However, *at the moment*, I don't think the data supports statements such as ' Our data provide an explanation for the longstanding puzzle why tumor heterozygosity tends to be unstable and why p53LOH strictly correlates with mutp53 stabilization and higher tumor aggressiveness.' *Further data is needed to support statements like this and/or a more balanced discussion is needed.*

Answer: Thank you for the encouragement and constructive criticism. As described above in the revised manuscript we did both, provided a gamut of new supporting data and balanced the discussion.

Reviewer #3

In this manuscript, the authors propose that p53 loss of heterozygosity (LoH) serves as the major driving force behind the progression and invasiveness of colorectal tumors (CRC). While p53 mutation rate is elevated in this type of cancers, mutant p53 require a stabilization process in order to exert their gain of pathogenic functions (GoF). The authors show that the loss of the remaining wildtype (wt) p53 allele in heterozygous tumors is a pre-requisite for the stabilization of mutant p53 proteins. The manuscript by Sener, Stender and coworkers identifies the proteotoxic stress defense response (HSR) and particularly HSF-1, its main effector, as the key factor behind p53 LoH. HSF-1 is indeed kept in check as long as a residual wtp53 remains through a repressive axis: WTP53-p21-MLK3-MAPK-HSF1 thereby preventing mutp53-R248Q protein stabilization and GoF in mutp53/+ tumors. The provided data shed a new light on the crucial yet understudied process that is p53 LoH which will prove valuable to the field. The manuscript is well-written, clearly illustrated and the authors rely on a relevant mouse model. However, several aspects need to be clarified and properly reviewed by the authors.

Point by point:

Rev3_1. Figure 1 and S1: the authors use 2 pertinent intestinal/colorectal models to clearly demonstrate an LOH-induced mutant p53 stabilization, which consequently increases colorectal tumors. However, the total number of tumors (Figure 1D, left panel) and the total mouse number with invasive tumors (figure 1G, right panel) in the TAM-inducible model are unchanged in the “LOH mice” versus “no LOH mice”. Only size (tumor growth) and invasiveness are changed. This should be clearly indicated in the text describing the figure.

Answer: For Figure 1D left panel we state a *trend* for increased total tumors/mouse in the LOH group. Tumor size was significantly increased.

Old Fig 1G (now Figure 1H) quantitates invasiveness in two complementary ways. In old Figure 1G right panel, we erroneously had mixed up the diagram legends which caused the Reviewer's confusion. We apologize, this is now corrected. Both panels of Figure 1H (*left* 'total invasive tumors' and *right* 'total mouse numbers with invasive/non-invasive tumors') indicate that invasion is enabled after p53LOH. So, the corrected Figure 1H *right* says that 100% of mice with LOH have invasion, while 100% of mice without LOH are free of invasion (highly significant in Fisher's Exact test).

Rev3_2. Figure 2 and S2: The presented data indicate that wtp53 retains the ability to activate several transcriptional targets (e.g. Cdkn1a, Gadd45a and Sfn) but not Mdm2. This interesting point needs to be clarified. Are the observed effects (i.e. no dominant negative effect of p53-R248Q, suppression of HSF1 target genes by residual WTP53 activity) restricted to the use of Nutlin or do other p53 inducers (irradiation, uv, doxorubicin...) lead to similar effects?

Answer: Good point. Concerning Mdm2 which is not activated by WTP53 in *tumors* of Nutlin-treated mice (now Suppl Figure S2C), we show that Mdm2 is strongly induced in Nutlin-treated p53^{Q/fl} heterozygous *organoids* (now Figure 2D with improved label, previously Figure 7B). Thus, the selective lack of Mdm2 induction *in vivo* versus in organoids suggests that *in*

in vivo Mdm2 requires higher levels of Nutlin and/or more time than other p53 targets. This is now discussed in Results. For improved clarity old Figures 7C-E, G were moved to Figure 2 and Suppl Figure S2.

Moreover, we now show that additional p53 activators such as Doxorubicin and 5-FU also induce the p53 target *Cdkn1a* (new Figure S2P) and suppress HSF1 targets (new Figure 2F) in heterozygous p53^{Q/m} organoids. Concerning DNE we now state that while the Q allele exerts a *partial* dominant negative effect (Figures 2B and S2K), importantly the remaining WTp53 allele retains broad residual transcriptional activity as indicated by induction of the Hallmark WTp53 pathway (also extensively discussed above, pages 5).

Rev3_The authors conclude an absence of DNE from p53-R248Q although Cdkn1A and Bbc3 (S2-F) show a marked reduction in p53Q/+ vs. both p53+/+ and p53+/- without Nutlin could suggest otherwise. Nutlin treatment leads to a balanced increase of both wt and mutant forms of p53 (p12-L332) thereby limiting the observation of a DNE which requires a high mt/wt ratio. The statement should be amended in order to reflect a more balanced conclusion as it is the case in discussion.

Answer: The reviewer is completely right. We do see a partial DN effect by qRT-PCR (for Cdkn1a and Bbc3, Figure S2J) and new RNAseq analysis (new Figure 2B). Importantly, however, despite a partial dominant-negative effect by the Q allele, we clearly identify that the remaining WTp53 allele in p53^{R248Q/+} tumors retains broad residual activity and still represses the HSF1 chaperone axis, thereby preventing mutp53^{R248Q} stabilization, GOF and invasion. The text has been changed accordingly.

Rev3_3. Figure 3: The authors have knocked p53 out using shp53 in homozygous WTp53 HCT116 and RKO lines to demonstrate that Nutlin-induced HSF1 suppression is rescued by p53 depletion. Conversely p53^{-/-} HCT116 cells can be used to express WTp53 and induce an expected repression of HSF-1 to complete the analysis.

Reviewer-only Figure 8. Non-specific transfection stress even with empty vector induced a strong HSR response with upregulation of Ser326-HSF1 that could not be overcome by WTp53 transfection + Nutlin. One of 2 representative experiments is shown.

Answer: In principle this is a good idea and we tried, but could not get an informative result. As shown in Reviewer-only Figure 8, non-specific transfection stress in HCT116 p53^{-/-} cells even with empty vector (500 ng pcDNA3.1 per 6-well for 72 hrs) induced a strong HSR response with upregulation of Ser326-HSF1 that could not be overcome by WTp53 transfection + Nutlin treatment. Thus it appears that transfection mainly exerts a p53-independent stress, because we do see pHSF1 activity repression with p53-dependent heat shock stress (Figure 3H), Nutlin (new Figure S2E) and genotoxic stress by Doxorubicin and 5-FU (new Figure 2F).

Rev3_Is p53-R248Q unable to activate HSF-1 in HCT116?

Answer: The Reviewer is asking whether Q shows GOF for pHSF1 activation. This is a very good point, thank you, since stabilized p53^{R248Q/W} might create a positive feed-forward loop to further increase tumor-promoting HSF1 activity which would be a GOF. Since we cannot use plasmid overexpression as explained above, to get at this question we used homozygous endogenous p53^{R248Q/W} CRC cell lines as we did in *Schulz-Heddergott et al. 2018*, and silenced mutp53 by p53 siRNAs (**Reviewer-only Figure 9**). Indeed, we see dephosphorylation of pSer326-HSF1 indicating suppression of HSF1 activity after mutp53 silencing which strongly points to the existence of such a positive feed-forward loop. However, since these are preliminary data and need more careful analysis we do not include them in this manuscript. But we thank the Reviewer for stimulating an interesting direction for a future study.

Reviewer-only Figure 9. pHSF1 levels were determined in human CRC cell lines carrying R248 missense mutations after knockdown of mutp53 by siRNA. At 72 hrs post transfection with different p53 siRNAs or scrambled (scr) control siRNAs, protein lysates were prepared. Immunoblot analysis.

Rev3_4. The ratio between wt and mutant p53 appears crucial to mutp53 DN activity which could be clarified by using a response dose of mutp53 in p53^{-/-} HCT116 cells. This should lead to a gradual effect on HSF-1 while the opposite could be observed by adding increasing doses of wtp53. Such effects on p53 targets have been documented in model organisms (see Brachmann et al. 1996, Monti et al. 2002, 2011, Billant et al. 2016).

Answer: We tried to do this response dose titration experiment but again ran into the non-specific transfection stress problem with plasmid overexpression as explained above. Moreover, *MUT vs WT* TP53 copy numbers are very important for mutp53 protein stability and DNE (mixed heterotetramers). However, with plasmid transfection we uncontrollably increase mutant and wildtype TP53 copy numbers, different from the stoichiometry of endogenous mixed protein complexes. This modelling problem is also emphasized in a recent review by G Lozano, a leader in the p53 field (*Gencel-Augusto & Lozano 2020 Genes Dev*, PMID: 32873579).

Rev3_Minor points:

1. It is indicated in the Methods section the use of HCT115 p53 null cells. Are they HCT 116 p53^{-/-} cells?

Answer: Yes, we used HCT116 p53^{-/-}. This typographical error has been corrected.

2. Sup Figure 3A: error bars are not homogeneous

Answer: Error bars are now homogenous.

3. Some typographical errors, ex.: In the chapter: “Activated Wtp53 represses HSF1 activity in human colorectal cancer cells”: line 197: Conversely, mutp53-harboring CRC...

Answer: Thank you, we corrected it.

4. [Does not exist]

5. The title suggests that the authors have studied the role of HSF-1 in mutant p53-mediated invasion thoroughly, but additional validation is required, preferentially using invasion assays to provide further knowledge of HSF1- mediated EMT process, particularly regarding the already largely accepted anti-invasive activity of p53.

Reviewer-only Figure 10. EMT marker expression is not directly regulated by HSF1 in WTP53-containing cells. HCT116 and RKO cells were transfected with two siRNAs against HSF1 or scrambled control siRNA (scr2) for 72 hrs. A 24 hrs Nutlin treatment serves as control. qRT-PCRs for the indicated mRNAs normalized to HPRT1 mRNA. Note that CDH1 is not expressed in RKO cells. Mean \pm SEM of 2 independent experiments, each with two technical replicates, pipetted in duplicates. Student's t-test, $p^*=0.05$, $p^{**}=0.01$, $p^{***}=0.001$.

Answer: The new title no longer implies a role of HSF1 in mutant p53-mediated invasion. We agree with the Reviewer, regarding p53, we clearly see that when one Wtp53 allele is present, there is no invasion *in vivo* (Figures 1G,H and S1D). HSF1 target genes per se might not directly drive EMT in the cells we tested. Rather, in the case of GOF mutp53 that is stabilized via HSF1-induced chaperones such as Hsp90/Hsp70/Hsp40, HSF1 could indirectly regulate EMT.

Still, to check whether HSF1 directly regulates EMT, we now *silenced HSF1 in Wtp53-harboring* HCT116 and RKO cells and found, not surprisingly, that EMT markers are not regulated after HSF1 depletion (Reviewer-only Figure 10). Thus, as we expected, there is no direct EMT regulation by HSF1 in Wtp53 cells.

Conversely, the role of HSF-1 on pro-invasive and EMT induction activities of mutant p53 should be clarified, in regards to its GOF activity versus dominant negative activity on Wtp53

Answer: Since the ratio between WT and MUT p53 is crucial to mutp53 DNE activity, the Reviewer is asking about stoichiometry. Unfortunately, due to the limitations with plasmid transfections introducing uncontrollable TP53 copy numbers as well as p53-independent non-specific stress responses (discussed above), we cannot address DNE *in vitro*. Instead, we used our p53^{Q/fl} \pm 4OHT organoid system in matrigel-based transwell assays following the validated Nakayama protocol (Nakayama et al 2017, PMID: 28628120 and Nakayama et al 2020 PMID: 32393735). Indeed we found that heterozygous p53^{Q/fl} organoids in matrigel

transwell assays are only minimally invasive, but after p53LOH they upregulate their EMT markers and more than double their invasiveness (new Figures 7C, D, F), concomitantly with upregulation of HSF1 target genes (new Figure 7E).

Moreover, we performed HSF1 silencing in the highly invasive mutp53 CRC cell lines SW480 and DLD1 (Reviewer-only Figure 3). Due to the long half-life of chaperone-stabilized mutp53, we had to silence for 6 days to analyze the effects of mutp53 degradation in transwell assays. Note that after 6 days of siHSF1 mutp53 degradation in DLD1 is stronger than in SW480. In complete agreement, the degree of migration inhibition correlates with the degree of mutp53 knockdown which in case of DLD1 is near-complete. In contrast, SW480 cells show significant but less drastic inhibition commensurate with their only modestly silenced mutp53 levels. After only 3 days of HSF1 silencing in both cell lines, while HSF1 knockdown is highly efficient, mutp53 levels remain high which correlates with a failure to inhibit migration. Together, these data show that the *level* of mutp53 determines GOF activities such as migration.

6 days silencing

3 days silencing

Reviewer-only Figure 3. *left* HSF1 knockdown in human CRC cell lines carrying p53 S241F(DLD1) and p53 R273H/P309S (SW480) missense mutations. At 72 hr after transfection with two different HSF1 siRNAs or scrambled (scr) control siRNA, protein lysates were prepared. Immunoblot analysis. *Top* 6 days of silencing (siRNA addition on day 1 and day 3); *bottom* 3 days silencing (siRNA addition on day 1 only).

right Quantification (right) of transwell migration assay of H1299 cells expressing p53 R248Q treated +/- IL-6. (E and F) Mean ± SEM of two independent experiments in duplicates, Student's t test.

6. Figure 6: as invasion is the acquired phenotype of p53LOH tumors, the analysis of tumor samples should also include “disease free survival” analysis, not only global survival.

Reviewer-only Figure 11. Kaplan-Meier disease free (DFS) survival curve of our TCGA COADREAD patients. *red* MS p53 + LOH, n=60 patients with 7 recurrences; *green* WTp53, n=87 patients with 10 recurrences.

Answer: As requested we did the DFS curve for our TCGA COADREAD cohort which encompasses all disease stages 1-4 (**Reviewer-only Figure 11**, log rank = 0.22). There is no statistical significance. As indicated, among 60 MS+p53LOH patients 7 patients experienced a recurrence; and among 87 WTp53 patients 10 patients experienced a recurrence. Unfortunately with such few events the mean survival cannot be calculated. Thus, this analysis is not informative.

Interestingly, however, all patients with p53LOH exhibit higher disease stages than WTp53 patients. As shown in **new Figures S6D,E**, we see higher lymph node (N) and metastatic (M) stages in patients with MS/NS/FS+LOH compared to WTp53.

7. Figure 6: It is claimed (in the discussion section) that in human CRCs strong constitutive oncogenic stress from K-RAS/EGFR/ TGFβR/ PDGFR mutations are preeminent, which promotes sufficient proliferative stress levels to drive p53LOH spontaneously. To strengthen this point, the analysis should also provide the mutation status of K-RAS/EGFR/ TGFβR/ PDGFR. Is there a correlation between MS p53 + LOH and K-RAS/EGFR/ TGFβR/ PDGFR mutations?

Answer: Very good point, we did as requested. In our COADREAD cohort, APC and KRAS mutations are predominant. EGFR, TGFbR and PDGFR are present but infrequent (between 2-4%) and thus insufficient for statistical analysis. KRAS mutations occurred with 42% frequency. However, we failed to detect a correlation between KRAS mutation and p53LOH (**Reviewer-only Figure 12**).

n = 435 p = 1	WTp53	FS/NS/MS + LOH	n = 222 p = 0.2748	FS/NS + LOH	MS + LOH
KRAS WT	121	127	KRAS WT	35	92
KRAS mut	92	95	KRAS mut	20	75

Reviewer-only Figure 12. CRC patients fail to show a correlation between mutant KRas and p53LOH. COADREAD TCGA data, patient numbers as indicated. Fisher's Exact Test.

In contrast, APC mutations occur with 72% frequency and strongly correlate with p53LOH (FS/NS/MS + LOH, **new Figure 6G**). MS + LOH cases are not enriched for APC mutations compared to FS/NS + LOH (see **Reviewer-only Figure 13**). While this excludes a GOF effect *when all missense p53 mutants are lumped together*, it does not exclude potential GOF for specific missense variants

n = 222 p = 0.5043	FS/NS + LOH	MS + LOH
APC WT	9	22
APC mut	45	146

Reviewer-only Figure 13. CRC patients fail to show a correlation between mutant APC and specific mutant p53 status (FS/NS versus MS). COADREAD TCGA data, patient numbers as indicated. Fisher's Exact Test.

such as 248Q. Moreover, we now show a strong correlation between high expression of HSF1 and HSF1 target genes (e.g. HSF1^{high}) and APC mutations in tumors from CRC patients (new Figure 6H). Thus, CRC-driving signaling pathways such as APC mutations likely serve as stress signal to enforce p53LOH in CRC.

8. Figure 6: On the same line, does upregulation of HSF1 targets correlate with mutation status of K-RAS/EGFR/ TGFβR/ PDGFR?

n = 487	KRAS WT	KRAS mut	P
HSF1 low	98	90	
HSF1 high	178	112	0.0472
HSPH1 low	117	97	
HSPH1 high	159	105	0.2276
HSP90AB1 low	138	88	
HSP90AB1 high	138	114	0.1659
HSPE1 low	116	83	
HSPE1 high	160	119	0.8515
HSPD1 low	61	55	
HSPD1 high	215	147	0.2348
HSPB1 low	156	107	
HSPB1 high	120	95	0.4576

Answer: As requested, we performed this analysis in Reviewer-only Figure 14. As seen in row 1, although HSF1 expression itself slightly correlates with KRAS mutations, HSF1 target genes do not correlate with the KRAS status. This suggests that KRAS is not a stress signal driving p53LOH. We revised our discussion accordingly. Instead, other stress signals such as APC mutations are likely involved in enforcing p53LOH in CRC (new Figure 6H). This notion is further supported by Davies *et al.*, 2015, PMID: 26320184) who showed that APC mutants activate HSF1-dependent stress pathways in intestinal tumors *in vivo*.

Reviewer-only Figure 14. CRC patients fail to show a correlation between KRas mutation and upregulation of HSF1 and its targets. COADREAD TCGA data, patient numbers as indicated. Fisher's Exact Test.

9. Figure 7B: How do the authors explain the increase in Mdm2 and Cdkn1a expression after treatment with Nutlin of 4OHT (p53LOH)?

Answer: This is now Figure 2D. We explain this residual induction by Nutlin with an incomplete loss of wtp53 due to incomplete recombination, and had stated this in line 290-294 of the original manuscript. Cre recombinase-mediated allele deletion is never 100% efficient. Indeed, as shown in the Wtp53-allele specific qRT-PCR in new Figure S20, residual Wtp53 mRNA post 4OHT is still present, explaining the partial Nutlin response in 4OHT-treated organoids.

10. Figure 7E: the figure is not sufficiently described, in particular for the E-Cadherin staining. It remains unclear what we need to see. Is it a loss of the E-Cadherin staining, in agreement with an EMT process in p53LOH organoids?

Answer: This is now Figure 2G which illustrates mutp53 stabilization after p53LOH in tumor cell nuclei. The purpose of E-Cadherin staining here was to counterstain the epithelial cells to better visualize the organoids.

11. Figure 7F: is there any significant difference in miR34 expression between EtOH+Nutlin and 4OHT+Nutlin, compared to their respective DMSO controls?

Answer: This is now Figure 2D. The D/N ratios in p53^{Q/fl} is 1.77 and in p53^{Q/Δ} is 1.34, respectively. Again there is residual induction as explained in point # 9 above.

More importantly, there is no difference between EtOH-DMSO in Heterozygous organoids and 4OHT-DMSO in p53LOH organoids (in miR32a, Vimentin and Snai1 expression), suggesting that the EMT process is not complete. The authors should clarify this point.

Answer: The Reviewer is correct and refers to what is now Figure 7A. Only 4 days after p53LOH when this was measured, levels of mesenchymal markers Vimentin and Snail had not yet increased.

To further support our hypothesis that the establishment of GOF after p53LOH takes time, we newly analyzed the p53^{Q/fl}; vilCreERT2 tumor-derived organoids over the longer time span of 20 days. To optimize recombination efficiency, we used 3 weekly pulses of 4OHT. p53LOH was confirmed with ~ 80% loss of the WTP53 allele (new Suppl Figure S7B) and decreased p53 target gene expression (new Suppl Figure S7C) which confirmed that the retained WTP53 allele in Q/fl had residual transcriptional activity. Indeed, after long-term p53LOH EMT markers were now increased (new Figure 7C) and invasive branching morphology appeared in p53^{Q/Δ} organoids (new Figure 7D). Moreover, upon p53LOH organoids exhibited upregulation of HSF1 target genes (new Figure 7E) and increased invasiveness in transwell assays (new Figure 7F). Notably, GOF activities that we had previously established such as deregulated pStat3 target genes (*Schulz-Heddergott et al. Cancer Cell 2018*) were also upregulated (new Suppl Figure S7D). Thus, combined upregulation of EMT genes and Stat3 signaling (new Suppl Figure S7D) enable invasiveness following p53LOH.

12. Figure 7: to conclude on an EMT process: - The levels of E-Cadherin should be analyzed and compared to N-Cadherin/Vimentin/Snail levels. Is there a Cadherin shift associated with EMT process?

Answer: As alternative to analyzing a possible E-cadherin/N-cadherin shift, we confirmed EMT induction with long-term p53LOH and analyzed tumor-derived organoids 20 days after 4OHT-induced p53LOH for EMT markers (Vim, Snai1, Snai2 and Zeb1) and HSF1 target genes. We now see a strong up-regulation of both groups of readouts (new Figures 7C,E). Moreover, transwell assays and induced morphological branching of organoids confirmed an increase in the invasive phenotype (new Figures 7D,F). Finally, extended Nutlin stress in p53^{Q/fl} organoids over 21 days induces spontaneous p53LOH concomitant with upregulation of EMT markers and HSF1 target genes (new Figures 7G-I).

The effect of HSF-1 on EMT process (i.e. miR34, Ecadherin/ Vimentin/Snail levels) should be analyzed, possibly in cell lines (HCT116, RKO...)

Answer: As requested, we did this analysis in HCT116 and RKO cells in Reviewer-only Figure 12. See also Rev 3 point # 5 above. We find that EMT marker expression is not regulated by HSF1 in WTP53-containing CRC cells.

13. The authors speculate that WTP53-mediated HSF1 suppression exerts a strong selection pressure for p53LOH. However, they also show that “murine CRCs might not be stressed enough to spontaneously activate WTP53 and suppress HSF1”, indicating that baseline AOM/DSS-induced tumors have insufficient stress levels to drive p53LOH spontaneously. The authors speculate that “in human CRCs strong constitutive oncogenic stress from K-RAS/EGFR/ TGFβR/ PDGFR mutations are preeminent”, providing sufficient proliferative

stress levels to drive p53LOH. However, the link between K-RAS/EGFR/ TGF β R/ PDGFR mutations and HSF1 activation/repression is not addressed in this manuscript.

Answer: To further support that a stressed tumor milieu enforces p53LOH, we now analyzed the KRAS-driven KPC (Kras; p53^{R172H/+}) pancreatic cancer model and show a correlation between spontaneous p53LOH (indicated by p53^{high} staining) and a trend to poorer survival (new Figure 7J) as well as a correlation to co-localized upregulated Hsp70^{high} and pHSF1^{high} staining in tumor epithelial cells (new Figures 7K and S7H, see below).

Importantly, this model undergoes high spontaneous p53LOH at a frequency of ~70%, indicated by conversion from undetectable negative to positive stabilized p53 staining (*Olive et al. 2004; Hingorani et al. 2005*). To this end, we used our own KPC cohorts (generated by coauthor Albrecht Neesse, PMID: 28077438, PMID: 31878349 and PMID: 24874484), stained KPC tumor sections for p53 and scored for the percentage of positive tumor cells (see Methods). Based on the very strong correlation between stabilized mutp53 (immunohistochemical p53 positivity) and sequence-verified p53LOH in the *same* tumor (Hingorani et al. 2005, PMID: 15894267), p53^{high} cases (60%) were scored as having undergone p53LOH. p53^{low} cases were scored as heterozygous, harboring instable mutp53.

I. Kaplan-Meier survival of p53LOH versus no LOH (new Figure 7J):

p53^{high} cases (LOH, n=25, median = 133 days) tended to have shorter survival than p53^{low} cases (no LOH, n=15, median = 187 days), p=0.0734.

Of note, because of the high stroma content of KPC PDAC tumors, qRT-PCR on whole tumor lysates is unsuitable as alternative technique. Indeed, we failed to see upregulation of p53 and HSF1 target genes due to dilution of the tumor epithelial signal by the abundant desmoplastic stroma. Thus, we think that staining for p53 by IHC is the better choice in KPC tumors.

II. Correlation between p53LOH and high Hsp70 and high phospho-HSF1 staining

We now show that p53^{high} (p53LOH) colocalizes with Hsp70^{high} and pHSF1^{high} staining in tumor epithelial cells. This correlation further supports HSF1 axis-mediated mutp53 stabilization (new Figures 7K and S7H).

REVIEWER COMMENTS

Reviewer #1 (Remarks to the Author):

The revised manuscript was substantially extended and reorganized. Since additional RNA-seq analyses of Nutlin treated p53^{+/+}, p53^{-/+}, and p53Q/+ tumors were done, Figure 2 was completely changed and combined with the previous Figure 7. The authors use RNA-seq data to prove that the WTp53 allele in heterozygous colorectal tumors retains partial activity. The repression of HSF1 target genes in vivo by Nutlin (previous Fig. 2D) is currently being studied in tumor-derived organoids. My question is why data about the expression of HSF1 target genes in tumors was not extracted from RNA-seq (consistency of qPCR and RNA-seq would strengthen the evidence)? The changes in mRNA levels shown in the manuscript in many cases are very small and the method used (RT-qPCR and relative values in [ratio (2^{-ddCT})] is not the best in this situation. RNA-seq data could reinforce the observation that Nutlin treatment represses HSF1 target genes. A completely new Figure 7 (and S7) was created. The newly added data is intended to further support the GOF activity of the p53R248Q allele. Based on Figures 7B-F and S7A-D, the authors stated that p53LOH derepresses HSF1 activity and triggers GOF mutp53 driven invasion. Invasion is evident, but the derepression of HSF1 activity is poorly documented. In my opinion, such small changes in mRNA level of HSF1-regulated genes (estimated by RT-qPCR and shown in Fig. 7E) are not credible. Hence, more convincing data documenting p53LOH derepression of HSF1 should be provided.

Furthermore, I do not understand why based on the results shown in Fig. 7H (showing the expression of HSF1 target genes) it was stated that long-term chronic Nutlin treatment of heterozygous p53Q/fl; viiCreERT2 organoids induced spontaneous p53LOH (Figure legend). To state this convincingly, the presence/absence of the p53 wt allele should be examined (although I accept the stabilization of p53 shown e.g. in Fig. 2G as a confirmation of LOH). In the main text (line 367), when describing the upregulation of HSF1 targets (Figure 7H), it would be reasonable to recall that these genes are repressed after a short treatment with Nutlin (Figure 2E).

In my first review, I had one major concern that has not been clarified. The authors state that wild type p53 actively represses the tumor-promoting HSF1-regulated chaperone system and proteotoxic stress response. To study this mechanism, they silenced p53, p21, MLK3, CDC2, CDC2, CDK4, CDK6, PLK4, but not HSF1. Based on obtained results, they proposed a repressive WTP53-p21-MLK3-MAPK-HSF1 signaling cascade as the underlying mechanism that creates a driving force for losing the WTP53 allele (and the selection pressure for p53LOH).

In my opinion, to finally confirm the significance of HSF1 in this signaling cascade, the expression of HSF1-regulated genes after Nutlin treatment (exactly as shown in Figs 3D and S3A after p53 downregulation) should be tested in cells after silencing/knockout of HSF1. Moreover, an evaluation of nascent RNA would be more appropriate to show the transcriptional repression. Instead, authors silenced HSF1 in CRC cell lines carrying p53 S241F(DLD1) and p53 R273H/P309S (SW480) missense mutations and studied their migratory properties (Reviewer-only Figure 3), which is interesting but does not contribute to direct verification of their hypothesis.

I strongly recommend checking the inhibitory effects of the Nutlin treatment after HSF1 silencing/knockout. If necessary, I can provide (without any obligation) colon carcinoma and breast adenocarcinoma cell lines with HSF1 functional knockout (obtained using DNA-free CRISPR/Cas9 system) and corresponding control cells for this experiment.

Some mistakes to be corrected:

All abbreviations should be explained on first use.

Line 228: "targets including HSP90AA1, HSPA1A, HSPH1, HSPB1, DnaJA1 and DNAJB1."

Please keep the same gene nomenclature throughout the manuscript (capital, italic).

Line 247/248: "Nutlin correlated with strong dephosphorylation of pSer326-HSF1."

Nutlin treatment correlated

Line 700: "a set of genes from Vilaboa et al.⁶⁴ upregulated by RNAseq in heat-shock treated HeLa cells."

Genes were not upregulated by RNA-seq.

Line 729: "the HSF1 target gene set was extracted from Vilaboa et al. (REF,"

Please complete the reference.

Fig. S6B – wrong number of cases for MS + LOH (?).
The description of Fig. S6D-E is not satisfactory.

I apologize that in my previous review I confused Fig. 4F and Fig. 4I (the concern was that two lower blots do not fit the upper ones). Nevertheless, it was corrected in the revised manuscript.

Reviewer #2 (Remarks to the Author):

The authors have provided an extensive rebuttal to my previous comments and have mostly addressed my previous concerns. I now think the paper is suitable for publication.

Reviewer #3 (Remarks to the Author):

The revised manuscript is much improved and addresses the reviewers concerns. Overall, the manuscript provides clear evidence supporting a role for HSF-1 axis in mutp53 stabilization. In particular the newly added data further support GOF activity of the p53R248Q allele (new figures 7 and sup7). In addition, the strong correlation found between APC mutations and p53LOH, further supports HSF1 axis-mediated mutp53 stabilization promoting invasion. The revised manuscript is much improved and addresses the concerns of reviewers. Overall, the manuscript provides clear evidence supporting a role of the HSF-1 axis in the stabilization of mutp53. In particular, the newly added data further supports the GOF activity of the p53R248Q allele (new Figures 7 and sup7). In addition, the strong correlation found between the APC and p53LOH mutations, enhances the role of the mutp53 stabilization / HSF1 axis in tumor invasion. Responses to reviewers clarify open questions and new figures reinforce key findings.

Point-by-Point Rebuttal Letter

We are very grateful and would like to thank all Reviewers for their careful, constructive and very positive evaluation of our paper. In response, we performed new experiments and extended our analyses.

→ new RNAseq analysis from long-term p53LOH organoids

Attached is the Point-by-Point Rebuttal letter. Reviewer text in black, our answers in blue.

Some newly generated data are shown as Rebuttal-only Figures since they were not deemed essential for the paper. In sum, we are confident that all remaining concerns have now been adequately addressed.

Added text is in RED.

The following figures were newly generated or revised and are marked in RED.

Suppl Figure S2F - Extraction of seven HSF1 target genes, originally measured by qRT-PCR, from our tumor RNAseq. Confirming consistency between both methods.

Figure 7E – GSEA Enrichment blot of HSF1 targets (667 gene set, Vilaboa et al) analyzed by an additional RNAseq analysis using the same long-term p53LOH *organoid* RNA samples that we had originally used for single gene qRT-PCR analysis. Together, this provides convincing data documenting p53LOH derepression of HSF1 target genes.

Suppl Figure S7C – GSEA Enrichment blot of p53 target genes analyzed by the additional new RNAseq analysis using the same long-term p53LOH *organoid* RNA samples that we had originally used for qRT-PCR analysis.

Figure 7H – Examination of the presence/absence of the p53 wt allele in chronic Nutlin treatment

Reviewer #1 (Remarks to the Author):

The revised manuscript was substantially extended and reorganized.

Rev1_ Since additional RNA-seq analyses of Nutlin treated p53^{+/+}, p53^{-/+}, and p53^{Q/+} tumors were done, Figure 2 was completely changed and combined with the previous Figure 7. The authors use RNA-seq data to prove that the WTp53 allele in heterozygous colorectal tumors retains partial activity. The repression of HSF1 target genes in vivo by Nutlin (previous Fig. 2D) is currently being studied in tumor-derived organoids.

My question is why data about the expression of HSF1 target genes in tumors was not extracted from RNA-seq (*consistency of qPCR and RNA-seq would strengthen the evidence*)? The changes in mRNA levels shown in the manuscript in many cases are very small and the method used (RT-qPCR and relative values in [ratio (2^{-ddCT})] is not the best in this situation. RNA-seq data could reinforce the observation that Nutlin treatment represses HSF1 target genes.

Answer: Excellent point, the Reviewer wants to see consistency between tumor qRT-PCRs and RNAseq for individual HSF1 target genes. We now did as requested (**new Supp Figure 2F**). (Note, the previous Figure 2D that Rev1 is referring to above is now Supp Figure 2E). We extracted from our tumor RNAseq data the four randomly chosen HSF1 target genes that we had measured in Supp Figure 2E, plus three additional HSF1 target genes that we had chosen for qRT-PCRs of organoids in Figure 2E.

Importantly, all HSF1 targets were also significantly down-regulated by RNAseq in p53^{+/+} tumors after Nutlin, indicating good consistency between both methods of target gene measurement, reinforcing our observation that Nutlin treatment represses HSF1 target genes (**new Supp Figure 2F**).

Rev1_A completely new Figure 7 (and S7) was created. The newly added data is intended to further support the GOF activity of the p53^{R248Q} allele. Based on Figures 7B-F and S7A-D, the authors stated that p53^{LOH} derepresses HSF1 activity and triggers GOF mutp53 driven invasion. Invasion is evident, but the derepression of HSF1 activity is poorly documented. In my opinion, such small changes in mRNA level of HSF1-regulated genes (estimated by RT-qPCR and shown in Fig. 7E) are not credible. Hence, more convincing data documenting p53^{LOH} derepression of HSF1 should be provided.

Answer: Valid point. In response, we have now performed an additional new RNAseq analysis using the same long-term (spontaneous) p53^{LOH} organoid RNA samples that we had originally used for qRT-PCR analysis in Figure 7F. We analyzed the new RNAseq data using the HSF1 target gene list (667 genes) published by Vilaboa et al. (generated from heat shock-induced Hela cells). Indeed, it reveals that in response to p53^{LOH}, organoids exhibited upregulation of approx. 20 % (124/667) of all HSF1 target genes (**new Figure 7E**), fully confirming the single gene results in Figure 7F. Additionally, we analyzed this RNAseq using the Hallmark p53 gene sets from GSEA analyses that are associated with WTp53 activity (**new Figure S7C**), fully confirming the loss of WTp53 activity after long-term p53^{LOH}.

Rev1_Furthermore, I do not understand why based on the results shown in Fig. 7H (showing the expression of HSF1 target genes) it was stated that long-term chronic Nutlin treatment of heterozygous p53^{Q/fl}; *vilCreERT2* organoids induced spontaneous p53^{LOH} (Figure legend).

To state this convincingly, the presence/absence of the p53 wt allele should be examined (although I accept the stabilization of p53 shown e.g. in Fig. 2G as a confirmation of LOH).

Answer: Actually, we had already done this - that is, examined the presence/absence of the p53 wt allele using a specific qRT-PCR reaction for the murine Wtp53 flox allele (see Revision_1, Suppl Figure 7F, left diagram). But to make sure that this important point is not overlooked, we now moved this data to new Figure 7H bottom.

Rev1_In the main text (line 367), when describing the upregulation of HSF1 targets (Figure 7H), it would be reasonable to recall that these genes are repressed after a short treatment with Nutlin (Figure 2E).

Answer: Good point. We corrected the text accordingly.

Rev1_In my first review, I had one major concern that has not been clarified. The authors state that wild type p53 actively represses the tumor-promoting HSF1-regulated chaperone system and proteotoxic stress response. To study this mechanism, they silenced p53, p21, MLK3, CDC2, CDK4, CDK6, PLK4, but not HSF1. Based on obtained results, they proposed a repressive Wtp53-p21-MLK3-MAPK-HSF1 signaling cascade as the underlying mechanism that creates a driving force for losing the Wtp53 allele (and the selection pressure for p53LOH).

In my opinion, to finally confirm the significance of HSF1 in this signaling cascade, the expression of HSF1-regulated genes after Nutlin treatment (exactly as shown in Figs 3D and S3A after p53 downregulation) should be tested in cells after silencing/knockout of HSF1. Moreover, an evaluation of nascent RNA would be more appropriate to show the transcriptional repression. Instead, authors silenced HSF1 in CRC cell lines carrying p53 S241F(DLD1) and p53 R273H/P309S (SW480) missense mutations and studied their migratory properties (Reviewer-only Figure 3), which is interesting but does not contribute to direct verification of their hypothesis.

I strongly recommend checking the inhibitory effects of the Nutlin treatment after HSF1 silencing/knockout. If necessary, I can provide (without any obligation) colon carcinoma and breast adenocarcinoma cell lines with HSF1 functional knockout (obtained using DNA-free CRISPR/Cas9 system) and corresponding control cells for this experiment.

Answer: Good point, for consistency, since we had silenced all other components in the signaling pathway, the Reviewer suggests to also silence HSF1. We already established that Nutlin suppresses HSF1 target gene expression. Moreover, the HSF1 transcription factor regulates stress-induced chaperone expression. Thus, we expect that HSF1 silencing suppresses chaperone expression (confirming them as true HSF1 targets). However, given the above, we expect that additional Nutlin treatment might or might not reduce chaperone expression further, depending on the relative sensitivity of HSF1 targets. We now addressed this question directly using wtp53-expressing RKO and HCT116 cells where we silenced HSF1 with two different siRNAs, together with DMSO or Nutlin (see Rebuttal-only Figure 1). Indeed, HSF1 silencing reduced well-known HSF1 target genes such as HSPBP1, HSPA1A and HSPH1. Additional Nutlin significantly decreased the expression of HSPH1, while expression of HSPBP1 and HSPA1A failed to show a consistent cumulative effect, thus giving a mixed overall answer. Since this experiment does not provide a clear answer, we did not include it in the manuscript.

Instead, HSF1 *overexpression* might better show the association between Nutlin/p53 and HSF1. We want to point out that we already did such experiments in Figures 3G, S3F, 4G and S4E. HSF1 target genes are up-regulated after HSF1 overexpression, whereas the same genes are repressed after additional Nutlin treatment.

Concerning nascent RNA to show the transcriptional repression, we respectfully point out that we are interested in the net sum mRNA levels of HSF1 targets after 24 hrs Nutlin treatment. To best measure this relationship we chose qRT-PCR assays which capture the steady-state mRNA level that is of interest here, rather than nuclear run-on assays which capture only the newly transcribed nascent RNA, a method more suitable for kinetic questions of newly synthesized RNA, a point not of interest here.

Rebuttal-only Figure 1: Nutlin treatment after *HSF1* mRNA silencing. 72 hrs after transfection with two different siRNAs against *HSF1* mRNA, cells were treated with DMSO or 10 μM Nutlin for 24 hrs. qRT-PCR, normalized to *RPLP0* mRNA. Relative values given in [ratio ($2^{-\text{ddCT}}$)]. Mean \pm SEM of ≥ 2 independent experiments, each with two technical replicates. Student's t-test, $p \leq 0.05$, $p^{**} \leq 0.01$, $p^{***} \leq 0.001$; ns, not significant.

Rev1 Some mistakes to be corrected:

- All abbreviations should be explained on first use.
- We changed the text accordingly.

- Line 228: “targets including HSP90AA1, HSPA1A, HSPH1, HSPB1, DnaJA1 and DNAJB1.” Please keep the same gene nomenclature throughout the manuscript (capital, italic). We changed the text accordingly.

- Line 247/248: “Nutlin correlated with strong dephosphorylation of pSer326-HSF1.” Nutlin treatment correlated We changed the text accordingly.

- Line 700: “a set of genes from Vilaboa et al.⁶⁴ upregulated by RNAseq in heat-shock treated HeLa cells”. Genes were not upregulated by RNA-seq. We changed the text accordingly.

- Line 729: “the HSF1 target gene set was extracted from Vilaboa et al. (REF,” Please complete the reference. Reference is completed accordingly.

- Fig. S6B – wrong number of cases for MS + LOH (?). This was corrected to n = 168, as in main Figure 6C.

- The description of Fig. S6D-E is not satisfactory. We expanded our description in the corresponding figure legends.

Rev1_I apologize that in my previous review I confused Fig. 4F and Fig. 4I (the concern was that two lower blots do not fit the upper ones). Nevertheless, it was corrected in the revised manuscript.

We thank the Reviewer for his/her very exact review and the thoughtful points raised.

Reviewer #2 (Remarks to the Author):

The authors have provided an extensive rebuttal to my previous comments and have mostly addressed my previous concerns. I now think the paper is suitable for publication.

We thank the Reviewer for his/her exclusively positive review.

Reviewer #3 (Remarks to the Author):

The revised manuscript is much improved and addresses the reviewers concerns. Overall, the manuscript provides clear evidence supporting a role for HSF-1 axis in mutp53 stabilization. In particular the newly added data further support GOF activity of the p53R248Q allele (new figures 7 and sup7). In addition, the strong correlation found between APC mutations and p53LOH, further supports HSF1 axis-mediated mutp53 stabilization promoting invasion.

The revised manuscript is much improved and addresses the concerns of reviewers. Overall, the manuscript provides clear evidence supporting a role of the HSF-1 axis in the stabilization of mutp53. In particular, the newly added data further supports the GOF activity of the p53R248Q allele (new Figures 7 and sup7). In addition, the strong correlation found between the APC and p53LOH mutations, enhances the role of the mutp53 stabilization / HSF1 axis in tumor invasion. Responses to reviewers clarify open questions and new figures reinforce key findings.

We thank the Reviewer for his/her exclusively positive review.

REVIEWERS' COMMENTS

Reviewer #1 (Remarks to the Author):

Assuming that HSF1 is necessary to maintain an elevated expression of chaperones in cancer cells, its silencing would lead to downregulation of HSF1-dependent genes. Such downregulation is observed by the Authors after nutlin treatment that is supposed to inhibit the HSF1 phosphorylation and activity. Assuming that HSF1 is at the end of the proposed signaling cascade, nutlin treatment should not affect HSF1-regulated genes in the absence of HSF1. In the Rebuttal-only Figure 1, HSPBP1, HSPA1A, and HSPH1 expression is shown. The conclusion is: "Since this experiment does not provide a clear answer, we did not include it in the manuscript." Why were ITGB3BP and CDC6 (whose expression is most inhibited by nutlin) not tested in HSF1 silenced cells?

The presented results indicate that the Authors cannot fully prove the involvement of HSF1 in the repression of the whole cytoprotective system after nutlin treatment. Therefore, HSF1 appears to be only one of a few/many possible elements at the end of the WTP53-p21-MLK3-MAPK signaling cascade, but evidence for its exclusive action is not convincing. The Authors used in analyses two gene sets: direct HSF1 target genes (a set from Mendillo et al.) and the set extracted from Vilaboa et al. The second set includes all genes upregulated after heat shock in HeLa cells, including those that are upregulated independently of HSF1. Thus, the conclusions should be focused more on a wide cytoprotective system that is ubiquitously activated in cancer cells (and inhibited by WTP53), not only on HSF1.

In my previous review, when asking for an explanation of the abbreviations, I meant terms like AOM/DSS or HUPKI, not gene names. They make it difficult to understand the text. Especially in the abstract "AOM/DSS-induced colorectal tumors" could be replaced by "chemically-induced colorectal tumors". Also "HUPKI" could be replaced or explained.

Point-by-Point Rebuttal Letter

We again thank Reviewer #1 for his/her positive evaluation of our paper. We revised our manuscript once more to address the remaining concerns and comply with the editorial requests.

Below is the Point-by-Point Rebuttal letter. Reviewer text in black, our answers in blue.

Added text in manuscript in RED.

Reviewer #1 (Remarks to the Author):

Question: Assuming that HSF1 is necessary to maintain an elevated expression of chaperones in cancer cells, its silencing would lead to downregulation of HSF1-dependent genes. Such downregulation is observed by the Authors after nutlin treatment that is supposed to inhibit the HSF1 phosphorylation and activity. Assuming that HSF1 is at the end of the proposed signaling cascade, nutlin treatment should not affect HSF1-regulated genes in the absence of HSF1.

Answer: The Reviewer expects to *not* see repression of HSF1 target genes in the Nutlin + siHSF1 knockdown condition, i.e. a 'rescue' of repressed HSF1 targets. However, in an epistatic signaling cascade with HSF1 at the end of the cascade, as we have here, isolated HSF1 knockdown alone - without Nutlin - intrinsically will give the *same* result. Therefore this approach will not proof causality of the WTP53-HSF1 axis as the Reviewer expects.

Question: In the Rebuttal-only Figure 1, HSPBP1, HSPA1A, and HSPH1 expression is shown. The conclusion is: "Since this experiment does not provide a clear answer, we did not include it in the manuscript." Why were ITGB3BP and CDC6 (whose expression is most inhibited by nutlin) not tested in HSF1 silenced cells?

Answer: ITGB3BP and CDC6 were not included here because of Reviewer's #1 earlier concern during the first revision. Reviewer #1 called both genes "... not-typical HSF1 targets (e.g. CDC6, ITGB3BP) ...".

Question: The presented results indicate that the Authors cannot fully prove the involvement of HSF1 in the repression of the whole cytoprotective system after nutlin treatment. Therefore, HSF1 appears to be only one of a few/many possible elements at the end of the WTP53-p21-MLK3-MAPK signaling cascade, but evidence for its exclusive action is not convincing. The Authors used in analyses two gene sets: direct HSF1 target genes (a set from Mendillo et al.) and the set extracted from Vilaboa et al. The second set includes all genes upregulated after heat shock in HeLa cells, including those that are upregulated independently of HSF1. Thus, the conclusions should be focused more on a wide cytoprotective system that is ubiquitously activated in cancer cells (and inhibited by WTP53), not only on HSF1.

Answer: We would like to point out that we do not claim any exclusive action of HSF1 in the repression of the whole cytoprotective system after Nutlin treatment. There might indeed be

additional cytoprotective pathways at the end of the WTp53-p21-MLK3-MAPK signaling cascade which remain to be identified in future research. To avoid any misunderstanding, we now changed the wording and toned down at 4 different spots in the Discussion.

In my previous review, when asking for an explanation of the abbreviations, I meant terms like AOM/DSS or HUPKI, not gene names. They make it difficult to understand the text. Especially in the abstract “AOM/DSS-induced colorectal tumors” could be replaced by “chemically-induced colorectal tumors”. Also “HUPKI” could be replaced or explained.

Answer: We corrected the text accordingly and defined AOM/DSS in Introduction. The term HUPKI was not necessary and therefore eliminated; the allele is called ‘humanized’ and is referenced.